# ANALYZING NEURAL SCALING LAWS IN TWO-LAYER NETWORKS WITH POWER-LAW DATA SPECTRA

**Roman Worschech**[1,2]* **Bernd Rosenow**[1]
[1]Institut für Theoretische Physik, Universität Leipzig, Brüderstraße 16, 04103 Leipzig, Germany
[2]Max Planck Institute for Mathematics in the Sciences, Inselstraße 22, 04103 Leipzig, Germany

## ABSTRACT

Neural scaling laws describe how the performance of deep neural networks scales with key factors such as training data size, model complexity, and training time, often following power-law behaviors over multiple orders of magnitude. Despite their empirical observation, the theoretical understanding of these scaling laws remains limited. In this work, we employ techniques from statistical mechanics to analyze one-pass stochastic gradient descent within a student-teacher framework, where both the student and teacher are two-layer neural networks. Our study primarily focuses on the generalization error and its behavior in response to data covariance matrices that exhibit power-law spectra. For linear activation functions, we derive analytical expressions for the generalization error, exploring different learning regimes and identifying conditions under which power-law scaling emerges. Additionally, we extend our analysis to non-linear activation functions in the feature learning regime, investigating how power-law spectra in the data covariance matrix impact learning dynamics. Importantly, we find that the length of the symmetric plateau depends on the number of distinct eigenvalues of the data covariance matrix and the number of hidden units, demonstrating how these plateaus behave under various configurations. In addition, our results reveal a transition from exponential to power-law convergence in the specialized phase when the data covariance matrix possesses a power-law spectrum. This work contributes to the theoretical understanding of neural scaling laws and provides insights into optimizing learning performance in practical scenarios involving complex data structures.

## 1 INTRODUCTION

Recent empirical studies have revealed that the performance of state-of-the-art deep neural networks, trained on large-scale real-world data, can be predicted by simple phenomenological functions Hestness et al. (2017); Maloney et al. (2022); Hoffmann et al. (2022); Porian et al. (2024). Specifically, the network's error decreases in a power-law fashion with respect to the number of training examples, model size, or training time, spanning many orders of magnitude. This observed phenomenon is encapsulated by neural scaling laws, which describe how model performance varies as key scaling factors change. Interestingly, the performance improvement due to one scaling factor is often limited by another, suggesting the presence of bottleneck effects Kaplan et al. (2020). Understanding these scaling laws theoretically is crucial for practical applications such as optimizing architectural design and selecting appropriate hyperparameters. However, the fundamental reasons behind the emergence of neural scaling laws have mainly been explored for linear Lin et al. (2024) and random feature models Bahri et al. (2024), and a more comprehensive theory is still absent.

**Scope of Study**. In this work, we employ techniques from statistical mechanics to analyze one-pass stochastic gradient descent within a student-teacher framework. Both networks are two-layer neural networks: the student has $K$ hidden neurons, the teacher has $M$, and we train only the student's input-to-hidden weights, realizing a so-called committee machine Biehl & Schwarze (1995). We begin our analysis with linear activation functions for both networks and then extend it to non-linear activation functions, focusing on the feature learning regime where the student weights undergo

---

*Corresponding author: `r.worschech@web.de`

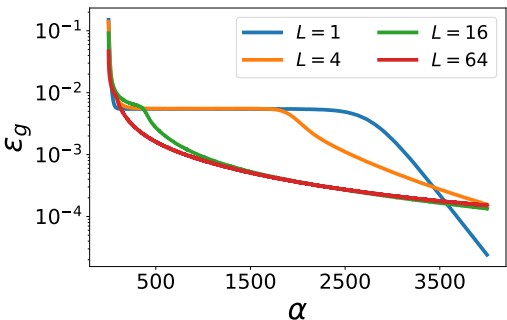

Figure 1: Generalization error $\epsilon_g$ as a function of $\alpha$ for $K = M = 2$, $\eta = 0.1$, $\beta = 1$, $\sigma_J = 0.01$, and $N = 1024$, with varying numbers of distinct eigenvalues $L$. As $L$ increases, the plateau length decreases until it disappears. Additionally, with increasing $L$, the convergence of the asymptotic generalization error slows down, transitioning from exponential to power-law scaling in the early asymptotic phase.

significant changes during training. Our primary focus is on analyzing the generalization error $\epsilon_g$ by introducing order parameters that elucidate the relationships between the student and teacher weights. Despite the diversity of datasets across various learning domains, a critical commonality is that their feature-feature covariance matrices often exhibit power-law spectra Maloney et al. (2022). To model realistic data, we therefore utilize Gaussian-distributed inputs with covariance matrices that display power-law spectra.

**White Noise vs. Power-Law Spectra**.   The student-teacher setup with isotropic input data has been extensively studied and is well-understood in the literature Saad & Solla (1995). In the realizable scenario where $K = M$, the generalization error typically undergoes three distinct phases: a rapid learning phase, a plateau phase, and an exponentially decaying phase with time $\alpha$. Introducing a power-law spectrum in the covariance matrix leads to observable changes in the plateau's height and duration, along with a slowdown in the convergence towards zero generalization error. Notably, as the number of distinct eigenvalues $L$ in the data covariance spectrum increases, the plateau shortens, and the convergence to perfect learning becomes progressively slower, as depicted in Figure 1. This observation indicates a potential transition from exponential decay to power-law scaling in the generalization error over time. Identifying and understanding this transition is a critical focus of our investigation. Our main **contributions** are:

- For linear activation functions, we derive an exact analytical expression for the generalization error as a function of training time $\alpha$ and the power-law exponent $\beta$ of the covariance matrix. We characterize different learning regimes for the generalization error and analyze the conditions under which power-law scaling emerges.

- In addition, for linear activation functions, we demonstrate a scaling law in the number of trainable student parameters, effectively reducing the input dimension of the network. This power-law is different from the power-law characterizing the training time dependence.

- We derive an analytical formula for the dependence of the plateau length on the number of distinct eigenvalues and the power-law exponent $\beta$ of the covariance matrix, illustrating how these plateaus behave under different configurations.

- We investigate the asymptotic learning regime for non-linear activation functions and find that, in the realizable case with $M = K$, the convergence to perfect learning shifts from an exponential to a power-law regime when the data covariance matrix has a power-law spectrum.

## 2   RELATED WORK

**Theory of Neural Scaling Laws for Linear Activation Functions**.   Previous studies on neural scaling laws have primarily focused on random feature models or linear (ridge) regression with

power-law features Wei et al. (2022). In particular, Maloney et al. (2022); Paquette et al. (2024) and Atanasov et al. (2024) analyzed random feature models for linear features and ridge regression, employing techniques from random matrix theory. Bahri et al. (2024) examined random feature models for kernel ridge regression within a student-teacher framework using techniques from statistical mechanics. In their analysis, either the number of parameters or the training dataset size was considered infinite, leading to scaling laws in the test loss with respect to the remaining finite quantity. Bordelon et al. (2024b) studied random feature models with randomly projected features and momentum, trained using gradient flow. Using a dynamical mean field theory approach, they derived a "bottleneck scaling" where only one of time, dataset size, or model size is finite while the other two quantities approach infinity. Additionally, Hutter (2021) investigated a binary toy model and found non-trivial scaling laws with respect to the number of training examples.

Bordelon & Pehlevan (2022) studied one-pass stochastic gradient descent for random feature models, deriving a scaling law for the test error over time in the small learning rate regime. Similarly, Lin et al. (2024) investigated infinite-dimensional linear regression under one-pass stochastic gradient descent, providing insights through a statistical learning theory framework. They derived upper and lower bounds for the test error, demonstrating scaling laws with respect to the number of parameters and dataset size under different scaling exponents.

Building upon the work of Lin et al. (2024) and Bordelon & Pehlevan (2022), we also consider one-pass stochastic gradient descent. However, our study extends to both linear and non-linear neural networks where we train the weights used in the pre-activations (i.e., feature learning), and use fixed hidden-to-output connections. Unlike Bordelon & Pehlevan (2022), we extend the analysis for linear activation functions to general learning rates and varying numbers of input neurons. Additionally, we derive upper and lower bounds for the time interval over which the generalization error exhibits power-law behavior. A significant difference from previous works is our focus on feature learning, where all pre-activation weights are trainable. In this regime, certain groups of student weights, organized by student vectors, begin to imitate teacher vectors during the late training phase, leading to specialization.

Other theoretical studies have explored different aspects of scaling laws. Some have focused on learnable network skills or abilities that drive the decay of the loss Arora & Goyal (2023); Michaud et al. (2023); Caballero et al. (2023); Nam et al. (2024). Others have compared the influence of synthetic data with real data Jain et al. (2024) or investigated model collapse phenomena Dohmatob et al. (2024b;a). Further works studying correlated and realistic input data are Goldt et al. (2020); Loureiro et al. (2021); Cagnetta et al. (2024); Cagnetta & Wyart (2024).

**Statistical Mechanics Approach**. Analytical studies using the statistical mechanics framework for online learning have traditionally focused on uncorrelated input data or white noise. Saad & Solla (1995) first introduced differential equations for two-layer neural networks trained via stochastic gradient descent on such data. Building upon this, Yoshida & Okada (2019) recently expanded these models to include Gaussian-correlated input patterns, deriving a set of closed-form differential equations. Their research primarily involved numerically solving these equations for covariance matrices with up to two distinct eigenvalues, exploring how the magnitudes of the eigenvalues affect the plateau's length and height. In our study, we extend this hierarchy of differential equations to investigate the dynamics of order parameters for data covariance matrices with power-law spectra, considering $L$ distinct eigenvalues.

## 3 SETUP

**Dataset**. We consider a student network trained on outputs generated by a teacher network, using $p$ input examples $\boldsymbol{\xi}^\mu \in \mathbb{R}^N$, where $\mu = 1, \ldots, p$. Each input $\boldsymbol{\xi}^\mu$ is drawn from a correlated Gaussian distribution $\mathcal{N}(0, \boldsymbol{\Sigma})$, with covariance matrix $\boldsymbol{\Sigma} \in \mathbb{R}^{N \times N}$. Although the covariance matrix generally has $N$ eigenvalues, we assume it has only $L$ distinct eigenvalues, each occurring with multiplicity $N/L$, where $1 \leq L \leq N$ and $N/L$ is an integer. The eigenvalues follow a power-law distribution:

$$\lambda_l = \frac{\lambda_+}{l^{1+\beta}} \tag{1}$$

where $\beta > 0$ is the power-law exponent of the covariance matrix, $\lambda_+ = \lambda_1$ is the largest eigenvalue, and $l \in \{1, \ldots, L\}$. We choose $\lambda_+$ such that the total variance satisfies $\sum_{l=1}^{L} \left(\frac{N}{L}\lambda_l\right) = N$, ensuring that the pre-activations of the hidden neurons remain of order one in our setup.

**Student-Teacher Setup**. The student is a soft committee machine – a two-layer neural network with an input layer of $N$ neurons, a hidden layer of $K$ neurons, and an output layer with a single neuron. In the statistical mechanics framework, we represent the weights between the input layer and the hidden layer as vectors. Specifically, the connection between the input layer and the $i$-th hidden neuron is represented by the student vector $\boldsymbol{J}_i \in \mathbb{R}^N$. Thus, we have $K$ student vectors $\boldsymbol{J}_i$, each representing the weights connecting the entire input layer to one of the hidden neurons. The pre-activation received by the $i$-th hidden neuron is defined as $x_i = \frac{1}{\sqrt{N}}\boldsymbol{\xi}^\mu \cdot \boldsymbol{J}_i$. The overall output of the student is given by

$$\sigma(\boldsymbol{J}, \boldsymbol{\xi}) = \frac{\sqrt{M}}{K}\sum_{i=1}^{K} g\left(x_i\right), \tag{2}$$

where $g(x_i)$ is the activation function, and the output weights are set to $\sqrt{M}/K$. In this setup, we train the student vectors $\boldsymbol{J}_i$ and keep the hidden-to-output weights fixed. The teacher network has the same architecture but with $M$ hidden neurons, and its weights are characterized by the teacher vectors $\boldsymbol{B}_n \in \mathbb{R}^N$. The pre-activations for the teacher are $y_n = \frac{1}{\sqrt{N}}\boldsymbol{\xi}^\mu \cdot \boldsymbol{B}_n$, and its overall output is $\zeta(\boldsymbol{B}, \boldsymbol{\xi}) = \sum_{n=1}^{M} g\left(y_n\right)$. We initialize the student and teacher vectors from normal distributions: $J_{ia} \sim \mathcal{N}(0, \sigma_J^2)$ and $B_{na} \sim \mathcal{N}(0, 1)$, where $\sigma_J^2$ is the variance of the student weights and $a \in \{1, \ldots, N\}$. To quantify the discrepancy between the student's output and the teacher's output, we use the squared loss function $\epsilon = \frac{1}{2}[\zeta - \sigma]^2$. Our main focus is the generalization error $\epsilon_g = \langle \epsilon(\boldsymbol{\xi}) \rangle_{\boldsymbol{\xi}}$, which measures the typical error of the student on new inputs. Throughout this work, we consider the error function as our non-linear activation function $g(x) = \mathrm{erf}\left(\frac{x}{\sqrt{2}}\right)$.

**Transition from Microscopic to Macroscopic Formalism**. Rather than computing expectation values directly over the input distribution, we consider higher-order pre-activations defined as $x_i^{(l)} = \boldsymbol{\xi}^\mu (\boldsymbol{\Sigma})^l \boldsymbol{J}_i / \sqrt{N}$ and $y_n^{(l)} = \boldsymbol{\xi}^\mu (\boldsymbol{\Sigma})^l \boldsymbol{B}_n / \sqrt{N}$, as suggested in Yoshida & Okada (2019). Here, $(\boldsymbol{\Sigma})^l$ denotes the $l$-th power of the covariance matrix, and we define $(\boldsymbol{\Sigma})^0 = \boldsymbol{I}$. In the thermodynamic limit $N \to \infty$, these higher-order pre-activations become Gaussian random variables with zero mean and covariances given by: $\langle x_i^{(k)} x_j^{(l)} \rangle = \frac{\boldsymbol{J}_i(\boldsymbol{\Sigma})^p \boldsymbol{J}_j}{N} := Q_{ij}^{(p)}$, $\langle x_i^{(k)} y_n^{(l)} \rangle = \frac{\boldsymbol{J}_i(\boldsymbol{\Sigma})^p \boldsymbol{B}_n}{N} := R_{in}^{(p)}$, and $\langle y_n^{(k)} y_m^{(l)} \rangle = \frac{\boldsymbol{B}_n(\boldsymbol{\Sigma})^p \boldsymbol{B}_m}{N} := T_{nm}^{(p)}$, where $p = k + l + 1$. The higher-order order parameters $Q_{ij}^{(l)}$, $R_{in}^{(l)}$, and $T_{nm}^{(l)}$ capture the relationships between the student and teacher weights at different levels. By expressing the generalization error as a function of these order parameters, we transition from a microscopic view – focused on individual weight components – to a macroscopic perspective centered on the relationships between student and teacher vectors without detailing their exact components.

**Dynamical Equations**. During the learning process, we update the student vectors $\boldsymbol{J}_i$ using stochastic gradient descent after each presentation of an input example:

$$\boldsymbol{J}_i^{\mu+1} - \boldsymbol{J}_i^\mu = -\eta \nabla_{\boldsymbol{J}_i} \epsilon\left(\boldsymbol{J}_i^\mu, \boldsymbol{\xi}^\mu\right), \tag{3}$$

where $\eta$ is the learning rate. In the thermodynamic limit, as $p, N \to \infty$ while maintaining a finite ratio $\alpha = p/N$, Yoshida & Okada (2019) derived a set of hierarchical differential equations describing the dynamics of the order parameters under stochastic gradient descent. Applying these findings to our specific setup, we obtain the following differential equations:

$$\frac{d\boldsymbol{R}^{(l)}}{d\alpha} = \frac{\eta}{K}F_1\left(\boldsymbol{R}^{(1)}, \boldsymbol{Q}^{(1)}, \boldsymbol{R}^{(l+1)}, \boldsymbol{Q}^{(l+1)}\right)$$

$$\frac{d\boldsymbol{Q}^{(l)}}{d\alpha} = \frac{\eta}{K}F_2\left(\boldsymbol{R}^{(1)}, \boldsymbol{Q}^{(1)}, \boldsymbol{R}^{(l+1)}, \boldsymbol{Q}^{(l+1)}\right) + \frac{\eta^2}{K^2}\nu_{l+1}F_3\left(\boldsymbol{R}^{(1)}, \boldsymbol{Q}^{(1)}\right), \tag{4}$$

where $\nu_l = \frac{1}{N}\sum_{k=1}^{N}\lambda_k^l$. The functions $F_1$, $F_2$, and $F_3$ are defined in Appendix A. The transition from Eq. (3) to Eq. (4) represents a shift from discrete-time updates indexed by $\mu$ to a continuous-time framework where $\alpha$ serves as a continuous time variable.

At this stage, the differential equations are not closed because the left-hand sides of Eqs. (4) involve derivatives of the $l$-th order parameters, while the right-hand sides depend on the next higher-order

parameters $\boldsymbol{R}^{(l+1)}$ and $\boldsymbol{Q}^{(l+1)}$. To close the system of equations, we employ the Cayley–Hamilton theorem, which states that every square matrix satisfies its own characteristic equation. Specifically, for the covariance matrix $\boldsymbol{\Sigma}$, the minimal polynomial is given by $P(\boldsymbol{\Sigma}) := \prod_{k=1}^{L} (\boldsymbol{\Sigma} - \lambda_k \boldsymbol{I}) = \sum_{k=0}^{L} c_k \boldsymbol{\Sigma}^k = 0$, where $c_k$ are the coefficients of the polynomial, and $\lambda_k$ are the distinct eigenvalues of $\boldsymbol{\Sigma}$. Consequently, we can express the highest-order order parameters in terms of lower-order ones: $\boldsymbol{R}^{(L)} = -\sum_{l=0}^{L-1} c_l \boldsymbol{R}^{(l)}$, $\boldsymbol{Q}^{(L)} = -\sum_{l=0}^{L-1} c_l \boldsymbol{Q}^{(l)}$, and $\boldsymbol{T}^{(L)} = -\sum_{l=0}^{L-1} c_l \boldsymbol{T}^{(l)}$. By substituting these expressions back into the differential equations, we close the system, resulting in $(KM + K^2) \times L$ coupled differential equations. Further details on the derivation of these differential equations are provided in Appendix A.

## 4 LINEAR ACTIVATION FUNCTION

### 4.1 SOLUTION OF ORDER PARAMETERS

For the linear activation function, a significant simplification occurs: the generalization error becomes independent of the sizes of the student and teacher networks. Specifically, we can replace the student and teacher vectors with their weighted sums, effectively acting as single resultant vectors. By defining $\tilde{\boldsymbol{B}} = \frac{1}{\sqrt{M}} \sum_n^M \boldsymbol{B}_n$, the student effectively learns this combined teacher vector. Consequently, we focus on the case where $K = M = 1$. In this scenario, the generalization error simplifies to $\epsilon_g = \frac{1}{2} \left( Q^{(1)} - 2R^{(1)} + T^{(1)} \right)$, which depends only on the first-order order parameters. Therefore, our main interest lies in solving the dynamics of these first-order parameters. Since we have only one student and one teacher vector, we represent the order parameters in vector form $\boldsymbol{R} = \left( R^{(0)}, R^{(1)}, ..., R^{(L-1)} \right)^\top$, $\boldsymbol{Q} = \left( Q^{(0)}, Q^{(1)}, ..., Q^{(L-1)} \right)^\top$ and $\boldsymbol{T} = \left( T^{(0)}, T^{(1)}, ..., T^{(L-1)} \right)^\top$. Using this setup and notation, along with Eq. (4), we derive the following dynamical equation:

$$\frac{d}{d\alpha} \begin{pmatrix} \boldsymbol{R} \\ \boldsymbol{Q} \end{pmatrix} = \eta \begin{pmatrix} \boldsymbol{A}_1 & \boldsymbol{0}_{L \times L} \\ -2\boldsymbol{A}_1 - 2\eta \boldsymbol{U} & 2\boldsymbol{A}_1 + \eta \boldsymbol{U} \end{pmatrix} \begin{pmatrix} \boldsymbol{R} \\ \boldsymbol{Q} \end{pmatrix} + \eta \begin{pmatrix} \boldsymbol{u} \\ \eta \boldsymbol{u} \end{pmatrix}, \tag{5}$$

where $\boldsymbol{u} = \left( T^{(1)}, T^{(2)}, \ldots, T^{(L)} \right)^\top$, $\boldsymbol{U} = \boldsymbol{u} \boldsymbol{e}_2^\top$, and $\boldsymbol{e}_2 = (0, 1, 0, \ldots, 0)^\top$. The matrix $\boldsymbol{A}_1 \in \mathbb{R}^{L \times L}$ is defined in Appendix B.1. From Eq. (5), we observe that the differential equations governing the higher-order student-teacher order parameters $\boldsymbol{R}$ can be solved independently of the student-student parameters $\boldsymbol{Q}$. Therefore, to understand the dynamical behavior of $\boldsymbol{R}(\alpha)$, we need to determine the eigenvalues of $\boldsymbol{A}_1$, and for the asymptotic solution, we require its inverse. Additionally, the solution for the student-student order parameters $\boldsymbol{Q}(\alpha)$ depends on $\boldsymbol{R}(\alpha)$ and the spectrum of $\boldsymbol{A}_1 + \eta \boldsymbol{U}$. In Appendix B.1, we derive an expression for the generalization error averaged over the teacher and initial student entries $B_a$ and $J_a^0$:

$$\langle \epsilon_g \rangle_{J_a^0, B_a} = \frac{\left(1 + \sigma_J^2\right)}{2L} \sum_{k=1}^{L} b_k \tilde{\lambda}_k \exp(-2\eta \tilde{\lambda}_k \alpha), \tag{6}$$

where $\tilde{\lambda}_k$ are the eigenvalues of $\boldsymbol{A}_1 + \eta \boldsymbol{U}$, $b_k = \sum_{l=1}^{L} \left( \boldsymbol{W}^{-1} \right)_{kl} T^{(l)}$, and $\boldsymbol{W}$ contains the eigenvectors of $\boldsymbol{A}_1 + \eta \boldsymbol{U}$. This equation generally requires numerical evaluation. However, in the regime of small learning rates $\eta$, where we retain terms up to $\mathcal{O}(\eta)$ in Eq. (5), we can determine the spectra of all involved matrices analytically and solve the differential equations. The solutions for the first-order order parameters are then given by

$$\langle R^{(1)} \rangle_{J_a^0, B_a} = 1 - \frac{1}{L} \sum_k^L \lambda_k \exp\left(-\eta \lambda_k \alpha\right),$$

$$\langle Q^{(1)} \rangle_{J_a^0, B_a} = 1 + \frac{1 + \sigma_J^2}{L} \sum_k^L \lambda_k \exp\left(-2\eta \lambda_k \alpha\right) - \frac{2}{L} \sum_k^L \lambda_k \exp\left(-\eta \lambda_k \alpha\right), \tag{7}$$

and the generalization error becomes

$$\langle \epsilon_g \rangle_{J_a^0, B_a} \underset{\eta \to 0}{=} \frac{1 + \sigma_J^2}{2L} \sum_{k=1}^{L} \lambda_k \exp(-2\eta \lambda_k \alpha). \tag{8}$$

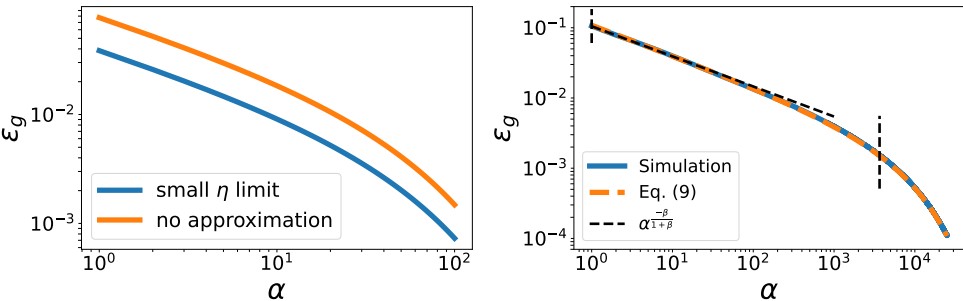

Figure 2: Generatlization error $\epsilon_g$ for linear activation function. Left: $\epsilon_g$ evaluated using Eq. (8) (blue) and Eq. (6) (orange) for $N = 128$, $K = M = 1$, $\sigma_J^2 = 1$, $\beta = 1$, and $\eta = 1$. Right: $\epsilon_g$ evaluated using Eq. (9) (dashed orange) compared to simulations experiments averaged over 15 random initializations (solid blue), with $N = L = 1024$, $\beta = 0.75$, $\eta = 0.01$, and $\sigma_J = 0.01$. The dashed vertical lines indicate the time window where the generalization error scales as $\epsilon_g \propto \alpha^{\frac{-\beta}{1+\beta}}$.

Here, $\lambda_k$ are the distinct eigenvalues of the data covariance matrix as defined in Eq. (1). Figure 2 compares the generalization error obtained from the exact solution in Eq. (6) with the small learning rate approximation in Eq. (8). We observe that the generalization error without approximations consistently lies above the small learning rate solution. This discrepancy arises from the fluctuations in the stochastic gradient descent trajectory, which become more pronounced at larger learning rates.

## 4.2 SCALING WITH TIME

To evaluate the sum on the right-hand side of Eq. (8), we employ the Euler-Maclaurin approximation, which allows us to approximate the sum by an integral. In Appendix B.2, we derive the following approximation for the generalization error:

$$\langle \epsilon_g(\alpha) \rangle_{J_a^0, B_a} \underset{\eta \to 0}{\approx} \lambda_+ \frac{1 + \sigma_J^2}{2L} \frac{(2\eta\lambda_+\alpha)^{-\frac{\beta}{1+\beta}}}{1+\beta} \left[ \Gamma\left( \frac{\beta}{1+\beta}, \frac{2\eta\lambda_+\alpha}{L^{\beta+1}} \right) - \Gamma\left( \frac{\beta}{1+\beta}, 2\eta\lambda_+\alpha \right) \right], \quad (9)$$

where $\Gamma(s, x)$ is the incomplete gamma function. This expression reveals that the generalization error exhibits a power-law scaling within the time window $\frac{1}{2\eta\lambda_+} < \alpha < \frac{L^{1+\beta}}{2\eta\lambda_+} \left( \frac{\beta}{10(1+\beta)} \Gamma\left( \frac{\beta}{1+\beta} \right) \right)^{\frac{1+\beta}{\beta}}$. In this regime, the generalization error scales as $\epsilon_g(\alpha) \propto \alpha^{-\frac{\beta}{1+\beta}}$, aligning with the results of Bordelon & Pehlevan (2022) and Lin et al. (2024) for the random feature model. The right panel of Figure 2 illustrates our analytical prediction from Eq. (9), alongside the generalization error observed in a student neural network trained on Gaussian input data with a power-law spectrum. Additional numerical analyses are provided in Appendix B.2.

## 4.3 FEATURE SCALING

Students typically learn directions associated with the largest eigenvalues of the data covariance matrix more rapidly Advani et al. (2020). To model this behavior, we assume the student can learn at most $N_l \leq L = N$ distinct eigenvalues of the data covariance matrix. Consequently, only the first $N_l$ entries of the student vector are trainable, while the remaining $N - N_l$ entries remain fixed at their initial random values. Our objective is to examine how the generalization error scales as the student explores more eigendirections of the data covariance matrix. We first consider a diagonal covariance matrix $\Sigma$, where each student vector entry $J_k$ independently converges to the corresponding teacher entry $B_k$ at a rate proportional to an eigenvalue $\lambda_k$. (see Appendix B.3).

Figure 3 displays the generalization error as a function of $\alpha$ for various values of $N_l$. We observe that the generalization error approaches a limiting asymptotic value $\epsilon_{g,\text{asymp}}$. In Appendix B.3, we derive the following expression for the expected generalization error in this model:

$$\langle \epsilon_g \rangle_{J_k^0, B_k} \underset{\eta \to 0}{=} \frac{1 + \sigma_J^2}{2L} \left[ \sum_{k=1}^{N_l} \lambda_k \exp\left( -2\eta\lambda_k\alpha \right) + \sum_{k=N_l+1}^{L} \lambda_k \right]. \quad (10)$$

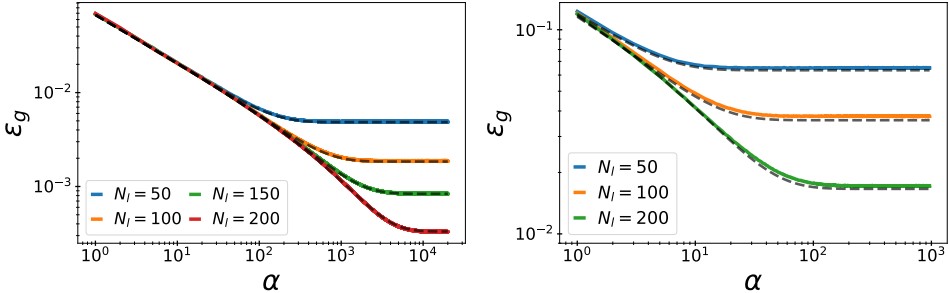

Figure 3: Generalization error $\epsilon_g$ for different trainable input dimensions $N_l$ of the student network. Left: $\epsilon_g$ as a function of $\alpha$ for various $N_l$, with $L = N = 256$, $K = M = 1$, $\sigma_J = 0.01$, $\eta = 0.05$, and $\beta = 1$. The student network is trained on synthetic data and the teacher's outputs. Right: $\epsilon_g$ as a function of $\alpha$, with $L = N = 1024$, $K = M = 1$, $\sigma_J = 0.01$, and $\eta = 0.05$. The student network is trained on the CIFAR-5m dataset Nakkiran et al. (2021) using the teacher's outputs. We estimate the scaling exponent $\beta \approx 0.3$ for this dataset. For the theoretical predictions, the empirical data spectrum is used to evaluate Eq. (11). Both plots compare the simulation results (solid curves) to the theoretical prediction from Eq. (11) (black dashed lines). For both plots, the generalization error is averaged over 50 random initializations of the student and teacher vectors.

Using the Euler-Maclaurin formula, we approximate the sums by integrals and find:

$$\langle \epsilon_g \left( N_l, \alpha \right) \rangle_{J_a^0, B_a} \underset{\eta \to 0}{\approx} \frac{1 + \sigma^2}{2} \frac{\lambda_+}{\beta L} \left( \frac{1}{N_l^\beta} - \frac{1}{L^\beta} \right) + \langle \epsilon_g(\alpha) \rangle_{J_a^0, B_a} . \tag{11}$$

From this, we derive the asymptotic generalization error as $\epsilon_{g,\text{asymp}} \approx \frac{1+\sigma^2}{2} \frac{\lambda_+}{\beta L} \left( \frac{1}{N_l^\beta} - \frac{1}{L^\beta} \right)$. Thus, when $L^\beta > N_l^\beta$, we find a power-law scaling of the asymptotic generalization error with respect to the number of learned features: $\epsilon_{g,\text{asymp}} \sim \frac{1}{N_l^\beta}$. A similar scaling result for feature scaling is presented in Maloney et al. (2022) for random feature models. However, our scaling exponent for the dataset size (parameterized by $\alpha$) differs from that for the number of features. In Appendix B.3, we analyze the student network trained with a non-diagonal data covariance matrix. In this setting, we find the same power-law exponent $\epsilon_{g,\text{asymp}} \sim \frac{1}{N_l^\beta}$.

## 5 NON-LINEAR ACTIVATION FUNCTION

### 5.1 PLATEAU

As discussed in the introduction and illustrated in Fig. 1, both the length and height of the plateau are influenced by the number of distinct eigenvalues in the data covariance matrix. Specifically, as the number of distinct eigenvalues increases, the plateau becomes shorter and can eventually disappear. In this section, we present our findings that explain the underlying causes of this behavior for the case where $K = M$. Biehl et al. (1996) derived a formula to estimate the plateau length $\alpha_P$ for a soft committee machine trained via stochastic gradient descent with randomly initialized student vectors. We adopt their heuristically derived formula for our setup, which takes the form

$$\alpha_P - \alpha_0 = \tau_{\text{esc}} \left( D - \frac{1}{2} \ln \left( \sigma_J^2 \right) + \frac{1}{2} \ln \left( N \right) \right) , \tag{12}$$

where $D$ is a constant of order $\mathcal{O}(1)$ that depends on the variances at initialization and during the plateau phase, $\alpha_0$ is an arbitrary starting point on the plateau, and $\tau_{\text{esc}}$ represents the escape time from the plateau. Our goal is to show how the escape time $\tau_{\text{esc}}$ is modified when the dataset has a power-law spectrum. However, there is not a single plateau or plateau length. As shown numerically by Biehl et al. (1996), multiple plateaus can exist, and their number depends on factors such as the network sizes $K$ and $M$, as well as hyperparameters like the learning rate $\eta$. To investigate how the plateau lengths depend on the number of distinct eigenvalues, we focus on the differential equations for the error function activation up to order $\mathcal{O}(\eta)$. This corresponds to the small learning rate regime, although the associated plateau behavior can occur for intermediate learning rates as well.

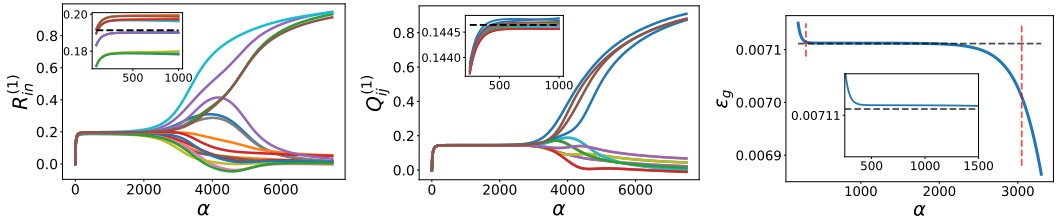

Figure 4: Symmetric plateau for a non-linear activation function. Left and center: Plateau behavior of the order parameters for $L = 10$, $N = 7000$, $\sigma_J = 0.01$, $\eta = 0.1$, and $M = K = 4$, using one random initialization of the student and teacher vectors. We solve the differential equations in the small learning rate regime, retaining terms up to $\mathcal{O}(\eta)$. The insets display the higher-order order parameters at the plateau. For the student-teacher order parameters, we observe $M$ distinct plateau heights, while the student-student order parameters exhibit a single plateau height with minor statistical deviations in the matrix entries $Q_{ij}^{(l)}$. The dashed horizontal lines in the insets correspond to the plateau heights predicted by Eq. (13). Right: Corresponding generalization error $\epsilon_g$ for the same setup. The vertical dashed lines indicate the estimated plateau length based on Eqs. (12) and (15).

### 5.1.1 PLATEAU HEIGHT

In contrast to isotropic input correlations, the higher-order order parameters in our setup are no longer self-averaging, resulting in more complex learning dynamics. For the teacher-teacher order parameters, we find the expectation value $\left\langle T_{nm}^{(l)} \right\rangle = \delta_{nm} \frac{1}{N} \text{Tr} \left( \Sigma^l \right)$ and the variance $\text{Var} \left( T_{nm}^{(l)} \right) = \frac{(\delta_{nm}+1)}{N^2} \sum_{k=1}^{N} \lambda_k^{2l}$. In Appendix C.2 we show that the teacher-teacher order parameters $T_{nn}^{(l)}$ with $l > 0$ are only self-averaging when $L/N \to 0$ in the thermodynamic limit $N \to \infty$, while for a finite ratio $L/N$ their variance does not decrease with increasing input dimension $N$. Consequently, in the latter case, the plateau height and length can fluctuate between different initializations, even in the thermodynamic limit. Figure 4 displays the first-order order parameters as functions of $\alpha$. For the student-teacher order parameters, we observe $M$ distinct plateau heights, while the student-student order parameters exhibit a single plateau height. In Appendix C.3, we discuss that these unique plateau heights are determined by the sum of off-diagonal elements $d_n^{(l)} = \sum_{m,m \neq n}^{M-1} T_{nm}^{(l)}$ for each row $n$ of the teacher-teacher matrix. To simplify the analysis, we assume that all diagonal elements are equal to $T^{(l)}$, and all off-diagonal elements are given by $T_{nm}^{(l)} = \frac{1}{M-1} D^{(l)}$, where $D^{(l)}$ represents the average sum of off-diagonal entries. This approximation captures the general behavior of the plateaus. By considering the stationary solutions to Eqs. (4), we find the fixed points for $l = 1$

$$R^{*(1)} = \frac{1}{\sqrt{\frac{M}{T^{(1)}+D^{(1)}} \left( \frac{MT^{(1)}}{T^{(1)}+D^{(1)}} \left(1 + \frac{1}{T^{(1)}}\right) - 1 \right)}}, \quad Q^{*(1)} = \frac{1}{\frac{MT^{(1)}}{T^{(1)}+D^{(1)}} \left(1 + \frac{1}{T^{(1)}}\right) - 1}. \quad (13)$$

Expressions for the fixed points of higher-order order parameters are provided in Appendix C.3.

### 5.1.2 ESCAPE FROM THE PLATEAU

To escape from the plateau, the symmetry in each order $l$ of the order parameters must be broken. To model this symmetry breaking, we introduce parameters $S^{(l)}$ and $C^{(l)}$, which indicate the onset of specialization for the student-teacher and student-student order parameters, respectively. Specifically, we use the parametrization $R_{im}^{(l)} = R^{(l)}\delta_{im} + S^{(l)}(1 - \delta_{im})$ and $Q_{ij}^{(l)} = Q^{(l)}\delta_{ij} + C^{(l)}(1 - \delta_{ij})$. To study the onset of specialization, we introduce small perturbation parameters $r^{(l)}$, $s^{(l)}$, $q^{(l)}$, and $c^{(l)}$ to represent deviations from the plateau values: $R^{(l)} = R^{*(l)} + r^{(l)}$, $S^{(l)} = S^{*(l)} + s^{(l)}$, $Q^{(l)} = Q^{*(l)} + q^{(l)}$, and $C^{(l)} = C^{*(l)} + c^{(l)}$, where $S^{*(l)} = R^{*(l)}$ and $C^{*(l)} = Q^{*(l)}$. Therefore, instead of analyzing the dynamics of the order parameters directly, we focus on the dynamics of these perturbative parameters and linearize the differential equations given in Eq. (4).

In Appendix C.4, we demonstrate that, due to the structure of the leading eigenvectors of the dynamical system, we can set $c^{(l)} = q^{(l)} = \frac{2T^{(l)}}{T^{(1)}+D^{(1)}} R^{*(l)} \left( r^{(l)} + (M-1)s^{(l)} \right)$ and $s^{(l)} = \frac{-1}{(M-1)} r^{(l)}$. This

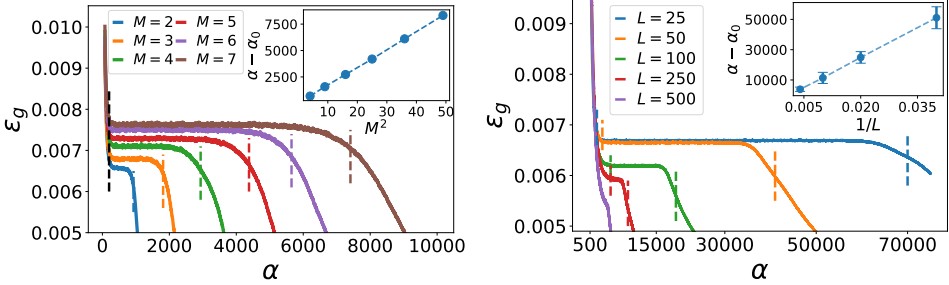

Figure 5: Plateau behavior of the generalization error $\epsilon_g$ from simulations for a random initialization of student and teacher vectors. Left: $\epsilon_g$ for various student-teacher sizes for $L = 4$, $N = 1024$, $\eta = 0.1$, $\sigma_J = 0.01$, $\beta = 1$. The inset shows the plateau length vs. $M^2$, averaged over 15 initializations. Right: $\epsilon_g$ for different $L$ with $K = M = 6$, $N = 500$, $\eta = 0.01$, $\sigma_J = 10^{-6}$, $\beta = 0.25$. The inset shows the plateau length vs. $1/L$, averaged over 10 initializations. The vertical lines represent measured plateau lengths.

allows us to obtain a reduced dynamical differential equation of the form

$$\frac{d\boldsymbol{r}}{d\alpha} = \eta \boldsymbol{A}_r \boldsymbol{r}, \tag{14}$$

where $\boldsymbol{r} = [r^{(1)}, r^{(2)}, \dots, r^{(L)}]^\top$, and $\boldsymbol{A}_r \in \mathbb{R}^{L \times L}$ is defined in Appendix C.4. After solving these differential equations, we find that the escape from the plateau follows $\epsilon_g^* - \epsilon_g \propto e^{\frac{\alpha}{\tau_{\mathrm{esc}}}}$, where the escape time $\tau_{\mathrm{esc}}$ is given by

$$\tau_{\mathrm{esc}} = \frac{\pi}{2\eta} \frac{\sqrt{(M-1)T^{(1)} - D^{(1)} + M} \left( D^{(1)} + (M+1)T^{(1)} + M \right)^{\frac{3}{2}}}{\left( T^{(2)} - \frac{D^{(2)}}{M-1} \right) \left( D^{(1)} + T^{(1)} \right)}. \tag{15}$$

For large $L$, one can show that $T^{(2)} \propto L$. Therefore, for large $M$ and $L$, the escape time scales as $\tau_{\mathrm{esc}} \sim \frac{M^2}{\eta L}$. This behavior is illustrated in Figure 5, where we train a student network with synthetic input data. Additional numerical results for the plateau length are provided in Appendix C.5.

## 5.2 ASYMPTOTIC SOLUTION

In this subsection, we investigate how the generalization error converges to its asymptotic value. To this end, we consider the typical teacher configuration where $\langle T_{nm}^{(l)} \rangle = \delta_{nm} T^{(l)}$, as this configuration effectively captures the scaling behavior of the generalization error. For the asymptotic fixed points of the order parameters, we find $R_{im}^{*(l)} = T^{(l)} \delta_{im}$ and $Q_{ij}^{*(l)} = T^{(l)} \delta_{ij}$. To model the convergence towards the asymptotic solution, we again distinguish between diagonal and off-diagonal entries, parametrizing the order parameters as $R_{im}^{(l)} = R^{(l)} \delta_{im} + S^{(l)}(1 - \delta_{im})$ and $Q_{ij}^{(l)} = Q^{(l)} \delta_{ij} + C^{(l)}(1 - \delta_{ij})$, similar to the plateau case. We then linearize the dynamical equations for small perturbations around the fixed points, setting $R^{(l)} = T^{(l)} + r^{(l)}$, $S^{(l)} = T^{(l)} + s^{(l)}$, $Q^{(l)} = T^{(l)} + q^{(l)}$, and $C^{(l)} = T^{(l)} + c^{(l)}$, and retain terms up to $\mathcal{O}(\eta)$. This yields the following linearized equation

$$\frac{d\boldsymbol{x}}{d\alpha} = a \boldsymbol{A}_{\mathrm{asym}} \, \boldsymbol{x} \tag{16}$$

where $\boldsymbol{x}_i = \left( r^{(i-1)}, q^{(i-1)}, s^{(i-1)}, c^{(i-1)} \right)^\top$, $a = \frac{2\sqrt{3}}{3\pi M}$, and $\boldsymbol{A}_{\mathrm{asym}} \in \mathbb{R}^{4L \times 4L}$ is defined in Appendix C.6. After solving Eq. (16), we find for the generalization error

$$\epsilon_g = \frac{1}{6\pi} \sum_{k=1}^{L} g_k^{(1)} e^{-a(2+\sqrt{3})\lambda_k \alpha} \left( 2\sqrt{3} v_{k,2L+2}^{(1)} - 4\sqrt{3} v_{k,2}^{(1)} + 3(M-1) v_{k,3L+2}^{(1)} - 6(M-1) v_{k,L+2}^{(1)} \right)$$

$$+ g_k^{(2)} e^{-a(2-\sqrt{3})\lambda_k \alpha} \left( 2\sqrt{3} v_{k,2L+2}^{(2)} - 4\sqrt{3} v_{k,2}^{(2)} + 3(M-1) v_{k,3L+2}^{(2)} - 6(M-1) v_{k,L+2}^{(2)} \right), \tag{17}$$

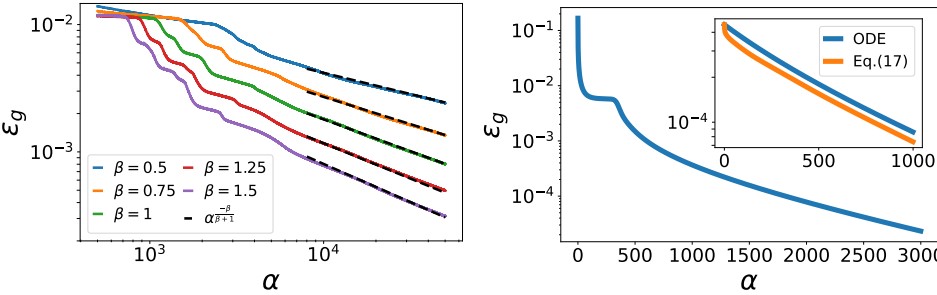

Figure 6: Scaling behavior of the generalization error $\epsilon_g$ in the asymptotic regime for a non-linear activation function. Left: $\epsilon_g$ as a function of $\alpha$ for $K = M = 40$, $\eta = 0.01$, $\sigma_J = 10^{-6}$ and $L = N = 512$ for simulations averaged over 10 different initializations. Right: $\epsilon_g$ obtained from equation Eq. (17) (orange) in comparison to the solution of differential equations of $\mathcal{O}(\eta)$ (blue) for $K = M = 2$, $\eta = 0.25$, $\beta = 1$ and $L = 9$.

where $\lambda_k$ are the eigenvalues of the data covariance matrix, $v_k^{(1)}$ and $v_k^{(2)}$ are two groups of eigenvectors corresponding to the eigenvalues $(2 + \sqrt{3}) \lambda_k$ and $(2 - \sqrt{3}) \lambda_k$, and the coefficients $g_k^{(1)}$ and $g_k^{(2)}$ depend on the initial conditions, as detailed in Appendix C.6. The weighted sum of exponentials results in a slowdown of the convergence of the generalization error, similar to the linear activation function case. Figure 6 illustrates the generalization error during the late phase of training for different $\beta$. In Appendices D and E, we investigate the scaling law through numerical analyses and simulation experiments for the over-parameterized regime $K > M$ and the ReLU activation function, as well as experiments on additional activations. In all configurations, we observe the previously derived scaling $\epsilon_g \propto \alpha^{-\frac{\beta}{1+\beta}}$, consistent with the linear activation function.

## 6 CONCLUSION

We have provided a theoretical analysis of neural scaling laws within a student-teacher framework using statistical mechanics. By deriving analytical expressions for the generalization error, we demonstrated how power-law spectra in the data covariance matrix influence learning dynamics across different regimes. For linear activation functions, we have established the conditions under which power-law scaling for the generalization error with $\alpha$ emerges and computed the power-law exponent for the scaling of the generalization error with the number of student parameters. For non-linear activations, we presented an analytical formula for the plateau length, revealing its dependence on the number of distinct eigenvalues and the covariance matrix's power-law exponent. In addition, we found that the convergence to perfect learning transitions from exponential decay to power-law scaling when the data covariance matrix exhibits a power-law spectrum. This highlights the significant impact of data correlations on learning dynamics and generalization performance.

**Limitations.** In Appendix F, we extended our analysis to include trainable second-layer weights. We found that for the error function activation, the scaling exponent did not change, whereas for the ReLU activation, there was an improvement in the scaling exponent. Numerical investigations of the differential equations reveal that with the error function activation, the weights converge to the configuration of the soft committee machine. In contrast, with the ReLU activation, the weights reach a distinct fixed point leading to improved scaling. This finding provides a foundation for future theoretical investigations into how architectural choices influence scaling laws.

Note added: After completion of this work, we became aware of the preprint Bordelon et al. (2024a), which also studies neural scaling laws in the feature learning regime.

ACKNOWLEDGMENTS

This work was supported by the IMPRS of MPI MiS Leipzig.

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

## A   DIFFERENTIAL EQUATIONS

From the stochastic gradient descent given in Eq. (3), Yoshida & Okada (2019) derived the following differential equations, generalization error, and covariance matrices in the thermodynamic limit $N \to \infty$. Following their work for our setup, one finds

$$
\frac{dQ_{ij}^{(l)}}{d\alpha} = \frac{\eta}{K} \left[ \sum_{m=1}^{M} I_3(x_i, x_j^{(l)}, y_m) - \frac{M}{K} \sum_{k=1}^{K} I_3(x_i, x_j^{(l)}, x_k) + \sum_{m=1}^{M} I_3(x_j, x_i^{(l)}, y_m) - \frac{M}{K} \sum_{k=1}^{K} I_3(x_j, x_i^{(l)}, x_k) \right]
$$

$$
+ \frac{\eta^2}{K^2} \nu_{l+1} \left[ \frac{M}{K^2} \sum_{k,l=1}^{K,K} I_4(x_i, x_j, x_k, x_l) + \sum_{n,m=1}^{M,M} I_4(x_i, x_j, y_n, y_m) - \frac{2}{K} \sum_{k,n=1}^{K,M} I_4(x_i, x_j, x_k, y_n) \right]
$$

$$
\frac{dR_{in}^{(l)}}{d\alpha} = \eta \left[ \sum_{m=1}^{M} I_3(x_i, y_n^{(l+1)}, y_m) - \sum_{j=1}^{K} I_3(x_i, y_n^{(l+1)}, x_j) \right], \tag{18}
$$

with $\nu_l = \frac{1}{N} \sum_k^N \lambda_k^l$, $I_3(z_1, z_2, z_3) = \langle g'(z_1) z_2 g(z_3) \rangle$ and $I_4(z_1, z_2, z_3, z_4) = \langle g'(z_1) g'(z_2) g(z_3) g(z_4) \rangle$. In this setting, the generalization error becomes

$$
\epsilon_g = \frac{1}{2} \left[ \sum_{m,n=1}^{M} I_2(y_n^{(1)}, y_m^{(1)}) + \sum_{i,j=1}^{K} I_2(x_j^{(1)}, x_i^{(1)}) - 2 \sum_{i=1}^{K} \sum_{n=1}^{M} I_2(x_i^{(1)}, y_n^{(1)}) \right]. \tag{19}
$$

where $I_2(z_1, z_2) := \langle g(z_1) g(z_2) \rangle$. Thereby, the $I_2, I_3$ and $I_4$ are integrals over the generalized pre-activations $x_i^{(l)}$ and $y_n^{(l)}$. Thereby, $z_i$ are normally distributed variables and stand for either $x_i^{(l)}$ or $y_n^{(l)}$. Therefore, the integrals $I_2, I_3$, and $I_4$, are multivariate Gaussian expectation values that are determined by the expectation values and covariance matrix of the generalized pre-activations. $I_2$ is a two-dimensional Gaussian integral. For example, for $I_2(x_j^{(1)}, x_j^{(1)})$, on obtains the following covariance matrix

$$
\boldsymbol{C}(i, j) = \begin{pmatrix} Q_{ii}^{(1)} & Q_{ij}^{(1)} \\ Q_{ij}^{(1)} & Q_{jj}^{(1)} \end{pmatrix}. \tag{20}
$$

The resulting elements of the covariance matrices depend on the higher-order order parameters. $I_3$ is a three-dimensional Gaussian integral, and an example of the covariance matrix is

$$
\boldsymbol{C}^{(l)}(i, j, n) = \begin{pmatrix} Q_{ii}^{(1)} & Q_{ij}^{(1)} & R_{in}^{(1)} \\ Q_{ij}^{(l+1)} & Q_{jj}^{(2l+1)} & R_{jn}^{(l+1)} \\ R_{in}^{(1)} & R_{jn}^{(l+1)} & T_{nn}^{(1)} \end{pmatrix}, \tag{21}
$$

for $I_3(x_j^{(1)}, x_j^{(l)}, y_n^{(1)})$. Note that $I_3$ depends on higher-order $l$ as compared to $I_2$ and $I_4$, which only depend on the first-order. Furthermore, $I_4$ is a four-dimensional Gaussian integral and depends, for example, on the following covariance matrix for $I_4(x_j^{(1)}, x_j^{(1)}, y_n^{(1)}, y_m^{(1)})$

$$
\boldsymbol{C}(i, j, n, m) = \begin{pmatrix} Q_{ii}^{(1)} & Q_{ij}^{(1)} & R_{in}^{(1)} & R_{im}^{(1)} \\ Q_{ij}^{(1)} & Q_{jj}^{(1)} & R_{jn}^{(1)} & R_{jm}^{(1)} \\ R_{in}^{(1)} & R_{jn}^{(1)} & T_{nn}^{(1)} & T_{nm}^{(1)} \\ R_{im}^{(1)} & R_{jm}^{(1)} & T_{nm}^{(1)} & T_{mm}^{(1)} \end{pmatrix}. \tag{22}
$$

The specific differential equations for the linear and error function activation are provided in their corresponding subsections.

## B    LINEAR ACTIVATION

### B.1    SOLUTION OF ORDER PARAMETERS

The linear activation function leads to the following teacher output $\zeta(\boldsymbol{B}, \boldsymbol{\xi}) = \frac{1}{\sqrt{M}} \sum_{n=1}^{M} \frac{\boldsymbol{\xi} \boldsymbol{B}_n}{\sqrt{N}}$. This makes it possible to rewrite the output as $\zeta(\boldsymbol{B}, \boldsymbol{\xi}) = \frac{1}{\sqrt{M}} \sum_{n=1}^{M} \frac{\boldsymbol{\xi} \boldsymbol{B}_n}{\sqrt{N}} = \frac{\boldsymbol{\xi}}{\sqrt{N}} \frac{1}{\sqrt{M}} \sum_{n=1}^{M} \boldsymbol{B}_n = \frac{\boldsymbol{\xi} \tilde{\boldsymbol{B}}}{\sqrt{N}}$, where we have defined a new teacher vector $\tilde{\boldsymbol{B}} = \frac{1}{\sqrt{M}} \sum_{n=1}^{M} \boldsymbol{B}_n$. Since $\tilde{\boldsymbol{B}}$ has the same statistical properties as one random teacher vector $\boldsymbol{B}_n$ it makes no difference whether we consider the case $M > 1$ and define $\tilde{\boldsymbol{B}}$ or $M = 1$. The same argument also applies to the linear student network. Therefore, in the following, we analyze the case $K = M = 1$.
For $K = M = 1$, the generalization error becomes

$$\epsilon_g = \frac{1}{2} \left( Q^{(1)} - 2R^{(1)} + T^{(1)} \right). \tag{23}$$

The differential equations for the order parameters are

$$\frac{dR^{(l)}}{d\alpha} = \eta \left[ T^{(l+1)} - R^{(l+1)} \right],$$
$$\frac{dQ^{(l)}}{d\alpha} = 2\eta \left[ R^{(l+1)} - Q^{(l+1)} \right] + \eta^2 T^{(l+1)} \left[ T^{(1)} + Q^{(1)} - 2R^{(1)} \right] \tag{24}$$

with $0 \le l \le L - 2$ and for the last component

$$\frac{dR^{(L-1)}}{d\alpha} = \eta \left[ \sum_{k}^{L-1} c_k \left( T^{(k)} - R^{(k)} \right) \right],$$
$$\frac{dQ^{(L-1)}}{d\alpha} = 2\eta \left[ \sum_{k}^{L-1} c_k \left( R^{(k)} - Q^{(k)} \right) \right] + \eta^2 T^{(L)} \left[ T^{(1)} + Q^{(1)} - 2R^{(1)} \right]. \tag{25}$$

Thereby, we have exploited the minimal polynomial for the order parameters for the $L$-th order, e.g. $R^{(L)} = -\sum_{k=0}^{L-1} c_k R^{(k)}$, with the coefficients of the minimal polynomial $c_k$. The set of coupled linear differential equations given by Eqs. (24) and (25) can be written in the following form

$$\frac{d}{d\alpha} \begin{pmatrix} \boldsymbol{R} \\ \boldsymbol{Q} \end{pmatrix} = \eta \begin{pmatrix} \boldsymbol{A}_1 & \boldsymbol{0}_{L \times L} \\ -2\boldsymbol{A}_1 - 2\eta \boldsymbol{U} & 2\boldsymbol{A}_1 + \eta \boldsymbol{U} \end{pmatrix} \begin{pmatrix} \boldsymbol{R} \\ \boldsymbol{Q} \end{pmatrix} + \eta \begin{pmatrix} \boldsymbol{u} \\ \eta \boldsymbol{u} \end{pmatrix}, \tag{26}$$

where $\boldsymbol{u} = \left( T^{(1)}, T^{(2)}, ..., T^{(L)} \right)^{\top}, \boldsymbol{U} = \boldsymbol{u} \boldsymbol{e}_2^{\top}$ with $\boldsymbol{e}_2 = (0, 1, 0, ..., 0)^{\top}$ and

$$\boldsymbol{A}_1 = \begin{pmatrix} 0 & -1 & 0 & \cdots & 0 & 0 \\ 0 & 0 & -1 & \ddots & \vdots & \vdots \\ 0 & 0 & 0 & \ddots & 0 & 0 \\ \vdots & \vdots & \ddots & \ddots & -1 & 0 \\ 0 & 0 & \cdots & 0 & 0 & -1 \\ c_0 & c_1 & c_2 & \cdots & c_{L-2} & c_{L-1} \end{pmatrix}, \qquad \boldsymbol{U} = \boldsymbol{u} \boldsymbol{e}_2^{\top}. \tag{27}$$

Therefore, the differential equations for the order parameters are

$$\frac{d\boldsymbol{R}}{d\alpha} = \eta \boldsymbol{A}_1 \boldsymbol{R} + \eta \boldsymbol{u} \tag{28}$$

$$\frac{d\boldsymbol{Q}}{d\alpha} = \eta \boldsymbol{A}_3 \boldsymbol{R} + \eta \boldsymbol{A}_4 \boldsymbol{Q} + \eta^2 \boldsymbol{u} \tag{29}$$

with $\boldsymbol{A}_3 = -2\boldsymbol{A}_1 - 2\eta\boldsymbol{U}$, $\boldsymbol{A}_4 = 2\boldsymbol{A}_1 + \eta\boldsymbol{U}$.

Thus, we can solve differential equations for the student-teacher order parameters independent from the student-student and find

$$\boldsymbol{R}(\alpha) = e^{\eta\boldsymbol{A}_1\alpha}\boldsymbol{R}_0 + e^{\eta\boldsymbol{A}_1\alpha}\boldsymbol{A}_1^{-1}\boldsymbol{u} - \boldsymbol{A}_1^{-1}\boldsymbol{u}, \tag{30}$$

where $\boldsymbol{R}_0$ are the student-teacher order parameters at initialization and we set $\boldsymbol{R}_0 = 0$ which is achieved on average. Before inserting this result into the differential equation for the student-student order parameters, we evaluate Eq. (30). For this, we need to find the eigenvalues of the matrix $\boldsymbol{A}_1$ and evaluate $\boldsymbol{A}_1^{-1}\boldsymbol{u}$.

First, we start with the eigenvalues. In order to find the determinant of $\boldsymbol{A}_1 - \lambda\boldsymbol{I}_L$, we apply the Laplace expansion with respect to the last row of $\boldsymbol{A}_1 - \lambda\boldsymbol{I}_{2L}$ and evaluate the determinants of $L$ different $L-1 \times L-1$ smaller matrices. The resulting sub-matrices are triangular, and their determinant is, therefore, simply given by the product of the diagonal entries. After applying the Laplace expansion, we find

$$\det(\boldsymbol{A}_1 - \lambda\boldsymbol{I}_L) = \sum_{i=0}^{L-2} c_i(-1)^{L+1+i}(-1)^{L-1-i}(-\lambda)^i + (c_{L-1} - \lambda)(-1)^{2L}(-\lambda)^{L-1}$$

$$= \sum_{i=0}^{L} c_i(-\lambda)^i = 0 \tag{31}$$

with $c_L = 1$. Since $c_0, ..., c_L$ are the coefficients of the minimal polynomial for the distinct eigenvalues of the data covariance matrix, we now know the roots of Eq. (31). Therefore, the eigenvalues of $\boldsymbol{A}_1$ are given by the negative eigenvalues of the distinct eigenvalues of the data covariance matrix $\lambda_{A_1,l} = -\lambda_l$ for $1 \le l \le L$. By applying the matrix $\boldsymbol{A}_1$ on a potential eigenvector $\boldsymbol{A}_1\boldsymbol{v}_k = \lambda_k\boldsymbol{v}_k$, we find the following conditions for the eigenvector entries

$$v_{k,i} = (-1)^{i-1}\lambda_k^{i-1}v_{k,1}, \tag{32}$$

obtained by a recursive method. Furthermore, we can choose $v_{k,1} = 1$ for all eigenvectors. The eigenvector matrix $\boldsymbol{V}$, for which an eigenvector gives each column, is given by the transpose of the Vandermonde matrix

$$\boldsymbol{V} = \begin{pmatrix} 1 & 1 & \cdots & 1 \\ \lambda_1 & \lambda_2 & \cdots & \lambda_L \\ \lambda_1^2 & \lambda_2^2 & \cdots & \lambda_L^2 \\ \vdots & \vdots & \ddots & \vdots \\ \lambda_1^{L-1} & \lambda_2^{L-1} & \cdots & \lambda_L^{L-1} \end{pmatrix}. \tag{33}$$

Second, we evaluate all matrix products given in Eq. (30). Since all eigenvalues are strictly negative, the student-teacher order parameters converge to $\lim_{\alpha\to\infty}\boldsymbol{R}(\alpha) = -\boldsymbol{A}_1^{-1}\boldsymbol{u}$ that we are going to evaluate using the eigenvector matrix. We insert the eigendecomposition $\boldsymbol{A}_1^{-1} = \boldsymbol{V}\boldsymbol{\Lambda}_2\boldsymbol{V}^{-1}$ into the asymptotic solution with $\Lambda_{2,kj} = -\delta_{kj}\frac{1}{\lambda_k}$ and find for the entries of the student-teacher order parameters

$$\lim_{\alpha\to\infty}\langle R_i\rangle_{J_{a,0},B_a} = -\sum_j^L V_{ij}\sum_k^L \Lambda_{2,jk}\sum_l^L (V^{-1})_{kl}\langle u_l\rangle_{J_{a,0},B_a}$$

$$= -\sum_j^L V_{ij}\Lambda_{2,jj}\sum_l^L (V^{-1})_{jl}\langle u_l\rangle_{J_{a,0},B_a}$$

$$= -\sum_j^L \lambda_j^{(i-1)}\frac{1}{-\lambda_j}\sum_l^L (V^{-1})_{jl}\langle u_l\rangle_{J_{a,0},B_a}$$

$$= \sum_j^L \lambda_j^{(i-2)}\sum_l^L (V^{-1})_{jl}\langle u_l\rangle_{J_{a,0},B_a}$$

$$\tag{34}$$

and further, evaluate

$$
\begin{aligned}
\sum_l \left(V^{-1}\right)_{jl} \langle u_l \rangle_{J_{a,0},B_a} &= \sum_{l=1}^L \left(V^{-1}\right)_{jl} \langle T^{(l)} \rangle_{J_{a,0},B_a} \\
&= \sum_{l=1}^L \left(V^{-1}\right)_{jl} \frac{1}{L} \sum_a^L \lambda_a^l \\
&= \frac{1}{L} \sum_a^L \lambda_a \sum_{l=1}^L \left(V^{-1}\right)_{jl} \lambda_a^{l-1} \\
&= \frac{1}{L} \sum_a^L \lambda_a \sum_{l=1}^L \left(V^{-1}\right)_{jl} V_{l,a} \\
&= \frac{1}{L} \lambda_j
\end{aligned}
\tag{35}
$$

where we have used $\sum_{l=1}^L \left(V^{-1}\right)_{jl} V_{l,a} = \delta_{j,a}$ obtained by the definition of the product between a matrix with its inverse. Thus, we find

$$
\lim_{\alpha \to \infty} \langle R_i \rangle_{J_{a,0},B_a} = \frac{1}{L} \sum_j^L \lambda_j^{(i-1)} = \langle T^{(i-1)} \rangle_{J_{a,0},B_a}.
\tag{36}
$$

Note that $R_i = R^{(i-1)}$ and therefore $\lim_{\alpha \to \infty} \langle R^{(i)} \rangle_{J_{a,0},B_a} = \langle T^{(i)} \rangle_{J_{a,0},B_a}$.
Next, we want to evaluate $e^{\boldsymbol{A}_1 \alpha} \boldsymbol{A}_1^{-1} \boldsymbol{u}$ and define

$$
\begin{aligned}
F_i &= \sum_j^L V_{ij} \sum_k^L \exp\left(-\eta \lambda_j \alpha\right) \delta_{jk} \sum_l^L \Lambda_{kl} \sum_m^L \left(V^{-1}\right)_{lm} \langle u_m \rangle_{J_{a,0},B_a} \\
&= \sum_j^L V_{ij} \exp\left(-\eta \lambda_j \alpha\right) \frac{1}{-\lambda_j} \sum_m^L \left(V^{-1}\right)_{lm} \langle u_m \rangle_{J_{a,0},B_a} \\
&= -\frac{1}{L} \sum_j^L \lambda_j^{i-2} \exp\left(-\eta \lambda_j \alpha\right) \lambda_j \\
&= -\frac{1}{L} \sum_j^L \exp\left(-\eta \lambda_j \alpha\right) \lambda_j^{i-1}
\end{aligned}
\tag{37}
$$

which leads to $\langle \boldsymbol{R}\left(\alpha\right) \rangle_{J_{a,0},B_a} = \boldsymbol{F} - \boldsymbol{A}_1^{-1} \langle \boldsymbol{u} \rangle = \boldsymbol{T} + \boldsymbol{F}$. Thus, we obtain for the expectation value of $R^{(1)}$

$$
\begin{aligned}
\langle R^{(1)} \rangle_{J_{a,0},B_a} &= \langle R_2 \rangle_{J_{a,0},B_a} \\
&= 1 - \frac{1}{L} \sum_a^L \exp\left(-\eta \lambda_a \alpha\right) \lambda_a.
\end{aligned}
\tag{38}
$$

As a next step, we insert the result given by Eq. (30) for $\langle \boldsymbol{R}_0 \rangle = 0$ into the differential equations for the student-student order parameters given by Eq. (29), in order to obtain a new expression

$$
\frac{d\boldsymbol{Q}}{d\alpha} = \eta \boldsymbol{A}_3 \left(e^{\eta \boldsymbol{A}_1 \alpha} \boldsymbol{A}_1^{-1} \boldsymbol{u} - \boldsymbol{A}_1^{-1} \boldsymbol{u}\right) + \eta \boldsymbol{A}_4 \boldsymbol{Q} + \eta^2 \boldsymbol{u}
\tag{39}
$$

In order to simplify the differential equation, we evaluate $\boldsymbol{A}_3 \boldsymbol{A}_1^{-1} \boldsymbol{T}$. The inverse of $\boldsymbol{A}_1$ can be obtained analytically, where we find

$$\boldsymbol{A}_1^{-1} = \begin{pmatrix} \frac{c_1}{c_0} & \frac{c_2}{c_0} & \frac{c_3}{c_0} & \cdots & \frac{c_{L-1}}{c_0} & \frac{1}{c_0} \\ -1 & 0 & 0 & \ddots & \vdots & \vdots \\ 0 & -1 & 0 & \ddots & 0 & 0 \\ \vdots & \vdots & \ddots & \ddots & 0 & 0 \\ 0 & 0 & \cdots & -1 & 0 & 0 \\ 0 & 0 & 0 & \cdots & -1 & 0 \end{pmatrix} \tag{40}$$

and obtain for the matrix-vector product $-\boldsymbol{A}_3\boldsymbol{A}_1^{-1}\boldsymbol{u} = 2\left(\eta - 1\right)\boldsymbol{u}$. We can further simplify $\eta\boldsymbol{u} - \boldsymbol{A}_3\boldsymbol{A}_1^{-1}\boldsymbol{u} = (\eta - 2)\,\boldsymbol{u}$ and insert this result in Eq. (39) in order to obtain

$$\frac{d\boldsymbol{Q}}{d\alpha} = \eta\boldsymbol{A}_3 e^{\boldsymbol{A}_1\alpha}\boldsymbol{A}_1^{-1}\boldsymbol{u} + \eta(2-\eta)\boldsymbol{u} + \eta\boldsymbol{A}_4\boldsymbol{Q}. \tag{41}$$

The solution of this differential equation is given by

$$\boldsymbol{Q}\left(\alpha\right) = e^{\eta\boldsymbol{A}_4\alpha}\boldsymbol{Q}_0 + e^{\eta\boldsymbol{A}_4\alpha}(\boldsymbol{A}_1 - \boldsymbol{A}_4)^{-1}\left(e^{(\eta\boldsymbol{A}_1 - \eta\boldsymbol{A}_4)\alpha} - \boldsymbol{I}\right)\boldsymbol{A}_3\boldsymbol{A}_1^{-1}\boldsymbol{u} \tag{42}$$

$$+ (2-\eta)e^{\eta\boldsymbol{A}_4\alpha}\boldsymbol{A}_4^{-1}\left(\boldsymbol{I} - e^{-\eta\boldsymbol{A}_4\alpha}\right)\boldsymbol{u} \tag{43}$$

with $\boldsymbol{Q}_0$ are the student-student order parameters at initialization. Note that the initial value $\boldsymbol{Q}_0$ does not vanish in the thermodynamic limit and is also not zero on average in contrast to $\boldsymbol{R}_0$. By the definition of the matrices, we find $\boldsymbol{A}_1 - \boldsymbol{A}_4 = \frac{1}{2}\boldsymbol{A}_3$ making it straightforward to evaluate their relations $(\boldsymbol{A}_1 - \boldsymbol{A}_4)^{-1}\boldsymbol{A}_3 = 2\boldsymbol{I}_L$. Furthermore, in order to estimate $(2-\eta)\boldsymbol{A}_4^{-1}\boldsymbol{u}$, we need to know the inverse of $\boldsymbol{A}_4$ for which we find

$$\boldsymbol{A}_4^{-1} = \begin{pmatrix} \frac{c_1}{2c_0} - \frac{\eta}{2-\eta} & \frac{c_2}{2c_0} & \frac{c_3}{2c_0} & \cdots & \frac{c_{L-1}}{2c_0} & \frac{1}{2c_0} \\ \frac{1}{\eta-2} & 0 & 0 & \ddots & \vdots & \vdots \\ \frac{\eta T^{(2)}}{2(\eta-2)} & -\frac{1}{2} & 0 & \ddots & 0 & 0 \\ \vdots & \vdots & \ddots & \ddots & 0 & 0 \\ \frac{\eta T^{(L-2)}}{2(\eta-2)} & 0 & \cdots & -\frac{1}{2} & 0 & 0 \\ \frac{\eta T^{(L-1)}}{2(\eta-2)} & 0 & 0 & \cdots & -\frac{1}{2} & 0 \end{pmatrix}. \tag{44}$$

Thus, we obtain for the product $(2-\eta)\boldsymbol{A}_4^{-1}\boldsymbol{u} = -\boldsymbol{T}$. Therefore, we can simplify Eq. (43) to

$$\boldsymbol{Q}\left(\alpha\right) = \boldsymbol{T} + e^{\eta\boldsymbol{A}_4\alpha}\boldsymbol{Q}_0 + \left(e^{\eta\boldsymbol{A}_4\alpha} - 2e^{\eta\boldsymbol{A}_1\alpha}\right)\boldsymbol{T} \tag{45}$$

Using Eq. (23) and taking the expectation value over initial student and teacher entries $B_a$ and $J_a^0$, we obtain for the generalization error

$$\langle\epsilon_g\rangle_{J_a^0, B_a} = \frac{\left(1 + \sigma_J^2\right)}{2L}\sum_{k=1}^{L} b_k\tilde{\lambda}_k\exp(-2\eta\tilde{\lambda}_k\alpha)\,, \tag{46}$$

where $\tilde{\lambda}_k$ are the eigenvalues of $\boldsymbol{A}_1 + \eta\boldsymbol{U}$, $b_k = \sum_{l=1}^{L}\left(\boldsymbol{W}^{-1}\right)_{kl}T^{(l)}$, and $\boldsymbol{W}$ contains the eigenvectors of $\boldsymbol{A}_1 + \eta\boldsymbol{U}$. Although $\boldsymbol{U}$ is a rank-1 matrix, standard perturbation methods are not applicable to find the eigenvalues of the shifted matrix $\boldsymbol{A}_1 + \eta\boldsymbol{U}$ because $\boldsymbol{U}$ may have a large eigenvalue, making it unsuitable as a small perturbation.

### B.1.1 SMALL LEARNING RATES

For small learning rates, we can approximate $\boldsymbol{A}_4 \approx 2\boldsymbol{A}_1$ and immediately find for the first order student-student order parameters

$$\langle Q^{(1)} \rangle_{J_{a,0},B_a} = \langle Q_2 \rangle_{J_{a,0},B_a}$$

$$= 1 + \left(1 + \sigma_J^2\right) \frac{1}{L} \sum_a^L \exp\left(-2\eta\lambda_a\alpha\right) \lambda_a - 2\frac{1}{L} \sum_a^L \exp\left(-\eta\lambda_a\alpha\right) \lambda_a, \quad (47)$$

where we have exploit that $\langle \boldsymbol{Q}_0 \rangle_{J_{a,0},B_a} = \sigma_J^2 \langle \boldsymbol{T} \rangle_{J_{a,0},B_a}$. After inserting Eqs. (38) and (47) into Eq. (23), we obtain for the generalization error

$$\langle \epsilon_g \rangle_{J_{a,0},B_a} = \left(1 + \sigma_J^2\right) \frac{1}{2L} \sum_{a=1}^L \lambda_a \exp(-2\eta\lambda_a\alpha), \quad (48)$$

leading to Eq. (8) in the main text.

### B.2 SCALING WITH TIME

In order to evaluate the sum for the generalization error given in Eq. (48), we consider the Euler-Maclaurin approximation. The formula for the approximation is given by (cf. Apostol (1999))

$$\sum_{i=m}^n f(i) = \int_m^n f(x)\,dx + \frac{f(n) + f(m)}{2} + \sum_{k=1}^{\lfloor \frac{p}{2} \rfloor} \frac{B_{2k}}{(2k)!} \left( f^{(2k-1)}(n) - f^{(2k-1)}(m) \right), \quad (49)$$

where $B_{2k}$ is the $k$th Bernoulli number. For the difference of the sum from the integral, one can find

$$\begin{aligned}
\sum_{i=m}^n f(i) - \int_m^n f(x)\,dx &= \frac{f(m) + f(n)}{2} + \int_m^n f'(x)P_1(x)\,dx \\
&= \frac{f(m) + f(n)}{2} + \frac{1}{6}\frac{f'(n) - f'(m)}{2!} + \int_m^n f'''(x)\frac{P_3(x)}{3!}\,dx,
\end{aligned} \quad (50)$$

where $P_k(x)$ are the periodized Bernoulli polynomials of order $k$. These are defined as:

$$P_k(x) = B_k(x - \lfloor x \rfloor),$$

where $B_k(x)$ are the Bernoulli polynomials and $\lfloor x \rfloor$ denotes the floor function. In our case, the function and its derivative are

$$f(k, \alpha, \beta) = \frac{\left(1 + \sigma_J^2\right)}{L} \frac{\lambda_+}{k^{\beta+1}} \exp\left(-\frac{2\eta\lambda_+\alpha}{k^{\beta+1}}\right)$$

$$f'(k) = \frac{\left(1 + \sigma_J^2\right)\lambda_+}{L} \exp\left(-\frac{2\eta\lambda_+\alpha}{k^{\beta+1}}\right) \left(\frac{2\eta\lambda_+\alpha(\beta+1)}{k^{2\beta+3}} - \frac{\beta+1}{k^{\beta+2}}\right).$$

We want to approximate the sum by an integral and, therefore, estimate their maximal difference. Due to the exponential pre-factor in all functions, the maximal deviation of the integral from the sum is obtained at initialization $\alpha = 0$ or for $\eta \to 0$. Therefore, we make the following ansatz for $\lim_{\eta \to 0}$

$$\sum_{i=m}^n f(i) - \int_m^n f(x)\,dx < \frac{f(m) + f(n)}{2} + \int_m^n f'(x)P_1(x)\,dx. \quad (51)$$

Our goal is to express the term $\int_m^n f'(x)P_1(x)\,dx$ for $\eta \to 0$. To do this, we consider the integral over the interval $[k, k+1)$, where $k$ is an integer. This allows us to handle the floor function more easily, as it is constant over each interval. Within each interval, one can simplify the expression for $\int_k^{k+1} f'(x)P_1(x)\,dx$, which can be written as:

$$\Delta_k = \frac{L}{(1 + \sigma_J^2)\lambda_+} \lim_{\eta \to 0} \int_k^{k+1} f'(x)P_1(x)\,dx.$$

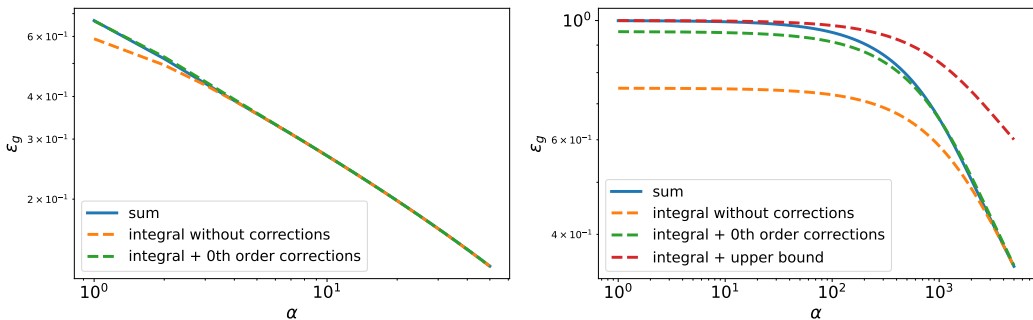

Figure 7: Generalization error evaluated by Eq. (48) (full blue) and Eq. (49) including different order of function derivatives (dashed) for $N = 128$, $K = M = 1$, $\sigma_J^2 = 1$ and $\beta = 0.5$. The integral without corrections (orange) is given by Eq. (54). Left: For $\eta = \frac{1}{2\lambda_+} \approx 0.001$, the difference between the sum and its integral representation is well represented by Eq. (50) including corrections up to 0th order in the derivative of $f$. Right: For $\eta = 10^{-6}$ and $\alpha < \frac{1}{2\eta\lambda_+} \approx 10^3$, the difference between the sum and its integral representation is quantified by the upper bound given in Eq. (52) (red dashed). However, for $\alpha > \frac{1}{2\eta\lambda_+}$, including derivatives up to 0th order in Eq. (50) are sufficient to quantify the deviation of the sum from its integral representation.

Substituting the expression for $f'(x)$ and taking the limit as $\eta \to 0$, $\Delta_k$ becomes:

$$\Delta_k = \frac{1}{2(k+1)^{\beta+1}} + \frac{1}{2k^{\beta+1}} + \frac{1}{\beta(k+1)^\beta} - \frac{1}{\beta k^\beta}.$$

To estimate the total correction across the entire interval from 1 to $L$, we sum this expression from $k = 1$ to $L - 1$ for large $L$. For the first term, we obtain

$$\frac{1}{2}\sum_{k=1}^{L-1} \frac{1}{(k+1)^{\beta+1}} = \frac{1}{2}\zeta(\beta+1) - \frac{1}{2},$$

where $\zeta(\beta + 1)$ is the Riemann zeta function. The second term is straightforward and sums to:

$$\frac{1}{2}\sum_{k=1}^{L-1} \frac{1}{k^{\beta+1}} = \frac{1}{2}\zeta(\beta+1).$$

For the difference between the third and fourth terms, we find

$$\sum_{k=1}^{L-1} \frac{1}{\beta(k+1)^\beta} - \frac{1}{\beta k^\beta} = -\frac{1}{\beta}$$

Combining all these results, the sum of the integral correction term over all intervals from 1 to $L - 1$ is:

$$\sum_{k=1}^{L-1} \Delta_k = \zeta(\beta+1) - \frac{1}{2} - \frac{1}{\beta}.$$

After inserting the start and end point of the sum into the function $f(m = 1)$ and $f(n = L)$ for $\eta \to 0$, we obtain an upper bound for the error

$$\sum_{i=m}^{n} f(i) - \int_m^n f(x)\, dx < \frac{\left(1 + \sigma_J^2\right)\lambda_+}{2L}\left[\frac{1}{L^{\beta+1}} + \zeta(\beta+1) - \frac{1}{\beta}\right]. \tag{52}$$

This "worst case" upper bound works excellent for moderate input sizes $N \sim \mathcal{O}\left(10^1\right)$ as well. The difference between the sum and its integral representation beyond the small learning rate limit can be estimated as well. If we consider a learning rate larger than $\eta > \frac{1}{2\lambda_+\alpha}$, then the difference between the sum and its integral representation is well quantified by including derivatives of $f$ up to

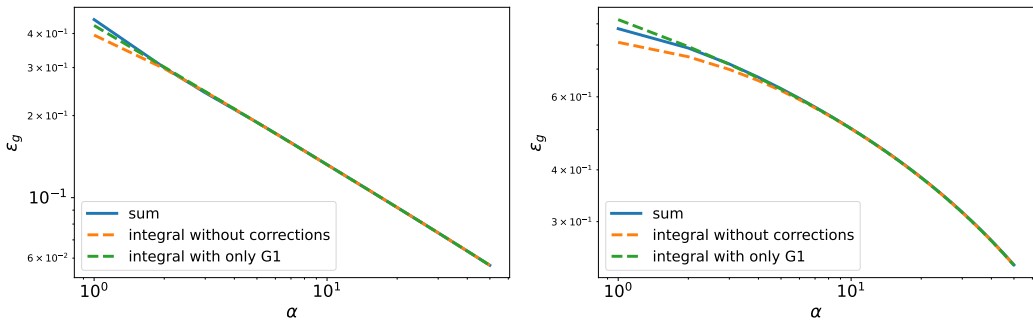

Figure 8: Generalization error evaluated by Eq. (48) (full blue), Eq. (54) (dashed orange) and (55) (dashed green) for $N = 128$, $K = M = 1$, $\sigma_J^2 = 1$ and $\eta = 0.01$. Left: For $\beta = 1$. Right: For $\beta = 0.01$.

0th order on the right-hand side of Eq. (50). Especially $f(m = 1)$ plays the most important role, and we find

$$\sum_{i=m}^{n} f(i) - \int_{m}^{n} f(x)\,dx < \frac{1}{2}f(1) = \frac{\left(1 + \sigma_J^2\right)\lambda_+}{2L} \exp\left(-2\eta\lambda_+\alpha\right). \tag{53}$$

Since we consider $\eta > \frac{1}{2\lambda_+\alpha}$, the difference decreases exponentially in time. For smaller learning rates $\eta < \frac{1}{2\lambda_+\alpha}$ higher order must be included up to the "worst case" bound given in Eq. (52). However, for the condition $\eta < \frac{1}{2\lambda_+\alpha}$ or equivalently $\alpha < \frac{1}{2\eta\lambda_+}$ no meaningful learning has occurred.

In the right panel of Figure 7, we have chosen $\eta = 10^{-6} < \frac{1}{2\lambda_+} \approx 10^{-2}$ in order to test our approximation of the sum. We find an excellent agreement. However, such a learning rate is very untypical since no learning would occur over many time orders. Typically, the learning rates that we use in our simulations are larger of order $\mathcal{O}\left(10^{-3}\right)$ to $\mathcal{O}\left(1\right)$. Therefore, including 0th order derivatives of $f$ in order to estimate the difference between the sum and its integral representation is enough to consider.

Since the corrections, including 0th order derivatives of $f$, decrease exponentially with $\alpha$, we use the integral in order to approximate the generalization error $\epsilon_g(L; \alpha) \approx \int_m^n f(x)\,dx$ for $m = 1$ to $n = L$. This integral can be solved analytically, and we obtain

$$\epsilon_g(L; \alpha) \approx \frac{\left(1 + \sigma_J^2\right)\lambda_+}{2L} \frac{(2\eta\lambda_+\alpha)^{-\frac{\beta}{1+\beta}}}{1 + \beta} \left[\Gamma\left(\frac{\beta}{1+\beta}, \frac{2\eta\lambda_+\alpha}{L^{\beta+1}}\right) - \Gamma\left(\frac{\beta}{1+\beta}, 2\eta\lambda_+\alpha\right)\right] \tag{54}$$

where $\Gamma\left(s, x\right)$ is the incomplete gamma function.

Next, we aim to clarify under which conditions the generalization error scales as a power-law in $\alpha$. For the second gamma function within the brackets in Eq. (54), we notice that its argument is $L^{1+\beta}$ larger than for the first gamma function. Thus, we can introduce a scaled time variable by $\tilde{\alpha} = 2\eta\lambda_+\alpha$ and insert this into the second gamma function $\Gamma\left(\frac{\beta}{1+\beta}, 2\eta\lambda_+\alpha\right) = \Gamma\left(\frac{\beta}{1+\beta}, \tilde{\alpha}\right)$ decreasing exponentially fast with $\tilde{\alpha}$. Therefore, we can neglect the second gamma function compared to the first one since both operate on different time scales, as presented in Figure 8. Thus, we further simplify

$$\epsilon_g(L; \alpha) \approx \frac{\left(1 + \sigma_J^2\right)\lambda_+}{2L} \frac{(2\eta\lambda_+\alpha)^{-\frac{\beta}{1+\beta}}}{1 + \beta} \Gamma\left(\frac{\beta}{1+\beta}, \frac{2\eta\lambda_+\alpha}{L^{\beta+1}}\right) \tag{55}$$

valid for $\frac{1}{2\eta\lambda_+} < \alpha$. Furthermore, from the previous discussion based on empirical observations, we know that no meaningful learning happens for $\alpha < \frac{1}{2\eta\lambda_+}$. The incomplete gamma function given in Eq. (55) can further be approximated by

$$\Gamma(s, z) \approx \Gamma(s) - \frac{z^s}{s} \tag{56}$$

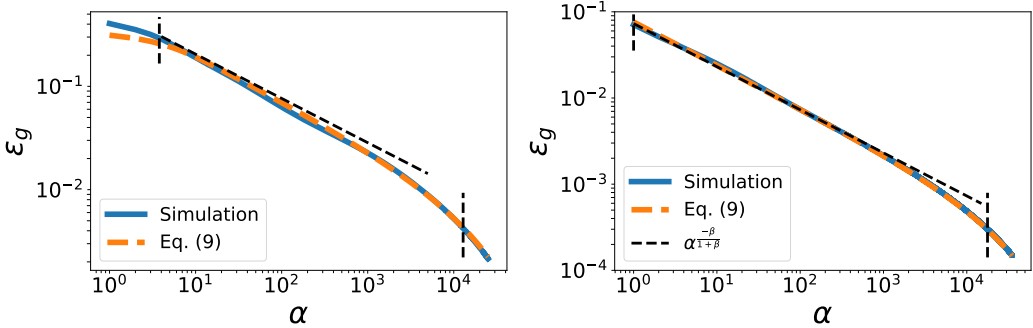

Figure 9: Generalization error as a function of $\alpha$ obtained by simulation experiments for the linear activation function for $K = M = 1$. We test our time window given in Eq. (58) (vertical lines) and power-law exponent $\epsilon_g \propto \alpha^{\frac{-\beta}{1+\beta}}$ (black dashed descending line) for and $\sigma_J = 0.01$. Left: A setup configuration for $N = L = 256$, $\beta = 0.75$ and $\eta = 0.001$, where we obtain $3.8 < \alpha < 12850$. Right: A setup configuration for $N = L = 1024$, $\beta = 1$ and $\eta = 0.01$, where we obtain $0.2 < \alpha < 17740$. Note that we have placed the lower bound at $\alpha = 1$ in the right panel.

for $z \ll 1$, $s = \frac{\beta}{\beta+1}$ and $z = \frac{2\eta\lambda_+\alpha}{L^{1+\beta}}$. As a last approximation, we use $\Gamma(s, z) \approx \Gamma(s)$ valid for $\Gamma(s) \gg \frac{z^s}{s}$, in order to find for the generalization error

$$\epsilon_g(L; \alpha) \approx \frac{\left(1 + \sigma_J^2\right)\lambda_+}{2L} \frac{(2\eta\lambda_+\alpha)^{-\frac{\beta}{1+\beta}}}{1+\beta} \Gamma\left(\frac{\beta}{1+\beta}\right). \tag{57}$$

From Eq. (57), we obtain a power-law scaling for the generalization error $\epsilon_g \propto \alpha^{\frac{-\beta}{1+\beta}}$.
For which range of $\alpha$ can this scaling law be observed? Clearly, the lower bound for $\alpha$ is $\frac{1}{2\eta\lambda_+} < \alpha$ as already discussed. The upper bound can be estimated from the approximation $\Gamma(s, z) \approx \Gamma(s)$, which is valid for the condition $\frac{z^s}{s} \ll \Gamma(s)$. In order to quantify the notation "much less than", we demand that $\frac{z^s}{s}$ is at least one order of magnitude smaller than $\Gamma(s)$, i.e. $\frac{z^s}{s} < \frac{1}{10}\Gamma(s)$. Using this inequality together with the definition of $z$ and $s$, we obtain the following time condition $\alpha < \frac{L^{1+\beta}}{2\eta\lambda_+}\left(\frac{\beta}{10(1+\beta)}\Gamma\left(\frac{\beta}{1+\beta}\right)\right)^{\frac{1+\beta}{\beta}}$. Thus, we estimate the following time window for which we expect a power-law of the $\epsilon_g$ in $\alpha$

$$\frac{1}{2\eta\lambda_+} < \alpha < \frac{L^{1+\beta}}{2\eta\lambda_+}\left(\frac{\beta}{10\left(1+\beta\right)}\Gamma\left(\frac{\beta}{1+\beta}\right)\right)^{\frac{1+\beta}{\beta}} \tag{58}$$

We observe that the power-law range is extended with an increasing number of distinct eigenvalues $L$ and covariance matrix power-law exponent $\beta$. In Figure 9, we test our upper and lower bound for simulation experiments and find very good agreement.

### B.2.1 GENERAL LEARNING RATE

Here, we reconsider Eq. (45) and want to evaluate the student-student order parameters not in the small learning rate limit. For this, we have to understand how the eigenvectors of $A_4$ influence $T$. We call the eigenvector matrix of $A_4$ simply $V_4$ which contains all eigenvectors as its columns and introduce the eigendecomposition $e^{A_4\alpha} = V_4 e^{\eta\Lambda_4\alpha}V_4^{-1}$ with $\Lambda_4$ containing the eigenvalues of $A_4$ on its diagonal. To find the eigenvalues of $A_4$ either analytically or numerically is no longer easy. However, we can pretend to know the eigenvalues of $A_4$ and call them $\lambda_{4,k}$ for $1 \leq k \leq L$. This makes it possible to find the structure of the eigenvector matrix.
Here, we present a more general solution for the matrix $B = aA_1 + \epsilon U$ and call the eigenvalues of $B$ simply $\lambda_{B,k}$ and define the eigenvectors $Bv_{B,k} = \lambda_{B,k}v_{B,k}$ for $k \in \{1, ..., L\}$. Note that for $a = 2$ and $\epsilon = \eta$, we can reproduce $B(a = 2, \epsilon = \eta) = A_4$. For each $\lambda_{B,k}$, the corresponding $k$-th eigenvector has the following structure

$$v_{B,1k} = 1, \quad v_{B,2k} = \frac{\lambda_{B,k}}{\epsilon - a}, \quad v_{B,3k} = \frac{1}{a}\frac{1}{\epsilon - a}(\epsilon T^2 - \lambda_{B,k}),$$

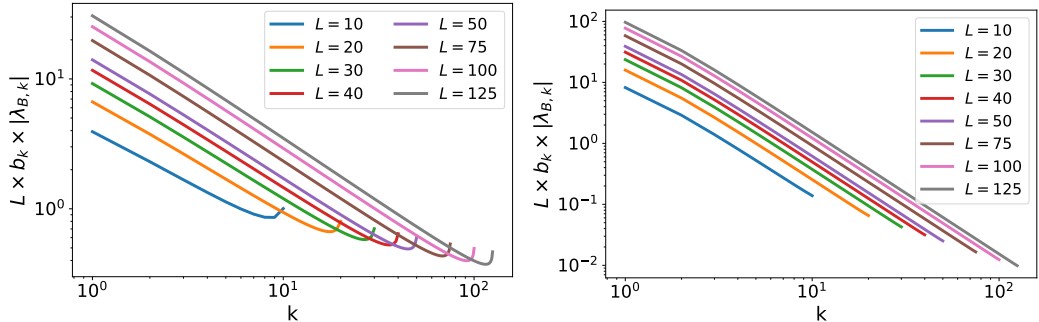

Figure 10: Coefficients $b_k$ defined by Eq. (62) multiplied with $\lambda_{B,k}$ for $\eta = 1.0$. Left: $\beta = 0.1$. Right: $\beta = 1.0$.

$$v_{B,ik} = -\frac{1}{a^{L-2}(\epsilon - a)}\left(-\lambda_{B,k}^{L-1} + \epsilon\sum_{i=1}^{L-2}(-1)^i a^{L-2-i}\lambda_{B,k}^i T^{L-i}\right).$$

The eigenvectors have an interesting property. Like for the companion matrix $A_1$ and its Vandermonde eigenvector matrix $V_1$, the first $l-1$ entries of the eigenvalue equation $Bv_{B,k} = \lambda_{B,k}v_{B,k}$ give a zero by construction $(Bv_{B,k} - \lambda_{B,k}v_{B,k})_l = 0$ for $l = 1, ..., L-1$. Only the last entry $l = L$ of the eigenvalue equation provides information about the eigenvalues of $B$

$$(\boldsymbol{B}\boldsymbol{v}_{B,k} - \lambda_{B,k}\boldsymbol{v}_{B,k})_L = \sum_{j=1}^{L} c_{B,j}\lambda_{B,k}^j.$$

This is exactly the minimal polynomial for $B$ where we have identified

$$c_{B,0} = ac_0, \quad c_{B,L} = \frac{(-1)^{L+1}}{a^{L-2}}\frac{1}{\epsilon - a} \tag{59}$$

$$c_{B,j} = \frac{1}{a^{j-1}}\frac{(-1)^{j+1}}{\epsilon - a}\left(ac_j + \epsilon\sum_{l=0}^{L-1-i} c_{L-l}T^{(L+1-j-l)}\right) \tag{60}$$

as the shifted coefficients of $B$ compared the coefficients $c_j$ of $A_1$ for $j = 1, ..., L-1$. Since we have assumed that $\lambda_{B,k}$ is an eigenvalue of $B$, the right-hand side of Eq. has to be zero, and the $c_{B,k}$ are indeed the new coefficients. Since we know the shifted coefficients, we can numerically calculate the corresponding eigenvalues as the roots of the new minimal polynomial.

Most important is the second entry of the eigenvector matrix $V_4$ since this will have an influence on the generalization error. Next, we evaluate $V_4^{-1}T$

$$\langle\epsilon_g\rangle_{J_{a,0},B_a} = (1+\sigma_J^2)\frac{1}{2L}\sum_{k=1}^{L}b_k\lambda_{B,k}\exp(-2\eta\lambda_{B,k}\alpha), \tag{61}$$

with $b_k = \sum_l^L (V_B^{-1})_{kl} T^{(l)}$ and $\lambda_{B,k}$ are the eigenvalues of $B$ for $a = 2$ and $\epsilon = \eta$. Thus, in order to calculate the generalization error, we have to find an expression for $V_B^{-1}$. We know that $B$ shows some properties of a companion matrix. Therefore, we perform a similarity transformation $B = SB_2S^{-1}$. Thereby, $B_2$ has again a companion matrix structure similar to $A_1$, but it has $c_{B,i}$ for its last row entry for $i = 0, ..., L-1$ given by Eq. (60). For the transformation matrix $S$, we obtain

$$S = \frac{1}{(a-\eta)}\begin{bmatrix} a-\eta & 0 & 0 & 0 & \cdots & 0 \\ 0 & -1 & 0 & 0 & \cdots & 0 \\ 0 & -T^2\frac{\eta}{a} & \frac{1}{a} & 0 & \cdots & 0 \\ 0 & -T^3\frac{\eta}{a} & T^2\frac{\eta}{a^2} & -\frac{1}{a^2} & \ddots & 0 \\ \vdots & \vdots & \vdots & \vdots & \ddots & \vdots \\ 0 & -T^{L-1}\frac{\eta}{a} & T^{L-2}\frac{\eta}{a^2} & -T^{L-3}\frac{\eta}{a^3} & \cdots & (-1)^{L-1}\frac{1}{a^{L-1}} \end{bmatrix}$$

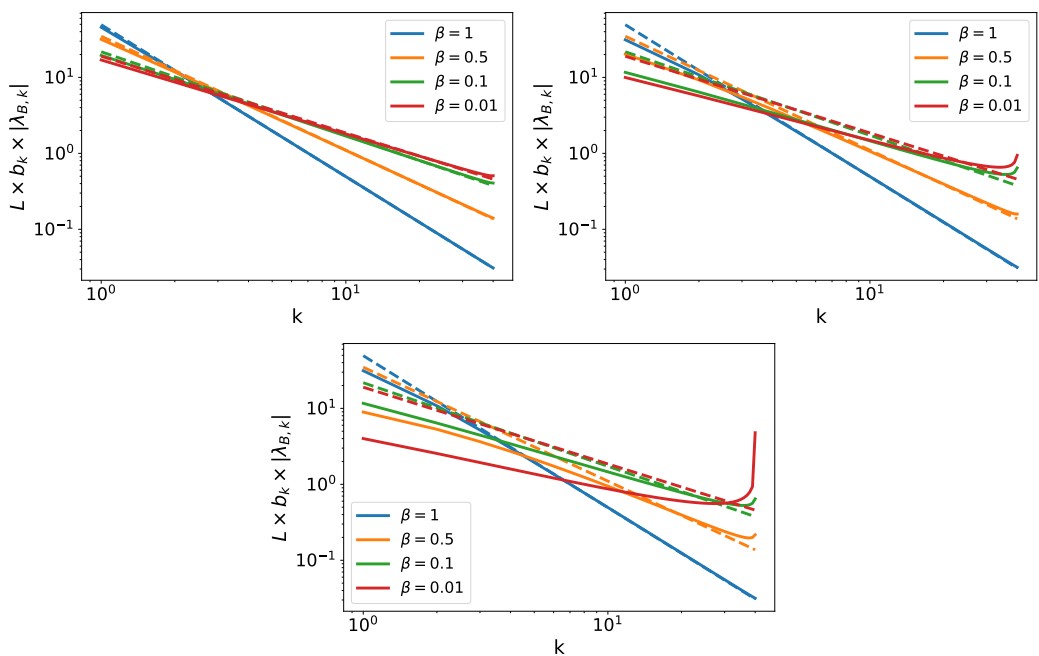

Figure 11: Coefficients $b_k$ defined by Eq. (62) multiplied with $\lambda_{B,k}$ and $L$ for $L = 40$ (solid) and eigenvalues of the data covariance matrix (dashed). Upper left: $\eta = 0.1$. Upper right: $\eta = 0.5$. Center: $\eta = 1.0$.

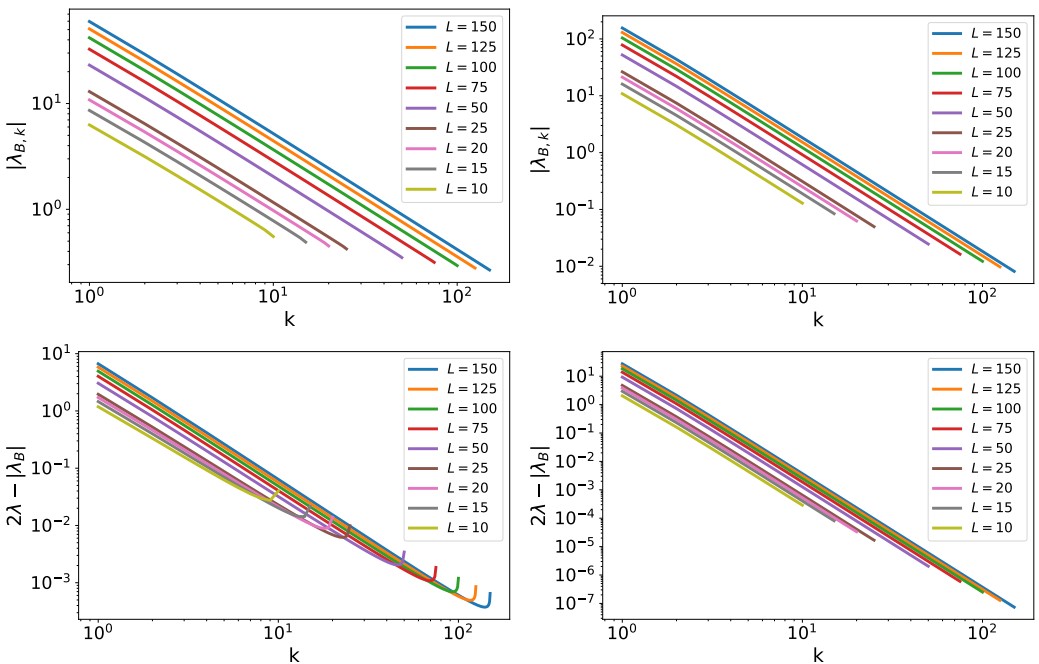

Figure 12: Spectrum of matrix $\boldsymbol{B}$ for $a = 2$ (upper figures) and corresponding difference with the spectra of the data covariance matrix. Left: $\epsilon = 1$ and $\beta = 0.1$. Right: $\epsilon = 0.5$ and $\beta = 1.0$.

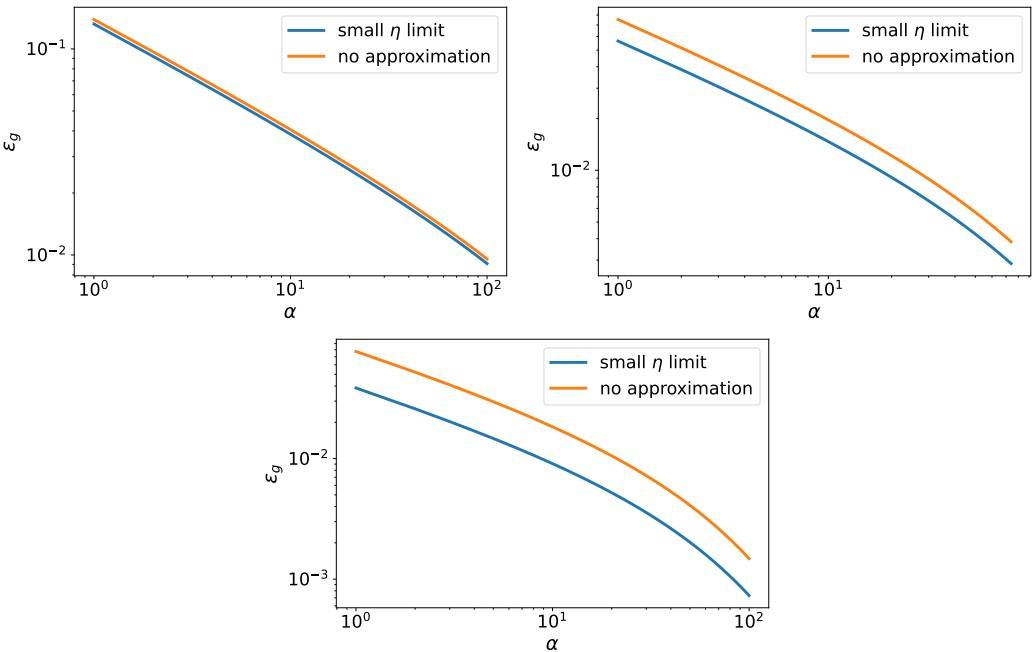

Figure 13: Generalization error evaluated by Eq. (8) (blue) and Eq. (6) (orange) for $N = 128$ , $K = M = 1$, $\sigma_J^2 = 0$ and $\beta = 1$. Upper left: $\eta = 0.1$. Upper Right: $\eta = 0.5$. Bottom: $\eta = 1.0$.

having a triangular structure. The entries of the inverse of a triangular matrix can be calculated by this simple relation

$$S_{ii}^{-1} = \frac{1}{S_{ii}}$$

$$S_{ij}^{-1} = -\frac{1}{S_{ii}} \sum_{k=i}^{j-1} S_{ik} S_{kj}^{-1}, \quad \text{for } i > j$$

Note that the eigenvector matrix $\boldsymbol{V}_{B2}$ is again a Vandermonde matrix, meaning that we know the entries and their inverse. Therefore, we obtain

$$b_k = \sum_l^L \left( (\boldsymbol{S}\boldsymbol{V}_{B2})^{-1} \right)_{kl} T^{(l)}. \tag{62}$$

In practice, we no longer have to directly invert any matrix, which allows us to consider higher values of $L$ and more different covariance matrix power-law exponents $\beta$. Calculating the inverse of $\boldsymbol{V}_B$ is numerically very unstable and just possible for small numbers of distinct eigenvalues $L$.

Figure 10 compares $b_k$ multiplied with $\lambda_{B,k}$ with the eigenvalues of the data covariance matrix $\lambda_k$ for various $L$. For small learning rates, the differences in the spectra are very small and increase with increasing learning rates until the trend of power-law decay is destroyed. Note that $b_k = \frac{1}{L}$ and $\lambda_{B,k} = -a\lambda_k$ if we just consider the differential equations up to order $\mathcal{O}(\eta)$. Figure 11 shows the same behavior but for different $\beta$. The smaller $\beta$ becomes, the larger the divination from a consistent power-law scaling. Figure 12 shows the spectrum of $\lambda_{B,k}$ and its difference from $\lambda_k$. We also observe in the spectra that the pure power-law behavior is perturbed. However, this effect seems to just increase the generalization error without changing its scaling, as shown in Figure 13.

## B.3 FEATURE SCALING

Instead of considering the statistical mechanics approach, we study directly the stochastic gradient descent here

$$\boldsymbol{J}^{\mu+1} - \boldsymbol{J}^\mu = -\eta \nabla_{\boldsymbol{J}} \epsilon \left( \boldsymbol{J}^\mu, \boldsymbol{\xi}^\mu \right). \tag{63}$$

From this difference equation, one can derive a Langevin equation for the weight dynamics in the thermodynamic limit $N \to \infty$ (see Rotskoff & Vanden-Eijnden (2022); Worschech & Rosenow (2024)). For the continuous equation describing the trajectories of the weights, we obtain

$$\frac{\mathrm{d}\boldsymbol{J}}{\mathrm{d}\alpha} = -\eta \nabla_{\boldsymbol{J}} \epsilon_g + \frac{\eta}{\sqrt{N}} \boldsymbol{\gamma} \tag{64}$$

where $\boldsymbol{\gamma}$ is a random vector describing the path fluctuations with $\langle \boldsymbol{\gamma} \rangle = 0$ and $\langle \gamma_i(\alpha) \gamma_j(\alpha') \rangle = C_{ij}\delta(\alpha - \alpha')$.

For small learning rates or a proper scaling of the learning rate with the network size, we can neglect path fluctuations and approximate the stochastic process by its mean path. This leads to the following differential equation

$$\frac{\mathrm{d}\boldsymbol{J}}{\mathrm{d}\alpha} \underset{\eta \to 0}{\approx} -\eta \nabla_{\boldsymbol{J}} \epsilon_g$$
$$= -\eta \boldsymbol{\Sigma} (\boldsymbol{J} - \boldsymbol{B}). \tag{65}$$

For a diagonal a covariance matrix $\Sigma_{kl} = \delta_{kl}\lambda_{kl}$, the solution of Equation (65) is

$$J_k = \exp(-\eta \lambda_k \alpha)(J_k^0 - B_k) + B_k \tag{66}$$

for $k \in \{1, ..., L\}$ and initial weight component $J_k^0$. Thus, the individual weights are learned exponentially fast, and each component $J_k$ converges thereby to the component of the teacher. For the first-order order parameters, we find

$$R^{(1)} = \sum_{k=1}^{N} \frac{J_k \lambda_k B_k}{N} = \frac{1}{N} \sum_{k=1}^{N} \left( \exp(-\eta \lambda_k \alpha)(J_k^0 - B_k) + B_k \right) \lambda_k B_k \tag{67}$$

$$Q^{(1)} = \sum_{k=1}^{N} \frac{J_k \lambda_k J_k}{N} = \frac{1}{N} \sum_{k=1}^{N} \left( \exp(-\eta \lambda_k \alpha)(J_k^0 - B_k) + B_k \right)^2 \lambda_k \tag{68}$$

and for their corresponding expectation value

$$\langle R^{(1)} \rangle_{J_k^0, B_k} = 1 - \frac{1}{N} \sum_{k=1}^{N} \lambda_k \exp(-\eta \lambda_k \alpha) \tag{69}$$

$$\langle Q^{(1)} \rangle_{J_k^0, B_k} = 1 + \frac{1 + \sigma_J^2}{N} \sum_{k=1}^{N} \lambda_k \exp(-2\eta \lambda_k \alpha) - \frac{2}{N} \sum_{k=1}^{N} \lambda_k \exp(-\eta \lambda_k \alpha) \tag{70}$$

For the expectation value of the generalization error, we obtain

$$\langle \epsilon_g \rangle_{J_k^0, B_k} = \frac{1 + \sigma_J^2}{2N} \sum_{k=1}^{N} \lambda_k \exp(-2\eta \lambda_k \alpha) \tag{71}$$

$$= \frac{1 + \sigma_J^2}{2L} \sum_{k=1}^{L} \lambda_k \exp(-2\eta \lambda_k \alpha) \tag{72}$$

where we have exploit that $\frac{N}{L} \in \mathbb{N}$ as assumed for our setup (see Section 3). Note that Equation (72) that we derived from the approximated Langevin equation, is the same as the generalization error derived from the statistical mechanics approach in the small learning rate limit $\eta \to 0$ given by Equation (8). Therefore, neglecting the higher order of the learning rate in the differential equations is equivalent to neglecting path fluctuations of the stochastic gradient descent.

Next, in order to model how the generalization error scales as more and more feature directions are explored, we assume that $N_l$ components of the student are already learned. Thereby, the other $N - N_l$ components are kept fixed and random. Here, we want to investigate the generalization error as a function of $N_l$. For this, we make the ansatz:

$$J_k = \begin{cases} B_k & \text{for } k = 1, \dots, N_l \\ J_k^0 & \text{else} \end{cases} \tag{73}$$

Therefore, we obtain two different parts for the order parameters

$$R^{(1)} = \frac{1}{N} \sum_{k=1}^{N} \lambda_k J_k B_k = \frac{1}{N} \left( \sum_{k=1}^{N_l} \lambda_k B_k^2 + \sum_{k=N_l+1}^{N} \lambda_k J_k^0 B_k \right)$$

$$Q^{(1)} = \frac{1}{N} \sum_{k=1}^{N} \lambda_k J_k J_k = \frac{1}{N} \left( \sum_{k=1}^{N_l} \lambda_k B_k^2 + \sum_{k=N_l+1}^{N} \lambda_k J_k^0 J_k^0 \right) \tag{74}$$

and their expectation values become

$$\langle R^{(1)} \rangle_{J_{k,0}, B_k} = \frac{1}{N} \sum_{a=1}^{N_l} \lambda_a,$$

$$\langle Q^{(1)} \rangle_{J_{k,0}, B_k} = \frac{1}{N} \sum_{k=1}^{N_l} \lambda_k + \frac{\sigma_J^2}{N} \sum_{k=N_l+1}^{N} \lambda_k \tag{75}$$

Next, we insert Equation (75) into the expression of the generalization error and obtain

$$\langle \epsilon_g \rangle_{J_{k,0}, B_k} = \frac{1 + \sigma_J^2}{2} \left( 1 - \frac{1}{L} \sum_{k=1}^{N_l} \lambda_k \right)$$

$$= \frac{1 + \sigma_J^2}{2} \frac{1}{L} \sum_{k=N_l}^{L} \lambda_k, \tag{76}$$

where we have exploit that $\sum_k^N \lambda_k = N$ and $\frac{N}{L} \in \mathbb{N}$. Next, we use the definition of the eigenvalues $\lambda_i = \lambda_+ \left( \frac{1}{i} \right)^{1+\beta}$ and define the partial sum $S(N_l) = \frac{\lambda_+}{L} \sum_{k=N_l}^{L} \left( \frac{1}{i} \right)^{1+\beta}$ which leads to

$$\epsilon_g = \frac{1 + \sigma_J^2}{2} S(N_l) \tag{77}$$

In order to approximate the sum, we use the Euler-Maclaurin formula defined in Equation (49) and replace the sum with an integral. For $\beta > 0$, we find

$$S(N_l) \approx \frac{\lambda_+}{L} \int_{N_l}^{L} \frac{1}{i^{\beta+1}} \, di$$

$$= \frac{\lambda_+}{N\beta} \left( \frac{1}{N_l^\beta} - \frac{1}{L^\beta} \right) \tag{78}$$

Thus, we obtain

$$\langle \epsilon_{g,\text{asymp}} \rangle \approx \frac{1 + \sigma_J^2}{2} \frac{\lambda_+}{L\beta} \left( \frac{1}{N_l^\beta} - \frac{1}{L^\beta} \right). \tag{79}$$

The zeroth-order error of this approximation, which we consider as the worst-case, is $\delta\epsilon_g = \frac{1+\sigma_J^2}{4} \frac{\lambda_+}{L} \left( \frac{1}{N_l^{1+\beta}} + \frac{1}{L^{1+\beta}} \right)$.

If we do not assume that $N_l$ eigenvalues are already converged, then we can make the ansatz

$$\tilde{J}_k = \begin{cases} J_k & \text{for } k = 1, \dots, N_l \\ J_k^0 & \text{else} \end{cases} \tag{80}$$

where $J_k$ are given in Equation (66). Therefore, we again obtain two different parts for the order parameters

$$
\begin{aligned}
R^{(1)} &= \frac{1}{N} \sum_{k=1}^{N} \lambda_k \tilde{J}_k B_k \\
&= \frac{1}{N} \left( \sum_{k=1}^{N_l} \lambda_k J_k + \sum_{k=N_l+1}^{N} \lambda_k J_k^0 B_k \right) \\
&= \frac{1}{N} \sum_{k=1}^{N} \lambda_k \left[ \exp\left( -\eta \lambda_k \alpha \right) \left( J_k^0 - B_k \right) + B_k \right] B_k + \frac{1}{N} \sum_{k=N_l+1}^{N} \lambda_k J_k^0 B_k \\
Q^{(1)} &= \frac{1}{N} \sum_{k=1}^{N} \lambda_k \tilde{J}_k \tilde{J}_k \\
&= \frac{1}{N} \left( \sum_{k=1}^{N_l} \lambda_k J_k^2 + \sum_{k=N_l+1}^{N} \lambda_k J_k^0 J_k^0 \right) \\
&= \frac{1}{N} \left( \sum_{k=1}^{N_l} \lambda_k \left[ \exp\left( -\eta \lambda_k \alpha \right) \left( J_k^0 - B_k \right) + B_k \right]^2 + \sum_{k=N_l+1}^{N} \lambda_k J_k^0 J_k^0 \right)
\end{aligned}
\tag{81}
$$

and their expectation values become

$$
\begin{aligned}
\langle R^{(1)} \rangle_{J_k^0, B_k} &= \frac{1}{N} \sum_{k=1}^{N_l} \lambda_k - \frac{1}{N} \sum_{k=1}^{N_l} \lambda_k \exp\left( -\eta \lambda_k \alpha \right), \\
\langle Q^{(1)} \rangle_{J_k^0, B_k} &= \frac{1 + \sigma_J^2}{N} \sum_{k=1}^{N_l} \lambda_k \exp\left( -2\eta \lambda_k \alpha \right) - \frac{2}{N} \sum_{k=1}^{N_l} \lambda_k \exp\left( -\eta \lambda_k \alpha \right) \\
&\quad + \frac{1}{N} \sum_{k=1}^{N_l} \lambda_k + \frac{\sigma_J^2}{N} \sum_{k=N_l+1}^{N} \lambda_k
\end{aligned}
\tag{82}
$$

Note that Equation (82 reduces to Equation (75) for $\alpha \to \infty$. Next, we insert Equation (82) into the expression of the generalization error and obtain

$$
\langle \epsilon_g \rangle_{J_k^0, B_k} = \frac{1 + \sigma_J^2}{2L} \left[ \sum_{k=1}^{N_l} \lambda_k \exp\left( -2\eta \lambda_k \alpha \right) + \sum_{k=N_l}^{L} \lambda_k \right]
\tag{83}
$$

Note that Equation (83) is basically a combination of Equation (76) and Equation (72). If we set $N_l = N$ in Equation (83), then we can reproduce Equation (72), and if we let $\alpha \to \infty$ in Equation (83), then we reach the asymptotic generalization error given by Equation (76). Therefore, we can repeat the analysis of this and the previous subsection separately and finally obtain the generalization error for small learning rates

$$
\epsilon_g \left( N_l, \alpha \right) \underset{\eta \to 0}{\approx} \frac{1 + \sigma^2}{2} \frac{\lambda_+}{L} \left[ \frac{1}{\beta} \left( \frac{1}{N_l^\beta} - \frac{1}{L^\beta} \right) \right.
$$
$$
\left. + \frac{(2\eta \lambda_+ \alpha)^{-\frac{\beta}{1+\beta}}}{1 + \beta} \left[ \Gamma\left( \frac{\beta}{1+\beta}, \frac{2\eta \lambda_+ \alpha}{L^{\beta+1}} \right) - \Gamma\left( \frac{\beta}{1+\beta}, 2\eta \lambda_+ \alpha \right) \right] \right].
\tag{84}
$$

Next, we consider a general $\boldsymbol{\Sigma}$, which is no longer diagonal. The solution of the Langevin equation is

$$
\tilde{\mathbf{J}} = \exp(-\eta \mathbf{D} \alpha)(\tilde{\mathbf{J}}^{(0)} - \tilde{\mathbf{B}}) + \tilde{\mathbf{B}},
\tag{85}
$$

where we have introduced the eigendecomposition $\boldsymbol{\Sigma} = \mathbf{W} \mathbf{D} \mathbf{W}^\top$ and expressed the weights in the eigenbasis $\tilde{\mathbf{J}} = \mathbf{W}^\top \mathbf{J}, \tilde{\mathbf{B}} = \mathbf{W}^\top \mathbf{B}$. Therefore, we obtain for the order parameters

$$
\begin{aligned}
Q^{(1)} &= \frac{\mathbf{J}^\top \boldsymbol{\Sigma} \mathbf{J}}{N} = \frac{\mathbf{J}^\top \mathbf{W} \mathbf{D} \mathbf{W}^\top \mathbf{J}}{N} = \frac{\tilde{\mathbf{J}}^\top \mathbf{D} \tilde{\mathbf{J}}}{N} \\
R^{(1)} &= \frac{\mathbf{J}^\top \boldsymbol{\Sigma} \mathbf{B}}{N} = \frac{\mathbf{J}^\top \mathbf{W} \mathbf{D} \mathbf{W}^\top \mathbf{B}}{N} = \frac{\tilde{\mathbf{J}}^\top \mathbf{D} \tilde{\mathbf{B}}}{N}
\end{aligned}
\tag{86}
$$

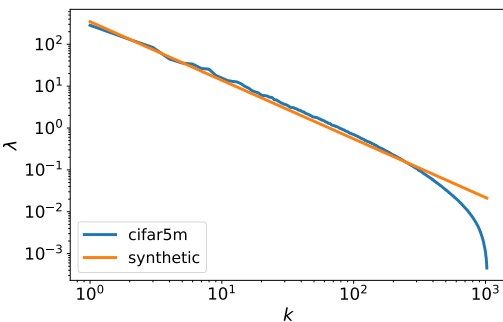

Figure 14: Left: Spectra of the feature-feature covariance matrix of CIFAR-5m data set vs. Eq. (1) for $\beta = 0.3$ and $N = L = 1024$.

Note that the expectation values for the order parameters and, therefore, for the generalization error are still the same.

We use the parametrization from Eq. (85) to obtain the results shown in the right panel of Figure 3. Thereby, we test our prediction for the generalization error from Eq. (84) to a student network trained on CIFAR-5M images using approximately $10^6$ (see Nakkiran et al. (2021) for details on the dataset). We use only the first channel of the images resulting in a total input dimension of $N = 1024$ after flattening. To approximate the true covariance matrix $\Sigma$, we numerically estimate the feature-feature covariance matrix based on the input examples from the training dataset. During training, we update only the first entries of $\tilde{J}$, while resetting the remaining entries to their initial values after each iteration. Based on the spectra of the feature-feature covariance matrix depicted in Figure 14, we estimate $\beta \approx 0.3$ and use this spectrum to evaluate Eq. (85).

### B.3.1 DEPENDENCE ON STUDENT SIZE FOR GENERAL COVARIANCE EIGENBASIS

In this Subsection, we consider the generalization error evaluated for a student with $N_l$ trainable weights and $N - N_l$ random weights. Thereby, we no longer train the weights of the student in the eigenbasis of the data-covariance matrix. In order to distinguish between trainable and untrainable components, we decompose the student vector $\boldsymbol{J}$ in two parts: the trainable components $\tilde{\boldsymbol{J}}$ and the non-trainable components $\hat{\boldsymbol{J}}$. For the covariance matrix $\Sigma$, we identify the following structure in block matrix form

$$\Sigma = \begin{pmatrix} \tilde{\Sigma} & \Sigma_{\text{cross}} \\ \Sigma_{\text{cross}}^{\top} & \hat{\Sigma} \end{pmatrix}, \tag{87}$$

where

- $\tilde{\Sigma} \in \mathbb{R}^{N_l \times N_l}$ represents the submatrix of the covariance matrix, acting exclusively on the subspace spanned by the first $N_l$ components of the vector $\boldsymbol{J}$, denoted as $\tilde{\boldsymbol{J}}$.
- $\hat{\Sigma} \in \mathbb{R}^{(N-N_l) \times (N-N_l)}$ refers to the submatrix corresponding to the subspace of the remaining $N - N_l$ components of $\boldsymbol{J}$, denoted as $\hat{\boldsymbol{J}}$.
- $\Sigma_{\text{cross}} \in \mathbb{R}^{N_l \times (N-N_l)}$ represents the cross-covariance matrix part, describing the interactions between the subspace of the first $N_l$ components and the complementary subspace of the last $N - N_l$ components

The evolution of the student vector $\boldsymbol{J}$ is governed by the differential equation for small learning rates

$$\frac{d\boldsymbol{J}_i}{d\alpha} \underset{\eta \to 0}{\approx} \eta \sum_{k=1}^{N} \Sigma_{ik}(B_k - J_k). \tag{88}$$

The solution for the trainable components $\tilde{\boldsymbol{J}}(\alpha)$ at time $\alpha$ is given by

$$\tilde{\boldsymbol{J}}(\alpha) = e^{-\eta \tilde{\Sigma} \alpha} \tilde{\boldsymbol{J}}^0 + \tilde{\boldsymbol{B}} \left( 1 - e^{-\eta \tilde{\Sigma} \alpha} \right) + \tilde{\Sigma}^{-1} \Sigma_{\text{cross}} \left( \hat{\boldsymbol{B}} - \hat{\boldsymbol{J}} \right) \left( 1 - e^{-\eta \tilde{\Sigma} \alpha} \right), \tag{89}$$

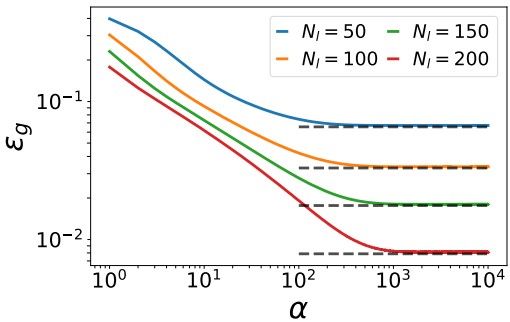

Figure 15: $\epsilon_g$ as a function of $\alpha$ for different trainable input dimensions $N_l$ of the student vector with $L = N = 256$, $K = M = 1$, $\sigma_J = 0.01$, $\eta = 0.05$, and $\beta = 1$. Here, we train the student outside the eigenbasis of the data covariance matrix. We compare results based on simulations (solid curves) to the theoretical prediction from Eq. (93) (black dashed lines). The student network is trained on synthetic data, where we average over 300 different initializations of student and teacher vectors.

where $\tilde{\boldsymbol{J}}^0$ is the initial condition for the trainable components, and the non-trainable components are kept at their random initial values. Moreover, $\tilde{\boldsymbol{B}}$ and $\hat{\boldsymbol{B}}$ are the components of the teacher vector equivalent to the trainable and non-trainable parts of the student vector, respectively. At stationarity ($\alpha \to \infty$), the trainable components become

$$\tilde{\boldsymbol{J}} = \tilde{\boldsymbol{B}} + \tilde{\boldsymbol{\Sigma}}^{-1}\boldsymbol{\Sigma}_{\text{cross}}\left(\hat{\boldsymbol{B}} - \hat{\boldsymbol{J}}\right). \tag{90}$$

For the first-order order parameters, we find

$$
\begin{aligned}
R^{(1)} &= \frac{1}{N}\boldsymbol{J}^\top\boldsymbol{\Sigma}\boldsymbol{B} \\
&= \frac{1}{N}\left(\tilde{\boldsymbol{J}}^\top\tilde{\boldsymbol{\Sigma}}\tilde{\boldsymbol{B}} + \tilde{\boldsymbol{J}}^\top\boldsymbol{\Sigma}_{\text{cross}}\hat{\boldsymbol{B}} + \hat{\boldsymbol{J}}^\top\boldsymbol{\Sigma}_{\text{cross}}^\top\tilde{\boldsymbol{B}} + \hat{\boldsymbol{J}}^\top\hat{\boldsymbol{\Sigma}}\hat{\boldsymbol{B}}\right) \\
Q^{(1)} &= \frac{1}{N}\boldsymbol{J}^\top\boldsymbol{\Sigma}\boldsymbol{J} \\
&= \frac{1}{N}\left(\tilde{\boldsymbol{J}}^\top\tilde{\boldsymbol{\Sigma}}\tilde{\boldsymbol{J}} + 2\tilde{\boldsymbol{J}}^\top\boldsymbol{\Sigma}_{\text{cross}}\hat{\boldsymbol{J}} + \hat{\boldsymbol{J}}^\top\hat{\boldsymbol{\Sigma}}\hat{\boldsymbol{J}}\right).
\end{aligned}
\tag{91}
$$

Taking the expectation over $\boldsymbol{B}$ and $\hat{\boldsymbol{J}}$ yields

$$
\begin{aligned}
\langle R^{(1)}\rangle_{B_i,\hat{J}_i} &= \frac{1}{N}\left(\text{Tr}(\tilde{\boldsymbol{\Sigma}}) + \text{Tr}\left(\boldsymbol{\Sigma}_{\text{cross}}^\top\tilde{\boldsymbol{\Sigma}}^{-1}\boldsymbol{\Sigma}_{\text{cross}}\right)\right) \\
\langle Q^{(1)}\rangle_{B_i,\hat{J}_i} &= \frac{1}{N}\left(\text{Tr}(\tilde{\boldsymbol{\Sigma}}) + (1+\sigma_J^2)\text{Tr}\left(\boldsymbol{\Sigma}_{\text{cross}}^\top\tilde{\boldsymbol{\Sigma}}^{-1}\boldsymbol{\Sigma}_{\text{cross}}\right) + \sigma_J^2\text{Tr}(\hat{\boldsymbol{\Sigma}}) - 2\sigma_J^2\text{Tr}\left(\boldsymbol{\Sigma}_{\text{cross}}^\top\tilde{\boldsymbol{\Sigma}}^{-1}\boldsymbol{\Sigma}_{\text{cross}}\right)\right).
\end{aligned}
\tag{92}
$$

Finally, we obtain for the generalization error

$$\langle\epsilon_{g,\text{asymp}}\rangle_{B_i,\hat{J}_i} = \frac{1}{2}\left[1 - \frac{1}{N}\text{Tr}(\tilde{\boldsymbol{\Sigma}}) + \frac{1}{N}\sigma_J^2\text{Tr}(\hat{\boldsymbol{\Sigma}}) - \frac{1+\sigma_J^2}{N}\text{Tr}\left(\boldsymbol{\Sigma}_{\text{cross}}^\top\tilde{\boldsymbol{\Sigma}}^{-1}\boldsymbol{\Sigma}_{\text{cross}}\right)\right]. \tag{93}$$

Note that for diagonal covariance matrices $\Sigma_{ij} = \delta_{ij}\lambda_i$, Eq. (93) reduces to Eq. (10) for $\alpha \to \infty$. Figure 15 presents the generalization error obtained from simulations for various $N_l$ as a function of $\alpha$. In this comparison, the asymptotic solution derived from Eq. (93) aligns closely with the simulation results, demonstrating excellent agreement.

Next, we compare the numerical solution of Eq. (93) with our theoretical prediction for training the student vector in the eigenbasis of the data covariance matrix, as given by Eq. (79). The results are presented in Figure 16. For a fixed number of trainable parameters, $N_l$, the generalization error is consistently lower when the student is trained in the eigenbasis of the data covariance matrix. This is expected, as training in the eigenbasis aligns with the directions of the largest $N_l$ eigenvalues, leading to a more efficient learning process. Consequently, the overall generalization error is reduced compared to training outside the eigenbasis. Under the condition $N^\beta \gg N_l^\beta$, we observe the same

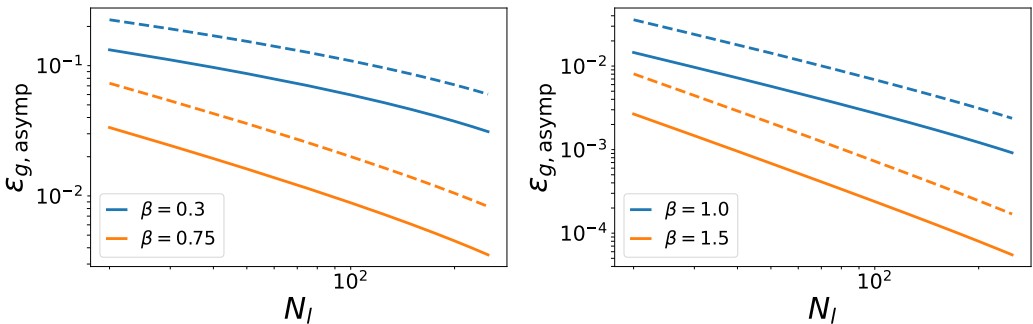

Figure 16: $\epsilon_{g,\mathrm{asymp}}$ as a function of $N_l$ for different $\beta$ and $N = L = 1024$. The student is trained with synthetic data. We compare the numerical solution of Eq. (93) (dashed) with our theoretical prediction for training the student vector in the eigenbasis of the data covariance matrix, as given by Eq. (79) (solid). The numerical solution is averaged over 100 different covariance matrices.

power-law scaling for the generalization error, $\epsilon_{g,\mathrm{asymp}} \propto N_l^{-\beta}$, in both scenarios. Additionally, for more general configurations, we find that the scaling behavior of the generalization error remains consistent across both setups. The data covariance matrices are generated by $\boldsymbol{\Sigma} = \boldsymbol{W}\boldsymbol{\Lambda}\boldsymbol{W}^\top$, where $\Lambda_{ij} = \delta_{ij}\lambda_i$ with eigenvalues defined by Eq. (1) and $\boldsymbol{W}$ is a random orthogonal matrix with zero mean and variance which scales as $\langle W_{ij}^2 \rangle \sim \frac{1}{N}$.

## C  NON-LINEAR ACTIVATION

### C.1  DIFFERENTIAL EQUATIONS

Throughout this work, we consider the error function as our non-linear activation function $g(x) = \mathrm{erf}\left(\frac{x}{\sqrt{2}}\right)$. For this activation function, one can solve the integrals $I_2, I_3$ and $I_4$ given in Section A analytically. Following Saad & Solla (1995); Yoshida & Okada (2019), one finds

$$I_2(C) = \frac{2}{\pi}\arcsin\left(\frac{C_{12}}{\sqrt{(1+C_{11})(1+C_{22})}}\right), \tag{94}$$

$$I_3(C) = \frac{2}{\pi}\frac{1}{\sqrt{(1+C_{11})(1+C_{33})-C_{13}^2}}\left(\frac{C_{23}(1+C_{11})-C_{12}C_{13}}{1+C_{11}}\right), \tag{95}$$

$$I_4(C) = \frac{4}{\pi^2}\frac{1}{\sqrt{A_4}}\arcsin\left(\frac{A_0}{\sqrt{A_1}\sqrt{A_2}}\right) \tag{96}$$

where

$$A_4 = (1+C_{11})(1+C_{22}) - C_{12}^2,$$

and

$$A_0 = A_4 C_{34} + C_{23}C_{24}(1+C_{11}) + C_{13}C_{14}(1+C_{22}) + C_{12}^2 C_{13}^2 C_{24}^2,$$

$$A_1 = A_4(1+C_{33}) + C_{23}(1+C_{11}) - C_{13}(1+C_{22}) + 2C_{12}C_{13}C_{23},$$

$$A_2 = A_4(1+C_{44}) - C_{24}(1+C_{11}) - C_{14}(1+C_{22}) + 2C_{12}C_{14}C_{24},$$

depending on the precise covariance matrix and its entries $C_{ij}$ as discussed in Section A. After evaluating all necessary covariance matrices, we obtain for $K = M$

$$\frac{dR_{in}^{(l)}}{d\alpha} = \frac{2\eta}{M\pi} \frac{1}{1+Q_{ii}} \left[ \sum_m^M \frac{T_{nm}^{(l+1)}\left(1+Q_{ii}\right) - R_{in}^{(l+1)}R_{im}}{\sqrt{\left(1+Q_{ii}\right)\left(1+T_{mm}\right) - R_{im}^2}} - \sum_j^M \frac{R_{jn}^{(l+1)}\left(1+Q_{ii}\right) - R_{in}^{(l+1)}Q_{ij}}{\sqrt{\left(1+Q_{ii}\right)\left(1+Q_{jj}\right) - Q_{ij}^2}} \right],$$

$$\tag{97}$$

$$\frac{dQ_{ik}^{(l)}}{d\alpha} = \frac{2\eta}{M\pi} \left[ \frac{1}{1+Q_{ii}} \left( \sum_m^M \frac{R_{km}^{(l+1)}\left(1+Q_{ii}\right) - Q_{ik}^{(l+1)}R_{im}}{\sqrt{\left(1+Q_{ii}\right)\left(1+T_{mm}\right) - R_{im}^2}} - \sum_j^M \frac{Q_{kj}^{(l+1)}\left(1+Q_{ii}\right) - Q_{ik}^{(l+1)}Q_{ij}}{\sqrt{\left(1+Q_{ii}\right)\left(1+Q_{jj}\right) - Q_{ij}^2}} \right) \right.$$

$$\left. + \frac{1}{1+Q_{kk}} \left( \sum_m^M \frac{R_{im}^{(l+1)}\left(1+Q_{kk}\right) - Q_{ik}^{(l+1)}R_{km}}{\sqrt{\left(1+Q_{kk}\right)\left(1+T_{mm}\right) - R_{km}^2}} - \sum_j^M \frac{Q_{ij}^{(l+1)}\left(1+Q_{kk}\right) - Q_{ik}^{(l+1)}Q_{kj}}{\sqrt{\left(1+Q_{kk}\right)\left(1+Q_{jj}\right) - Q_{kj}^2}} \right) \right]$$

$$+ \mathcal{O}\left(\eta^2\right), \tag{98}$$

where we have omitted higher-order terms in $\eta$ for notational simplicity. Similarly, we have also omitted the superscripts for the first-order parameters, meaning that all instances of $R_{in}$, $T_{nm}$, and $Q_{ij}$ without superscripts correspond to their first-order forms: $R_{in}^{(1)}$, $T_{nm}^{(1)}$, and $Q_{ij}^{(1)}$. Note that the differential equations are not closed given in Equation 98. For the last component $L-1$ one has to apply the Cayley-Hamilton theorem for the order parameters as discussed in Section 3. In the same notation, the generalization error becomes

$$\epsilon_g = \frac{1}{M\pi} \left[ \sum_{i,k} \arcsin\left( \frac{Q_{ik}^{(1)}}{\sqrt{1+Q_{ii}^{(1)}}\sqrt{1+Q_{kk}^{(1)}}} \right) + \sum_{n,m} \arcsin\left( \frac{T_{nm}^{(1)}}{\sqrt{1+T_{nn}^{(1)}}\sqrt{1+T_{mm}^{(1)}}} \right) \right.$$

$$\left. - 2\sum_{i,n} \arcsin\left( \frac{R_{in}^{(1)}}{\sqrt{1+Q_{ii}^{(1)}}\sqrt{1+T_{nn}^{(1)}}} \right) \right],$$

$$\tag{99}$$

which just depends on the first-order order parameters. This system is gonna be analyzed in the following.

## C.2    FLUCTUATIONS IN ORDER PARAMETERS

For isotropic input data, the order parameters are self-averaging, i.e., as $N$ increases, the fluctuations of the order parameters decrease, and in the thermodynamic limit $N \to \infty$, one can replace these order parameters by their expectation values (Saad & Solla, 1995). However, for structured input data, fluctuations in the order parameters can still be present. In order to show this, we calculate the relative variance of the teacher-teacher order parameters as a measure for fluctuations.

For the expectation value and variance of the teacher-teacher order parameters $T_{nm}^{(l)} = \frac{1}{N}\boldsymbol{B}_n(\boldsymbol{\Sigma})^l\boldsymbol{B}_m$, we obtain

$$\left\langle T_{nm}^{(l)} \right\rangle_{B_n,B_m} = \frac{\delta_{nm}}{N}\sum_k^N \lambda_k^l$$

$$\left\langle \left( T_{nm}^{(l)} - \langle T_{nm}^{(l)}\rangle_{B_n,B_m} \right)^2 \right\rangle_{B_n,B_m} = \frac{(1+\delta_{nm})}{N^2}\sum_k^N \lambda_k^{2l}, \tag{100}$$

where we have exploit that $\boldsymbol{B}_n$ and $\boldsymbol{B}_m$ are independently distributed random vectors with $B_{n,k} \sim \mathcal{N}(0,1)$ for each entry of the vector $\boldsymbol{B}_n$. Note that the variance for $n=m$ is twice as large as for $n \neq m$. After substituting Eq. (1) into Eq. (100), we find

$$\langle T_{nm}^{(l)}\rangle_{B_n,B_m} = \delta_{nm}\frac{\lambda_+^l}{L}\sum_{k=1}^L \frac{1}{k^{l(1+\beta)}},$$

$$\left\langle \left( T_{nm}^{(l)} - \langle T_{nm}^{(l)}\rangle_{B_n,B_m} \right)^2 \right\rangle_{B_n,B_m} = (1+\delta_{nm})\frac{\lambda_+^{2l}}{NL}\sum_{k=1}^L \frac{1}{k^{2l(1+\beta)}}. \tag{101}$$

Next, we discuss the dependence of $\lambda_+$ on $L$ and $N$. Due to the normalization condition $\frac{1}{N} \sum_{k=1}^{N} \lambda_k = 1$, we can infer $\frac{1}{N} \sum_{k=1}^{N} \lambda_k = \frac{\lambda_+}{L} \sum_{k=1}^{L} \frac{1}{k^{(1+\beta)}}$ and thus $\lambda_+ = \frac{L}{\sum_{k=1}^{L} \frac{1}{k^{(1+\beta)}}} = \mathcal{O}(L)$ since $\sum_{k}^{L} \frac{1}{k^{(1+\beta)}}$ is a convergent sum. Thus, we find

$$\frac{\lambda_+^l}{L} \sum_{k=1}^{L} \frac{1}{k^{l(1+\beta)}} = O(L^{l-1}), \qquad \frac{\lambda_+^{2l}}{NL} \sum_{k=1}^{L} \frac{1}{k^{2l(1+\beta)}} = \mathcal{O}\left(\frac{L^{2l-1}}{N}\right). \tag{102}$$

Therefore, for the relative variance for $n = m$, we obtain

$$\frac{\left\langle \left(T_{nn}^{(l)} - \langle T_{nn}^{(l)} \rangle_{B_n}\right)^2 \right\rangle_{B_n}}{\left\langle T_{nn}^{(l)} \right\rangle_{B_n}^2} = \mathcal{O}\left(\frac{L}{N}\right). \tag{103}$$

This result indicates that we need to distinguish two different scenarios when taking the thermodynamic limit $N \to \infty$: i) when $\frac{L}{N} \to 0$, the relative variance vanishes, and the order parameters become self-averaging quantities, and ii) the case where $\frac{L}{N} = \mathcal{O}(1)$ stays finite, and the teacher-teacher overlaps are not self-averaging. We note that the case of isotropic inputs is recovered for $L = 1$, where the relative variance vanishes with $1/N$.

For the case $l = 0$, we find $\sum_{k=1}^{L} \frac{1}{k^{l(1+\beta)}} = \sum_{k=1}^{L} 1 = L$ which is no longer converging. Therefore, we obtain

$$\frac{\left\langle \left(T_{nn}^{(0)} - \langle T_{nn}^{(0)} \rangle_{B_n}\right)^2 \right\rangle_{B_n}}{\langle T_{nn}^{(0)} \rangle_{B_n, B_n}^2} = O\left(\frac{1}{N}\right), \tag{104}$$

which decreases with $1/N$.

## C.3 PLATEAU HEIGHT

Here, we consider the differential equations given in Eq. (98) for the higher-order order parameters up to the first order in the learning rate $\mathcal{O}(\eta)$. As already mentioned in the main text, the order parameters are no longer self-averaging, i.e. we cannot replace the random variables $T_{nm}^{(l)}$ by their expectation value in the thermodynamic limit $N \to \infty$. However, without any assumptions on the teacher-teacher order parameters, we find approximately the following fixed point for $l = 1$

$$R_{in}^{*(1)} \approx \frac{1}{\sqrt{\frac{M}{T_{nn}^{(1)} + d_n^{(1)}} \left(\frac{MT_{nn}^{(1)}}{T_{nn}^{(1)} + d_n^{(1)}} \left(1 + \frac{1}{T_{nn}^{(1)}}\right) - 1\right)}}, \quad Q^{*(1)} \approx \frac{1}{M} \sum_{n}^{M} \frac{1}{\frac{MT_{nn}^{(1)}}{T_{nn}^{(1)} + d_n^{(1)}} \left(1 + \frac{1}{T_{nn}^{(1)}}\right) - 1} \tag{105}$$

and for $l \neq 1$

$$R_{in}^{*(l)} \approx \frac{T_{nn}^{(l)}}{\sqrt{\frac{M}{T_{nn}^{(1)} + d_n^{(1)}} \left(\frac{MT_{nn}^{(1)}}{T_{nn}^{(1)} + d_n^{(1)}} \left(1 + \frac{1}{T_{nn}^{(1)}}\right) - 1\right)}}, \quad Q^{*(l)} \approx \frac{1}{M} \sum_{n}^{M} \frac{T_{nn}^{(l)}}{\frac{MT_{nn}^{(1)}}{T_{nn}^{(1)} + d_n^{(1)}} \left(1 + \frac{1}{T_{nn}^{(1)}}\right) - 1}, \tag{106}$$

with $d_n^{(1)} = \sum_{m, m \neq n}^{M-1} T_{nm}^{(1)}$. Thus, for the student-teacher order parameters, we obtain $M$ different plateau heights for each order $l$ depending on the sum over the off-diagonal entries of the higher-order teacher-teacher order parameters $d_n^{(1)}$ and $T_{nn}^{(1)}$. This approximation is exact if all $T_{nm}^{(l)} = D^{(l)}$ for $n \neq m$ and $T_{nn} = T^{(l)}$ and all plateau heights are the same $R_{in}^{*(l)} = R^{*(l)}$. In Figure 17, we compare the numerically found generalization error by evaluating the differential equations given in Eq. (98) up to $\mathcal{O}(\eta)$ with the generalization error based on the approximately found fixed points given in Eq. (105). For small $L$, we find very good agreement between the true plateau and the approximation. In order to proceed, we make the ansatz for the off-diagonal entries $T_{nm}^{(l)} = D^{(l)} = \frac{1}{M} \sum_{n}^{M} d_n^{(l)}$

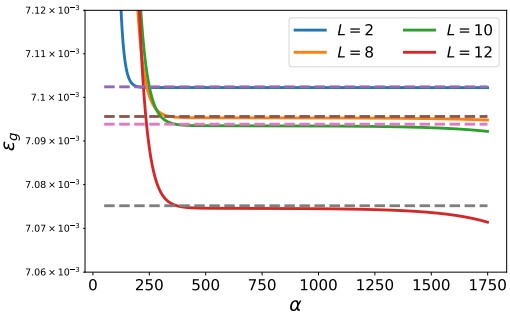

Figure 17: Comparison of the generalization error plateau evaluated by numerical solutions of the differential equations given in Eq. (98) (solid) and approximated fixed point given by Eq. (105) (dashed) for $N = 7000$, $K = M = 4$, $\sigma_J = 0.01$, $\beta = 0.25$ and $\eta = 0.1$. We solve the differential equations up to $\mathcal{O}(\eta)$.

for $n \neq m$. This approach preserves the statistical properties of the sum represented by a single parameter $D^{(l)}$. Moreover, for $l = 0$, the teacher-teacher order parameters are still self-averaging in the thermodynamic limit, and we can assume $T_{nn}^{(0)} = 1$ and $T_{nm}^{(0)} = 0$ for large $N$. Thus in summary, we assume $T_{nm}^{(0)} = \delta_{nm}$ and $T_{nm}^{(l)} = \delta_{nm}T^{(l)} + (1 - \delta_{nm})D^{(l)}$ for $l \neq 0$. After these assumptions, we find new stationary points for our differential equations for $l = 1$

$$R^{*(1)} = \frac{1}{\sqrt{\frac{M}{T^{(1)}+D^{(1)}}\left(\frac{MT^{(1)}}{T^{(1)}+D^{(1)}}\left(1+\frac{1}{T^{(1)}}\right)-1\right)}}, \quad Q^{*(1)} = \frac{1}{\frac{MT^{(1)}}{T^{(1)}+D^{(1)}}\left(1+\frac{1}{T^{(1)}}\right)-1}, \quad (107)$$

and $l \neq 1$

$$R^{*(0)} = \frac{1}{T^{(1)} + D^{(1)}}\frac{1}{\sqrt{\frac{M}{T^{(1)}+D^{(1)}}\left(\frac{MT^{(1)}}{T^{(1)}+D^{(1)}}\left(1+\frac{1}{T^{(1)}}\right)-1\right)}},$$

$$Q^{*(0)} = \frac{1}{T^{(1)} + D^{(1)}}\frac{1}{\frac{MT^{(1)}}{T^{(1)}+D^{(1)}}\left(1+\frac{1}{T^{(1)}}\right)-1}$$

$$R^{*(l)} = \frac{T^{(l)}}{T^{(1)}}\frac{1}{\sqrt{\frac{M}{T^{(1)}+D^{(1)}}\left(\frac{MT^{(1)}}{T^{(1)}+D^{(1)}}\left(1+\frac{1}{T^{(1)}}\right)-1\right)}},$$

$$Q^{*(l)} = \frac{T^{(l)}}{T^{(1)}}\frac{1}{\frac{MT^{(1)}}{T^{(1)}+D^{(1)}}\left(1+\frac{1}{T^{(1)}}\right)-1}. \quad (108)$$

As one can see in Eqs. (107), and (108), we end up in one plateau height for the order parameters $Q_{ij}^{(l)} = Q^{*(l)}$ and $R_{im}^{(l)} = R^{*(l)}$. In addition to Figure 4 given in the main text, we provide the plateau behavior for higher order-order parameters in Figure 18 and compare our newly obtained stationary solutions given in Eq. (108) with the solutions of the differential equations given in Eq. (98) up to $\mathcal{O}(\eta)$. We observe that the student-teacher order parameters defined in given in Eq. (108) yield an approximation for the mean value of the $M$ groups of order parameters. For the student-student order parameters, we observe a small systematic error which appears to be small compared to the magnitude of $Q^{*(l)}$.

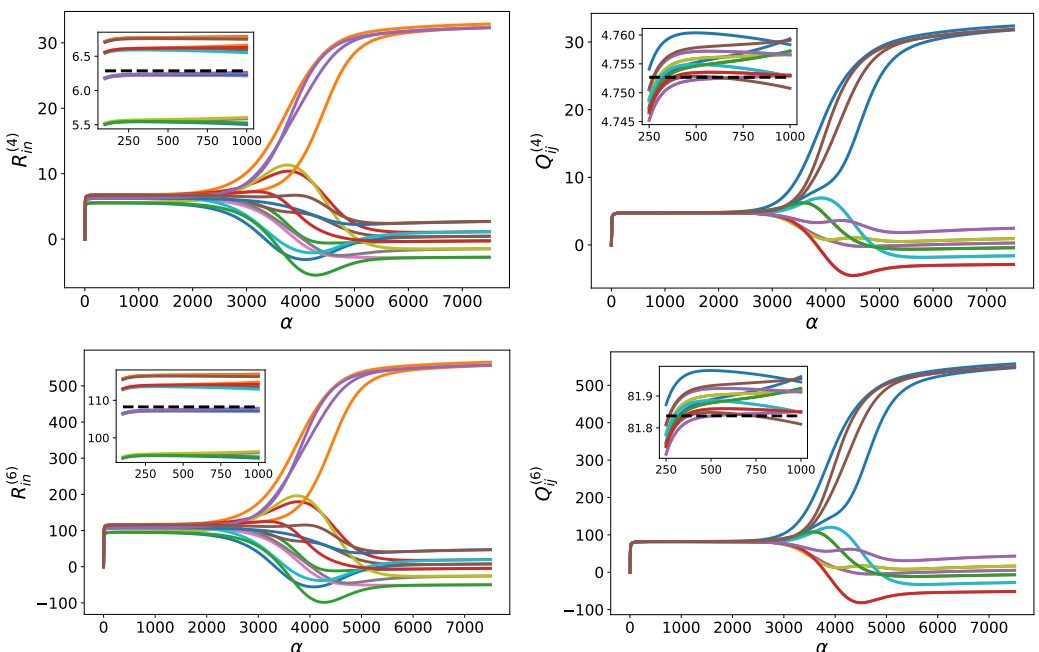

Figure 18: Plateau behavior of the order parameters with $L = 10$, $N = 7000$, $\sigma_J = 0.01$, $\eta = 0.1$ and $M = K = 4$ for one random initialization of student and teacher vectors. We solve the differential equations for the small learning rate case where we consider terms up to $\mathcal{O}(\eta)$. The inset shows the higher-order order parameters at the plateau. For the student-teacher order parameters, we obtain $M$ different plateau heights. For the student-student order parameters, we observe one plateau height with small statistical deviations for the particular matrix entry $Q_{ij}^{(l)}$. The dashed horizontal lines in the insets show the plateau heights according to Eq. (108).

Next, we insert Eq. (107) into the expression of the generalization error and obtain

$$
\epsilon_g^* = \frac{1}{\pi} \left( M \arcsin \left( \frac{D^{(1)} + T^{(1)}}{MT^{(1)} + M} \right) + (M - 1) \arcsin \left( \frac{D^{(1)}}{T^{(1)} + 1} \right) \right.
$$

$$
- 2M \arcsin \left( \frac{1}{\sqrt{T^{(1)} + 1} \sqrt{\frac{-(D^{(1)}M - M^2 - (M^2 - M)T^{(1)})}{(D^{(1)})^2 + 2D^{(1)}T^{(1)} + (T^{(1)})^2}} \sqrt{\frac{MT^{(1)} + M}{(M-1)T^{(1)} - D^{(1)} + M}}} \right)
$$

$$
\left. + \arcsin \left( \frac{T^{(1)}}{T^{(1)} + 1} \right) \right)
\tag{109}
$$

for the plateau height. The right panel in Figure 4 shows an example for the estimated plateau height given by Equation (109) against the numerical solution of the differential equations.

### C.4 PLATEAU ESCAPE

In this subsection, we want to present the escape from the plateau. The found stationary equations given by Eqs. (107) and (108) are unstable such that after a certain time on the plateau, the generalization error will eventually escape from it. In order to escape from the plateau, the unique solution of the fixed points must be broken for each $l$ at a certain time. We want to study the dynamics in the vicinity of the fixed point and clarify how the generalization error leaves it. For this, introduce parameters $S^{(l)}$ and $C^{(l)}$ indicating the onset of specialization for the student vectors towards one teacher vector. Therefore, parameterized the order parameters by $R_{im}^{(l)} = R^{(l)}\delta_{im} + S^{(l)}(1 - \delta_{im})$, $Q_{ij}^{(l)} = Q^{(l)}\delta_{ij} + C^{(l)}(1 - \delta_{ij})$. Moreover, to study the escape from the plateau, we introduce small perturbation parameters $r^{(l)}, s^{(l)}, q^{(l)}$ and $c^{(l)}$ modeling the repelling characteristic of the unstable

fixed point. Thus we parametrized the order parameters by their stationary solution and a small perturbation $R^{(l)} = R^{*(l)} + r^{(l)}$, $S^{(l)} = S^{*(l)} + s^{(l)}$, $Q^{(l)} = Q^{*(l)} + q^{(l)}$ and $C^{(l)} = C^{*(l)} + c^{(l)}$ with $S^{*(l)} = R^{*(l)}$ and $C^{*(l)} = Q^{*(l)}$. Next, we insert this parametrization into the differential equations given by Eq. (98) up to $\mathcal{O}(\eta)$. In order to study the dynamics in the vicinity of the fixed point, we linearized the dynamical equations in $\left(r^{(l)}, s^{(l)}, q^{(l)}, c^{(l)}\right)^\top$ around zero.

In the following, we present the differential equations, eigenvalues, and eigenvectors for the specific case where $D^{(l)} = 0$ and $T^{(1)} = 1$ for notational simplicity. The full system is too large to display in its entirety. However, we provide insights throughout on how these results generalize. After reducing the system of differential equations, we conclude by presenting the full solution for the general case.

After a first-order Taylor expansion in $\left(r^{(l)}, s^{(l)}, q^{(l)}, c^{(l)}\right)^\top$, we find the following linearized equation

$$\frac{d}{d\alpha} \begin{pmatrix} \boldsymbol{r} \\ \boldsymbol{s} \\ \boldsymbol{q} \\ \boldsymbol{c} \end{pmatrix} = \frac{2}{\pi M} \sqrt{\frac{2M-1}{2M+1}} \boldsymbol{A}_p \begin{pmatrix} \boldsymbol{r} \\ \boldsymbol{s} \\ \boldsymbol{q} \\ \boldsymbol{c} \end{pmatrix} \tag{110}$$

with $r_i = r^{(i-1)}$, $s_i = s^{(i-1)}$, $q_i = q^{(i-1)}$, $c_i = c^{(i-1)}$ and $\boldsymbol{A}_p = \boldsymbol{A} + \boldsymbol{B}$ with $\boldsymbol{A}_\mathrm{p} \in \mathbb{R}^{4L \times 4L}$. The individual matrices can be written as Kronecker products $\boldsymbol{A} = \boldsymbol{G} \otimes \boldsymbol{A}_1$, $\quad \boldsymbol{B} = \boldsymbol{H} \otimes \boldsymbol{U}$ with $\boldsymbol{U} = \boldsymbol{u}\boldsymbol{e}_2^\top$ and

$$\boldsymbol{G} = \begin{bmatrix} 1 & M-1 & 0 & 0 \\ 1 & M-1 & 0 & 0 \\ -2\sqrt{\frac{M}{2M-1}} & -2\sqrt{\frac{M}{2M-1}}(M-1) & 2 & 2(M-1) \\ -2\sqrt{\frac{M}{2M-1}} & -2\sqrt{\frac{M}{2M-1}}(M-1) & 2 & 2(M-1) \end{bmatrix}, \tag{111}$$

$$\boldsymbol{H} = \begin{bmatrix} \frac{-M}{(2M+1)(2M-1)} & \frac{-2M(M-1)}{(2M+1)(2M-1)} & \frac{1}{2}\frac{2M^2+3M-2}{\sqrt{M}(2M+1)\sqrt{2M-1}} & \frac{(M-1)\sqrt{2M-1}}{\sqrt{M}(2M+1)} \\ \frac{-2M}{(2M+1)(2M-1)} & \frac{-M(2M-3)}{(2M+1)(2M-1)} & \frac{1}{2}\frac{2M^2+3M-2}{\sqrt{M}(2M+1)\sqrt{2M-1}} & \frac{(M-1)\sqrt{2M-1}}{\sqrt{M}(2M+1)} \\ \frac{-2\sqrt{M}}{(2M+1)\sqrt{2M-1}} & \frac{-2(M-1)\sqrt{M}}{(2M+1)\sqrt{2M-1}} & \frac{M+2}{2M+1} & \frac{2(M-1)}{2M+1} \\ \frac{-2\sqrt{M}}{(2M+1)\sqrt{2M-1}} & \frac{-2(M-1)\sqrt{M}}{(2M+1)\sqrt{2M-1}} & \frac{M+2}{2M+1} & \frac{2(M-1)}{2M+1} \end{bmatrix}. \tag{112}$$

We obtain the following eigenvalues and eigenvectors for $\boldsymbol{G}$

$$\lambda_{G,1} = M, \qquad \mathbf{v}_{G,1} = \left(1, 1, \frac{2\sqrt{M}}{\sqrt{2M-1}}, \frac{2\sqrt{M}}{\sqrt{2M-1}}\right)^\top, \tag{113}$$

$$\lambda_{G,2} = 2M, \qquad \mathbf{v}_{G,2} = (0, 0, 1, 1)^\top, \tag{114}$$

$$\lambda_{G,3} = 0, \qquad \mathbf{v}_{G,3} = \left(1, -\frac{1}{M-1}, 0, 0\right)^\top, \quad \mathbf{v}_{G,4} = \left(0, 0, 1, -\frac{1}{M-1}\right)^\top, \tag{115}$$

and for $\boldsymbol{H}$

$$\lambda_{H,1} = \frac{M}{4M^2-1}, \quad \mathbf{v}_{H,1} = \left(1, -\frac{1}{M-1}, 0, 0\right)^\top, \tag{116}$$

$$\lambda_{H,2} = \frac{2M}{2M+1}, \quad \mathbf{v}_{H,2} = \left(1, 1, \frac{2\sqrt{M}}{\sqrt{2M-1}}, \frac{2\sqrt{M}}{\sqrt{2M-1}}\right)^\top, \tag{117}$$

$$\lambda_{H,3} = 0, \qquad \mathbf{v}_{H,3} = \left(1, 1, 0, \frac{M^{3/2}\sqrt{2M-1}}{2M^2-3M+1}\right), \mathbf{v}_{H,4} = \left(0, 0, 1, -\frac{1}{2}\frac{2M^2+3M-2}{2M^2-3M+1}\right)^\top. \tag{118}$$

Since $\boldsymbol{A}$ is of block matrix structure expressed by a Kronecker product, we obtain for its spectrum $\lambda_A = \lambda_G \lambda_{A_1}$ and corresponding eigenvectors $\boldsymbol{v}_A = \boldsymbol{v}_G \otimes \boldsymbol{v}_{A_1}$ for which we multiply each eigenvalue of $\boldsymbol{A}_1$ with each of $\boldsymbol{G}$. The same also applies for the eigenvalues and -vectors of $\boldsymbol{B}$: $\lambda_B = \lambda_G \lambda_{A_1}$ and corresponding eigenvectors $\boldsymbol{v}_B = \boldsymbol{v}_G \otimes \boldsymbol{v}_{A_1}$. The eigenvalues of $\boldsymbol{A}_1$ were already studied in Subsection B.1 and are the negative eigenvalues of the data covariance matrix $-\lambda_k$ with eigenvectors $\boldsymbol{v}_k$ summarized by the matrix $\boldsymbol{V}$ (cf. Eq. (33)). Since $\boldsymbol{A}_1$ possesses $L$ eigenvalues and -vectors, we obtain multiple groups of different eigenvalues and -vectors for $\boldsymbol{A}$ and $\boldsymbol{B}$. The eigenvalues and -vectors of $\boldsymbol{U}$ are also already known. We have one eigenvector $\boldsymbol{u}$ with eigenvalue $\lambda_u = T^{(2)}$ and $L - 1$ eigenvectors $\boldsymbol{e}_l$ with zero eigenvalue for $l = 1, 3, 4..., L$. Thereby $\boldsymbol{e}_l$ is the $l$th unit vector. In the following, the superscript for the eigenvalues and -vectors indicates the corresponding group.

For $\boldsymbol{A}$, the first two groups of eigenvalue combinations $\lambda_{A,k}^{(1)} = -M\lambda_k$ with eigenvector $\boldsymbol{v}_{A,k}^{(1)} = \left(\boldsymbol{v}_k, \boldsymbol{v}_k, 2\boldsymbol{v}_k \frac{\sqrt{M}}{\sqrt{2M-1}}, 2\boldsymbol{v}_k \frac{\sqrt{M}}{\sqrt{2M-1}}\right)^\top$ and $\lambda_{A,k}^{(2)} = -2M\lambda_k$ with $\boldsymbol{v}_{A,k}^{(2)} = (0, 0, \boldsymbol{v}_k, \boldsymbol{v}_k)^\top$ are plateau attractive. Their corresponding eigenvalues are negative and their directions are against the breaking of order parameter symmetry. The latter fact can be seen that the first two entries of the eigenvectors and the last two are the same. This would drive the dynamics in the direction corresponding to $r^{(l)} = s^{(l)}$ and $q^{(l)} = c^{(l)}$ which is exactly the plateau condition. The third group of eigenvalue combinations $\lambda_{A,k}^{(3)} = 0$ with eigenvectors $\boldsymbol{v}_{A,k}^{(3)} = \left(\boldsymbol{v}_k, -\boldsymbol{v}_k \frac{1}{M-1}, 0, 0\right)^\top$ and $\boldsymbol{v}_{A,k}^{(4)} = \left(0, 0, \boldsymbol{v}_k, -\boldsymbol{v}_k \frac{1}{M-1}\right)^\top$ are neither attractive nor repelling. However, their directions indicate a symmetry breaking in the order parameters, at least for one group $r^{(l)} \neq s^{(l)}$ or $q^{(l)} \neq c^{(l)}$.

For the matrix $\boldsymbol{B}$, we observe that $\lambda_{B,1}^{(1)} = 0$, with a total of $4L - 2$ distinct eigenvectors. However, the more significant impact comes from the directions associated with non-zero eigenvalues. These eigenvalues play a crucial role in influencing the spectrum of $\boldsymbol{A}$, particularly when $\boldsymbol{B}$ is viewed as a non-negligible perturbation of $\boldsymbol{A}$. The second eigenvalue of $\boldsymbol{B}$ is $\lambda_{B,2} = \frac{MT^{(2)}}{4M^2-1}$ with eigenvector $\boldsymbol{v}_{B,2} = \left(\boldsymbol{u}, -\boldsymbol{u} \frac{1}{M-1}, 0, 0\right)^\top$. For the third one, we obtain $\lambda_{B,3} = \frac{2MT^{(2)}}{2M+1}$ with eigenvector $\boldsymbol{v}_{B,3} = \left(\boldsymbol{u}, \boldsymbol{u}, \frac{2\sqrt{M}}{\sqrt{2M-1}}\boldsymbol{u}, \frac{2\sqrt{M}}{\sqrt{2M-1}}\boldsymbol{u}\right)^\top$.

In summary, we obtain two important directions for the escape of the plateau. The first one corresponds to the eigenvectors $\boldsymbol{v}_{A,k}^{(3)}$ and $\boldsymbol{v}_{B,2}$ and the second one is in the direction of $\boldsymbol{v}_{B,3}$ and $\boldsymbol{v}_{A,k}^{(1)}$. Note that these directions are also present for the sum of $\boldsymbol{A} + \boldsymbol{B}$ resulting in $\boldsymbol{A}_p$. Therefore, we make as a first ansatz $q^{(l)} = c^{(l)}$ since this condition is fulfilled for all important eigendirections. Moreover, we can make the following second ansatz $q^{(l)} = c^{(l)} = 2R^{*^{(l)}}\left(r^{(l)} + (M-1)s^{(l)}\right)$. The second ansatz is fulfilled by both eigendirections as well. For $D^{(l)} \neq 0$ and general $T^{(l)}$, we find with similar steps the relation $q^{(l)} = c^{(l)} = \frac{2T^{(l)}}{T^{(1)}+D^{(1)}}R^{*^{(l)}}\left(r^{(l)} + (M-1)s^{(l)}\right)$.

Next, we re-parametrize the dynamical equations under the condition $q^{(l)} = c^{(l)} = 2R^{*^{(l)}}\left(r^{(l)} + (M-1)s^{(l)}\right)$ and find

$$\frac{d}{d\alpha}\begin{pmatrix}\boldsymbol{r}\\\boldsymbol{s}\end{pmatrix} = \frac{2}{\pi M}\sqrt{\frac{2M-1}{2M+1}}\boldsymbol{A}_{rs}\begin{pmatrix}\boldsymbol{r}\\\boldsymbol{s}\end{pmatrix}, \tag{119}$$

with $\boldsymbol{A}_{rs} = \boldsymbol{A} + \boldsymbol{B}$ and $\boldsymbol{A}_{\text{rs}} \in \mathbb{R}^{2L \times 2L}$. The matrices $\boldsymbol{A}$ and $\boldsymbol{B}$ are re-defined as follows:

$$\boldsymbol{A} = \boldsymbol{G} \otimes \boldsymbol{A}_1, \qquad\qquad \boldsymbol{B} = \boldsymbol{H} \otimes \boldsymbol{U} \tag{120}$$

with re-defined $\boldsymbol{G}$ and $\boldsymbol{H}$

$$\boldsymbol{H} = \frac{1}{(2M+1)(2M-1)}\begin{bmatrix}5M-3 & 4M^2-7M+3\\\frac{4M^2-7M+3}{M-1} & 4M^2-6M+3\end{bmatrix}, \qquad \boldsymbol{G} = \begin{bmatrix}1 & (M-1)\\1 & (M-1)\end{bmatrix} \tag{121}$$

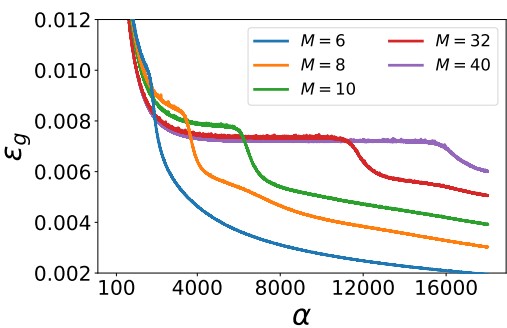

Figure 19: Plateau behavior of the generalization error $\epsilon_g$ from simulations with a single random initialization of student and teacher vectors. Parameters: $K = M$, $\eta = 0.01$, $\sigma_J = 10^{-6}$, and $\beta = 0.25$. The results are shown for varying student-teacher sizes with $L = N = 512$.

For $\boldsymbol{A}$, the eigenvectors $\boldsymbol{v}_A$ are given by $\boldsymbol{v}_A = \boldsymbol{v}_G \otimes \boldsymbol{v}_{A_1}$, where $\lambda_{A_1,k} = -\lambda_k$ and $\boldsymbol{v}_{A_1,k} = \boldsymbol{v}_k$ for $k \in (1, \ldots, L)$. The corresponding groups of eigenvalues are $\lambda_{A,k}^{(1)} = -M\lambda_k$, with eigenvectors $\boldsymbol{v}_{A,k}^{(1)} = (\boldsymbol{v}_k, \boldsymbol{v}_k)^\top$. The second group is given by $\lambda_A^{(2)} = 0$ and the corresponding eigenvectors are $\boldsymbol{v}_{A,k}^{(2)} = \left(\boldsymbol{v}_k, \frac{-\boldsymbol{v}_k}{M-1}\right)^\top$.

For $\boldsymbol{B}$, the first eigenvalue is $\lambda_{B,1} = \frac{MT^{(2)}}{(2M-1)(2M+1)}$, with the corresponding eigenvector $\boldsymbol{v}_{B,1} = \left(\boldsymbol{u}, \frac{-\boldsymbol{u}}{M-1}\right)^\top$. The second eigenvalue is $\lambda_{B,2} = \frac{2MT^{(2)}}{2M+1}$, with eigenvector $\boldsymbol{v}_{B,2} = (\boldsymbol{u}, \boldsymbol{u})^\top$. Furthermore, we have multiple eigenvectors for the eigenvalue zero. For $m \in (1, 3, 4, \ldots, L)$, we have $\lambda_{B,m} = 0$, with corresponding eigenvectors $\boldsymbol{v}_{B,m} = \left(\boldsymbol{e}_m, \frac{-\boldsymbol{e}_m}{M-1}\right)^\top$. Similarly, for $n \in (1, 3, 4, \ldots, L)$, $\lambda_{B,n} = 0$, with eigenvectors $\boldsymbol{v}_{B,n} = (\boldsymbol{e}_n, \boldsymbol{e}_n)^\top$.

Note that all eigenvalues were already encountered for the larger system verifying our analytical ansatz. Moreover, the new eigenvectors are the first two entries of the eigenvectors of the large original system.

Due to the special structure of $\boldsymbol{A}$, the eigenvector $\boldsymbol{v}_{B,1}$ are also eigenvectors of $\boldsymbol{A}$, both associated with the eigenvalue 0, meaning $\boldsymbol{A}\boldsymbol{v}_{B,m} = 0$ and $\boldsymbol{A}\boldsymbol{v}_{B,1} = 0$. Among the eigenvalues of $\boldsymbol{A}_{rs}$, $2L - 2$ of them are zero. The first non-zero eigenvalue is $\lambda_{A_{rs},1} = \frac{MT^{(2)}}{(2M-1)(2M+1)}$ which is larger than zero indicating a repelling character for direction $\boldsymbol{v}_{B,1}$ and $\boldsymbol{v}_{A,k}^{(2)}$. For the eigenvector $\boldsymbol{v}_{B,2}$, we have $\boldsymbol{A}\boldsymbol{v}_{B,2} = M\left(\boldsymbol{A}_1\boldsymbol{u}, \boldsymbol{A}_1\boldsymbol{u}\right)^\top$. Therefore, $\boldsymbol{v}_{B,2}$ is an eigenvector of $\boldsymbol{A}$, provided that $\boldsymbol{u}$ is an eigenvector of $\boldsymbol{A}_1$. For the product, we find $\boldsymbol{A}_1\boldsymbol{u} = -\boldsymbol{u}_+$, with $\boldsymbol{u}_+ = \left(T^{(2)}, T^{(3)}, \ldots, T^{(L+1)}\right)^\top$. $\boldsymbol{A}\left(\boldsymbol{v}_{A,k}^{(1)} + \boldsymbol{v}_{B,2}\right) = -\lambda_k\left(\boldsymbol{v}_k, \boldsymbol{v}_k\right)^\top - M\left(\boldsymbol{u}_+, \boldsymbol{u}_+\right)^\top$ and this group of eigenvalues is therefore negative.

Finally, we obtain one important eigendirection showing an eigenvalue larger than zero. This direction corresponds to $\boldsymbol{v}_{B,1}$ and $\boldsymbol{v}_{A,k}^{(2)}$. Therefore, we make the following last ansatz $s^{(l)} = -\frac{r^{(l)}}{M-1}$ in order to reduce the system for a second time. Note that the exact same relation also holds for $D^{(l)} \neq 0$ and general $T^{(1)}$.

For the final form of the differential equations, we return to the case where $D^{(l)} \neq 0$ and $T^{(1)} \neq 1$, as the expressions are now more manageable to display and no longer excessively large. The final re-definition of the dynamical system yields

$$\frac{d\boldsymbol{r}}{d\alpha} = \eta g_r \tilde{\boldsymbol{U}} \boldsymbol{r}, \tag{122}$$

with $g_r = \frac{2}{\pi} \frac{\left(D^{(1)} + T^{(1)}\right)}{\sqrt{(M-1)T^{(1)} - D^{(1)} + M}\left(D^{(1)} + (M+1)T^{(1)} + M\right)^{\frac{3}{2}}}$, $\tilde{\boldsymbol{U}} = \tilde{\boldsymbol{u}}\boldsymbol{e}_2^\top$ and $\tilde{\boldsymbol{u}} = \left(T^{(1)} - \frac{D^{(1)}}{M-1}, T^{(2)} - \frac{D^{(2)}}{M-1}, \ldots, T^{(L)} - \frac{D^{(L)}}{M-1}\right)^\top$. Note that we define $\boldsymbol{A}_r = g_r\tilde{\boldsymbol{U}}$ for the main

text. Since $\tilde{U}$ is a rank-1 matrix, we obtain $L - 1$ zero eigenvalues and one eigenvalue

$$\lambda_r = T^{(2)} - \frac{D^{(2)}}{M - 1} \tag{123}$$

larger than zero. Thus, $\lambda_r$ drives the escape from the plateau. We can solve the differential equation directly and find for the first-order perturbation parameter

$$r^{(1)} = e^{\eta g_r \alpha} r_0^{(1)}, \tag{124}$$

where $r_0^{(1)} = r^{(1)}(\alpha_0)$ and $\alpha_0$ denotes an arbitrary time at the plateau. For the escape of the generalization error within our re-defined dynamical system, we find

$$\epsilon_g^* - \epsilon_g = \frac{\left((M - 1)T^{(1)} + M - D^{(1)}\right)\left(D^{(1)} + T^{(1)}\right)}{8\pi M (M - 1)\left(T^{(1)} - \frac{\left(D^{(1)} + T^{(1)}\right)^2}{4M^2}\right)^{\frac{3}{2}}} e^{\frac{\alpha}{\tau_{\text{esc}}}} r_0^{(1)2} \tag{125}$$

where we have introduced the escape time

$$\begin{aligned} \tau_{\text{esc}} &= \frac{1}{\eta g_r \lambda_r} \\ &= \frac{\pi}{2\eta} \frac{\sqrt{(M - 1)T^{(1)} - D^{(1)} + M}\left(D^{(1)} + (M + 1)T^{(1)} + M\right)^{\frac{3}{2}}}{\left(T^{(2)} - \frac{D^{(2)}}{M-1}\right)\left(D^{(1)} + T^{(1)}\right)}. \end{aligned} \tag{126}$$

Furthermore, we can approximate $T^{(2)} = \frac{1}{L}\operatorname{Tr}(\Sigma^2) = \frac{\lambda_+^2}{L}\sum_i^L \frac{1}{i^{2(1+\beta)}} \approx \frac{\lambda_+^2}{L} \propto L$ for large $L$. The same applies to $D^{(2)}$. For large $M$ and $L$, we find $\tau_{\text{esc}} \propto \frac{M^2}{\eta L}$.

Figure 5 presents the plateau scaling observed in simulation experiments, confirming the predicted relationship for the escape time $\tau_{\text{esc}} \propto \frac{M^2}{\eta L}$. In addition, Figure 19 highlights the plateau dependence under a different configuration.

## C.5 NUMERICALLY ESTIMATED PLATEAU LENGTH

In this subsection, we demonstrate how to combine the analytically derived formula from Eq. (12) with our calculated escape time (Eq. (15)) to estimate the plateau length. The plateau escape is described by the equation

$$\alpha_P - \alpha_0 = \tau_{\text{esc}}\left(D - \frac{1}{2}\ln\left(\sigma_J^2\right) + \frac{1}{2}\ln\left(N\right)\right), \tag{127}$$

where $D$ is a constant of order $\mathcal{O}(1)$, dependent on the variance at initialization and the plateau; $\alpha_0$ is an arbitrary starting point on the plateau; and $\tau_{\text{esc}}$ is the escape time from the plateau. To estimate the constant $D$, we interpret the results of Biehl et al. (1996) and find $D = \ln\left(\frac{B}{c}\right)$, where $B$ represents the deviation of the order parameter responsible for the plateau escape at $\alpha_P$ from its value at $\alpha_0$. Thereby, $c$ is a proportionality constant between the fluctuations at the plateau and at initialization. In our case, the order parameter that drives the plateau escape is the first-order student-teacher order parameter (cf. Subsection C.4). Thus, we define $B = |R^{(1)}(\alpha_0) - R^{(1)}(\alpha_P)|$. Additionally, to estimate $\alpha_P - \alpha_0$, we set $R^{(1)}(\alpha_P) = eR^{*(1)}$, following the definition of the escape time for the generalization error. Next, we estimate $\sigma_P = c\sigma_J$, where the plateau variance $\sigma_P$ is derived from numerical simulations of $R^{*(1)}$ at the plateau.

For the example shown in Figure 4, we set $\alpha_0 = 300$ and find $\sigma_P \approx 0.000279$, with $c \approx 0.0279$ (since $\sigma_J = 0.01$ is given), $B \approx 0.33$, and $D \approx 2.47$. For the escape time, we find $\tau_{\text{esc}} \approx 239$ by averaging the diagonal terms of $T_{nm}^{(1)}$ to obtain $T^{(1)}$ and using the averaged sum of the off-diagonal entries in order to estimate $D^{(1)}$ and $D^{(2)}$. Finally, we obtain $\alpha_P - \alpha_0 \approx 2751$.

This procedure provides valuable insight into how the plateau length behaves with respect to various parameters. Figure 20 shows the generalization error based on the solution of the differential equations and presents additional examples for the plateau length estimation.

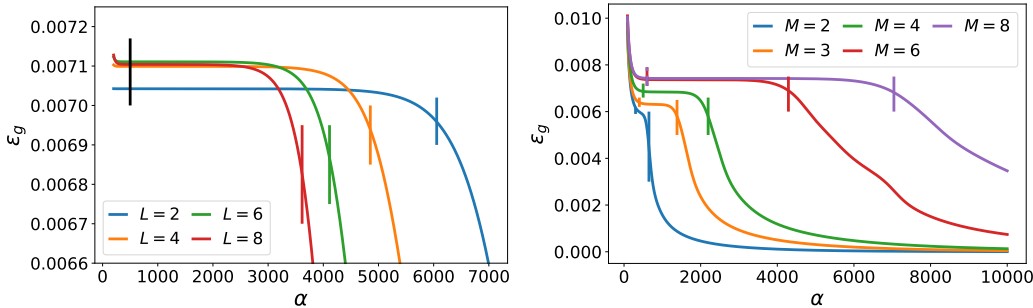

Figure 20: Plateau phase of the generalization error evaluated by numerical solutions of the differential equations for one random initialization (solid) and plateau length estimations given by Eq. (127) (verticle lines) for $N = 7000$, $K = M = 4$, $\sigma_J = 0.01$, $\beta = 0.25$ and $\eta = 0.1$. The black verticle line indicates the arbitrarily chosen plateau start $\alpha_0 = 500$, and the colored verticle lines show $\alpha_P$ for different $L$. We retain terms up to $\mathcal{O}(\eta)$ for the differential equations.

## C.6 ASYMPTOTIC SOLUTION

Here, we want to investigate how the generalization error converges to its asymptotic value in more detail. For this, we consider the typical teacher configuration $\langle T_{nm}^{(l)} \rangle = \delta_{nm} T^{(l)}$ since this configuration already captures the scaling behavior of the generalization error. For the asymptotic fixed points of the order parameters, we find $R_{im}^{(l)} = T_{im}^{(l)} \delta_{in}$ and $Q_{ij}^{(l)} = T_{ij}^{(l)} \delta_{ij}$. Here, we distinguish again between diagonal and off-diagonal entries for $R_{im}^{(l)} = R^{(l)} \delta_{im} + S^{(l)}(1 - \delta_{im})$ and $Q_{ij}^{(l)} = Q^{(l)} \delta_{ij} + C^{(l)}(1 - \delta_{ij})$ as for the plateau case. Furthermore, we linearized the dynamical equations for small perturbation around its fixed point $R^{(l)} = T^{(l)} + r^{(l)}$, $S^{(l)} = T^{(l)} + s^{(l)}$, $Q^{(l)} = T^{(l)} + q^{(l)}$, and $C^{(l)} = T^{(l)} + c^{(l)}$.
We find the following dynamic equations

$$\frac{d}{d\alpha} \begin{pmatrix} r \\ s \\ q \\ c \end{pmatrix} = \frac{2\sqrt{3}}{3\pi M} \boldsymbol{A}_a \begin{pmatrix} r \\ s \\ q \\ c \end{pmatrix} \tag{128}$$

with $r_i = r^{(i-1)}$, $s_i = s^{(i-1)}$, $q_i = q^{(i-1)}$, $c_i = c^{(i-1)}$ and $\boldsymbol{A}_a = \tilde{\boldsymbol{A}} + \tilde{\boldsymbol{B}} + \tilde{g}\tilde{\boldsymbol{C}}$ and $\tilde{g} = \eta \frac{2\sqrt{3}(\sqrt{45}+5(M-1))}{15\pi M}$. The individual matrices can be written as Kronecker products $\tilde{\boldsymbol{A}} = \boldsymbol{G} \otimes \boldsymbol{A}_1$, $\tilde{\boldsymbol{B}} = \boldsymbol{H} \otimes \boldsymbol{U}$, $\tilde{\boldsymbol{C}} = \boldsymbol{F} \otimes \boldsymbol{U}$ with

$$\boldsymbol{G} = \begin{bmatrix} 1 & \frac{\sqrt{3}}{2} & 0 & 0 \\ \frac{\sqrt{3}}{2} & 1 & 0 & 0 \\ -2 & -\sqrt{3} & 2 & \sqrt{3} \\ -\sqrt{3} & -2 & \sqrt{3} & 2 \end{bmatrix}, \quad \boldsymbol{H} = \begin{bmatrix} -\frac{1}{3} & -\frac{\sqrt{3}}{4} & \frac{1}{2} & \frac{\sqrt{3}}{4} \\ 0 & 0 & \frac{\sqrt{3}}{8} & 0 \\ -\frac{2}{3} & -\frac{\sqrt{3}}{2} & 1 & \frac{\sqrt{3}}{2} \\ 0 & 0 & \frac{\sqrt{3}}{4} & 0 \end{bmatrix}, \tag{129}$$

$$\boldsymbol{F} = \begin{bmatrix} 0 & 0 & 0 & 0 \\ 0 & 0 & 0 & 0 \\ -2 & -\frac{f_1 f_3}{\sqrt{3}} & 1 & \frac{f_1 f_3}{2\sqrt{3}} \\ -\frac{f_1 f_4}{\sqrt{3}} & -\frac{f_5 f_6}{2} & \frac{f_1 f_4}{2\sqrt{3}} & \frac{f_5 f_6}{4} \end{bmatrix} \tag{130}$$

where $f_1 = \sqrt{6} + M - 2$, $f_2 = \sqrt{45} + 5(M - 1)$, $f_3 = \frac{15(M-1)}{b}$, $f_4 = \frac{15}{b}$, $f_5 = 4(M-2)\sqrt{6} + 3M^2 - 15M + 26$, $f_6 = \frac{5}{b}$. Thereby, $\boldsymbol{A}_1$ and $\boldsymbol{U}_1$ are the same matrices as for the

linear activation function case. Therefore, the linearized version of the dynamical equation for the non-linear activation function resembles the dynamical equation for the linear activation. However, we encounter an additional "perturbation" by $\tilde{B}$, whereas $\tilde{C}$ describes the influence of higher-order terms in the learning learning rate. Moreover, compared to the linear case, the differential equation has more variables due to correlations between different student and teacher vectors. In order to analyze the behavior of the dynamical system, we need to determine the eigenvalues and eigenvectors of all sub-matrices. Here, we analyze the system for first order in the learning rate $\mathcal{O}(\eta)$ neglecting the contribution by $\tilde{g}\tilde{C}$.

The eigenvalues of $G$ are $\lambda_{G,1} = 2 - \sqrt{3}$ with eigenvector $v_{G,1} = (0,0,1,-1)^\top$, $\lambda_{G,2} = 2 + \sqrt{3}$ with eigenvector $v_{G,2} = (0,0,1,1)^\top$, $\lambda_{G,3} = 1 - \frac{1}{2}\sqrt{3}$ with eigenvector $v_{G,3} = (1,-1,2,-2)^\top$, $\lambda_{G,4} = 1 + \frac{1}{2}\sqrt{3}$ with eigenvector $v_{G,4} = (1,1,2,2)^\top$. The eigenvalues of $A_1$ were already studied in Subsection B.1 and are the negative eigenvalues of the data covariance matrix $-\lambda_k$ with eigenvectors $v_k$ summarized by the matrix $V$ (cf. Eq. (33)). Since $\tilde{A}$ is of block matrix structure expressed by a Kronecker product, we obtain for its spectrum $\lambda_{\tilde{A}} = \lambda_G \lambda_{A_1}$ and corresponding eigenvectors $v_{\tilde{A}} = v_G \otimes v_{A_1}$ for which we multiply each eigenvalue of $A_1$ with each of $G$. Thus, we obtain four different groups of eigenvalues for $\tilde{A}$ and in total $4L$ eigenvalues. The first group is obtained by multiplying the first eigenvalue of $G$ with all eigenvalues of $A_1$ leading to $\lambda_{\tilde{A},k}^{(1)} = -\left(2 - \sqrt{3}\right)\lambda_k$ with eigenvector $v_{\tilde{A},k}^{(1)} = (0,0,v_k,-v_k)^\top$. With the same procedure, we obtain for the other groups $\lambda_{\tilde{A},k}^{(2)} = -\left(2 + \sqrt{3}\right)\lambda_k$ with eigenvector $v_{\tilde{A},k}^{(2)} = (0,0,v_k,v_k)^\top$, $\lambda_{\tilde{A},k}^{(3)} = \left(1 - \frac{1}{2}\sqrt{3}\right)\lambda_k$ with eigenvector $v_{\tilde{A},k}^{(3)} = (v_k,-v_k,2v_k,-2v_k)^\top$, and $\lambda_{\tilde{A},k}^{(4)} = -\left(1 + \frac{1}{2}\sqrt{3}\right)\lambda_k$ with eigenvector $v_{\tilde{A},k}^{(4)} = (v_k,v_k,2v_k,2v_k)^\top$. The upper index for the eigenvalues and -vectors indicates the corresponding group.

The eigenvalues of $H$ are $\lambda_{H,1} = \frac{1}{3} - \frac{\sqrt{43}}{12}$ with eigenvector $v_{H,1} = \left(1, -\frac{\sqrt{3}}{9}\left(\sqrt{43}+4\right), 2, -\frac{2\sqrt{3}}{9}\left(\sqrt{43}+4\right)\right)^\top$, $\lambda_{H,2} = \frac{1}{3} + \frac{\sqrt{43}}{12}$ with eigenvector $v_{H,2} = \left(1, \frac{\sqrt{3}}{9}\left(\sqrt{43}+4\right), 2, \frac{2\sqrt{3}}{9}\left(\sqrt{43}+4\right)\right)^\top$, and $\lambda_{H,3} = 0$ with eigenvectors $v_{H,3} = \left(1,0,0,\frac{2\sqrt{3}}{9}\right)^\top$ and $v_{H,4} = (0,1,0,1)^\top$. For the matrix $U$, we have just one eigenvalue distinct from zero $\lambda_{U,1} = T^{(2)}$ with eigenvector $u$ since $U$ has rank one. The remaining eigenvectors are given by $v_{U,l} = e_l$ for $l \in (1,3,4,...,L)$ and especially $l \neq 2$. Thus, we obtain two different eigenvalues distinct from zero and $4L - 2$ zero eigenvalues for $\tilde{B}$. The eigenvalues distinct from zero are $\lambda_{\tilde{B},1} = \left(\frac{1}{3} - \frac{\sqrt{43}}{12}\right)T^{(2)}$ and $v_{\tilde{B},1} = \left(u, -\frac{\sqrt{3}}{9}\left(\sqrt{43}+4\right)u, 2u, -\frac{2\sqrt{3}}{9}\left(\sqrt{43}+4\right)u\right)^\top$ and $\lambda_{\tilde{B},2} = \left(\frac{1}{3} + \frac{\sqrt{43}}{12}\right)T^{(2)}$ and $v_{\tilde{B},2} = \left(u, \frac{\sqrt{3}}{9}\left(\sqrt{43}+4\right)u, 2u, \frac{2\sqrt{3}}{9}\left(\sqrt{43}+4\right)u\right)^\top$. Then, we have two eigenvectors with zero eigenvalue $v_{\tilde{B},3} = \left(u,0,0,\frac{2\sqrt{3}}{9}u\right)^\top$ and $v_{\tilde{B},4} = (0,u,0,u)^\top$. Further eigenvectors have the structure $e_l \otimes v_{H,1}$, $e_l \otimes v_{H,2}$, $e_l \otimes v_{H,3}$ and $e_l \otimes v_{H,4}$.

Strictly speaking, none of the eigenvectors of $\tilde{A}$ and $\tilde{B}$ are the same and we cannot calculate the spectrum of their sum directly. However, we notice that $\tilde{B}$ has just two eigenvalues distinct from zero and that its corresponding eigenvectors $v_{\tilde{B},1}$ and $v_{\tilde{B},2}$ are of the same structure as the last two groups of $\tilde{A}$ namely $v_{\tilde{A}}^{(3)}$ and $v_{\tilde{A}}^{(4)}$. For each of the vectors, the third and the fourth components are twice as large as the first and the second one. Therefore, only the eigenvalues of the last two groups of $\tilde{A}$ are influenced by adding the matrix $\tilde{B}$ leading to the eigenvectors of $A_a$ with the structure $v_{A_a,k}^{(3)} = \left(z_k^{(3)}, w_k^{(3)}, 2z_k^{(3)}, 2w_k^{(3)}\right)^\top$ and $v_{A_a,k}^{(4)} = \left(z_k^{(4)}, w_k^{(4)}, 2z_k^{(4)}, 2w_k^{(4)}\right)^\top$ with vectors $z_k^{(3)}$, $w_k^{(3)}$, $z_k^{(4)}$ and $w_k^{(4)}$ that has to be determined. Moreover, since the other eigenvalues of $\tilde{B}$ are zero, the eigenvalues of the first and second group $\lambda_{\tilde{A},k}^{(1)}$ and $\lambda_{\tilde{A},k}^{(2)}$ remain the same for $A_a$ as for $\tilde{A}$. However, the corresponding eigenvectors are no longer analytically determinable and we have to rely on numerical solutions. All these claims for the spectra and eigenvectors are in excellent agreement

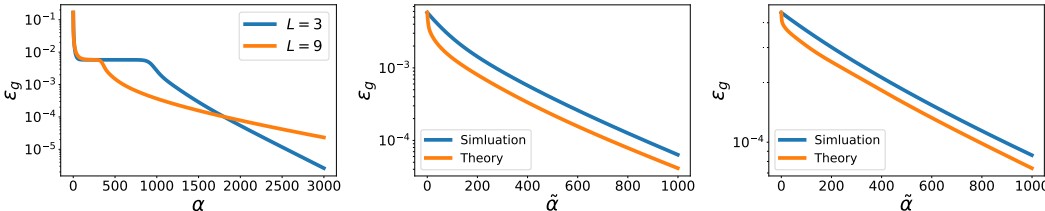

Figure 21: Left: Generalization error as a function of $\alpha$ obtained via the solution of differential equations of the order parameters up to $\mathcal{O}(\eta)$ and $K = M = 2$, $\eta = 0.25$, $\sigma_J = 0.01$, $N = 9000$ and $\beta = 1$. Center and Right: Comparison of the first-order Taylor expansion of the generalization error based on Eq. (131) for numerically obtained $r^{(1)}, s^{(1)}, q^{(1)}$ and $c^{(1)}$ (Simulation) with Eq. (132) (Theory) where the initial conditions $r^{(1)}(\alpha_0), s^{(1)}(\alpha_0), q^{(1)}(\alpha_0)$ and $c^{(1)}(\alpha_0)$ are obtained from simulations. Thereby, we choose $\alpha_0 = 1000$ and parameterize $\tilde{\alpha} = \alpha - \alpha_0$. Center: $L = 3$. Right: $L = 9$.

with numerical experiments.

Next, we Taylor expand the generalization error up to first order in the small perturbation parameters $r^{(1)}, s^{(1)}, q^{(1)}$ and $c^{(1)}$. We find

$$\epsilon_g = \frac{1}{6\pi} \left( 2\sqrt{3}q^{(1)} - 4\sqrt{3}r^{(1)} + 3(M-1)c^{(1)} - 6(M-1)s^{(1)} \right). \tag{131}$$

From this expansion, we observe that the eigen-directions $\boldsymbol{v}_{A_a,k}^{(3)}$ and $\boldsymbol{v}_{A_a,k}^{(4)}$ do not contribute to the generalization error in first-order since their components cancel out. After inserting the expressions for the first and second groups of eigenvectors, we obtain

$$\epsilon_g = \frac{1}{6\pi} \sum_{k=1}^{N} g_k^{(1)} e^{-a(2-\sqrt{3})\lambda_k \alpha} \left( 2\sqrt{3}v_{k,2L+2}^{(1)} - 4\sqrt{3}v_{k,2}^{(1)} + 3(M-1)v_{k,3L+2}^{(1)} - 6(M-1)v_{k,L+2}^{(1)} \right)$$
$$+ g_k^{(2)} e^{-a(2+\sqrt{3})\lambda_k \alpha} \left( 2\sqrt{3}v_{k,2L+2}^{(2)} - 4\sqrt{3}v_{k,2}^{(2)} + 3(M-1)v_{k,3L+2}^{(2)} - 6(M-1)v_{k,L+2}^{(2)} \right), \tag{132}$$

where $\boldsymbol{v}_k^{(1)}, \boldsymbol{v}_k^{(2)} \in \mathbb{R}^{4L}$ are the eigenvectors of $\boldsymbol{A}_a$ to the eigenvalues $-(2-\sqrt{3})\lambda_k$ and $-(2+\sqrt{3})\lambda_k$, respectively. Thereby, $g_k^{(1)} = \sum_l^{4L} \left( (\boldsymbol{V}^{(1)})^{-1} \right)_{kl} x_l$, $g_k^{(2)} = \sum_l^{4L} \left( (\boldsymbol{V}^{(2)})^{-1} \right)_{kl} x_l$ where $\boldsymbol{V}^{(1)}$ and $\boldsymbol{V}^{(2)}$ containing the first and second group of eigenvectors $\boldsymbol{v}_k^{(1)}$ and $\boldsymbol{v}_k^{(2)}$, respectively and $\boldsymbol{x} = (\boldsymbol{r}(\alpha_0), \boldsymbol{s}(\alpha_0), \boldsymbol{q}(\alpha_0), \boldsymbol{c}(\alpha_0))^\top \in \mathbb{R}^{4L}$ is some reference point at arbitrary chosen $\alpha_0$ in the asymptotic phase. The asymptotic convergence is governed by the smaller group of eigenvalues $(2-\sqrt{3})\lambda_k$.

Figure 21 compares the generalization error derived from the first-order Taylor expansion in Eq. (131) where we obtain the parameters $r, q, s, c$ by solutions of the differential equations and our theoretical results based on Eq. (132). For both, we use the same initial conditions, with $\alpha_0 = 1000$ as in the numerical solutions. The comparison shows excellent agreement. The small discrepancies between the graphs arise from the arbitrariness of $\alpha_0$ and the chosen initial conditions. Similar to linear activation functions, we observe a slowdown in convergence towards perfect generalization as $L$ increases, which eventually leads to a transition from exponential convergence to power-law scaling. We solve Eq. (132) using the Julia programming language; however, we encounter limitations in increasing $L$ due to constraints in numerical precision (see Section G).

Rather than solving the differential equations for the order parameters, we next solve the differential equations for the student weight vectors directly. As for the linear activation function, we approximate

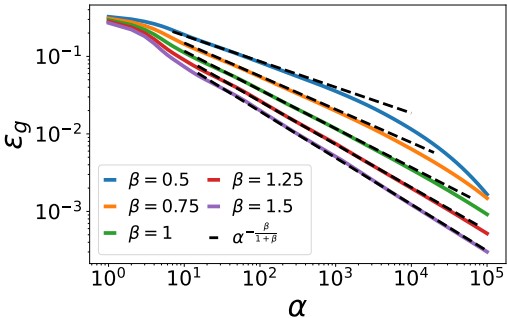

Figure 22: Generalization error as a function of $\alpha$ obtained via the solution of differential equations given in Eq. (133) for $K = M = 2$, $K = N = 1024$, $\eta = 0.001$, and $\sigma_J = 1$. We find the same scaling exponent as experiments and for the linear activation function $\alpha^{\frac{-\beta}{1+\beta}}$. We average over 50 seeds of student and teacher initializations.

$\frac{d\boldsymbol{J}_i}{d\alpha} \approx -\eta\nabla_J\epsilon_g$ and obtain

$$\frac{d\boldsymbol{J}_i}{d\alpha} = \frac{2\eta\boldsymbol{\Sigma}}{M\pi\left(1 + Q_{ii}^{(1)}\right)}\left[\sum_{m=1}^{M} \frac{\boldsymbol{B}_m\left(1 + Q_{ii}^{(1)}\right) - \boldsymbol{J}_i R_{im}^{(1)}}{\sqrt{\left(\left(1 + Q_{ii}^{(1)}\right)\left(1 + T_{mm}^{(1)}\right) - \left(R_{im}^{(1)}\right)^2\right)}}\right.$$

$$\left. - \sum_{j=1}^{M} \frac{\boldsymbol{J}_j\left(1 + Q_{ii}^{(1)}\right) - \boldsymbol{J}_i Q_{ij}^{(1)}}{\sqrt{\left(1 + Q_{ii}^{(1)}\right)\left(1 + Q_{jj}^{(1)}\right) - \left(Q_{ij}^{(1)}\right)^2}}\right]. \tag{133}$$

The results for the solution of Eq. (133) are given in Figure 22. We find a scaling exponent of $\epsilon_g \propto \alpha^{\frac{-\beta}{1+\beta}}$ consistent results obtained from simulations.

## D OVER-PARAMETRIZED SCENARIO $K > M$

Here we present numerical results for the over-parametrized scenario $K > M$.

The left panel of Figure 23 shows the scaling of the plateau length of $T^{(2)}$. As for the realizable scenario, we find a linear dependency on the inverse of $T^{(2)}$ as predicted by Eq. (15). Note that $T^{(2)} \propto L$ for large $L$.

The right panel of Figure 23 shows simulation results for a student-teacher setup for $K > M$, where we train a student network based on the outputs of a teacher and synthetic input data with a power-law covariance matrix. We obtain that the scaling law in the asymptotic phase is not altered by over-parametrization. Moreover, increasing the learning rate with the student size $K$ for a fixed $M$ leads to curve collapse, as shown in the inset. Thereby, we use the same initial student and teacher weights as before and choose the learning rate such that the ratio $\frac{\eta}{K}$ is constant. For this example, we choose $\eta_{K=2} = 0.005$, $\eta_{K=4} = 0.01$ and $\eta_{K=8} = 0.02$.

The phenomenon of curve collapse can be understood as follows: for a constant teacher size $M$, an increase in the student size $K$ results in a scaling of the student output, which is proportional to $\frac{\sqrt{M}}{K}$ (cf Eq. (2)). Given that the output weights remain constant, this scaling reduces the overall learning rate.

For the over-parametrized scenario, (Richert et al., 2022) demonstrated that the asymptotic generalization error obeys a power-law scaling $\epsilon_g \propto \alpha^{-2}$ even for isotropic input data. This can be characterized as an "intrinsic" power-law since it depends on the architectural configuration. We find that the asymptotic power-law $\alpha^{-\frac{\beta}{\beta+1}}$ is independent of $K \geq M$, meaning that the power-law determined by the data covariance matrix is not altered by the intrinsic power-law $\alpha^{-2}$. This likely occurs because the power law exponent is consistently smaller than the intrinsic power law exponent $\frac{\beta}{\beta+1} < 2$.

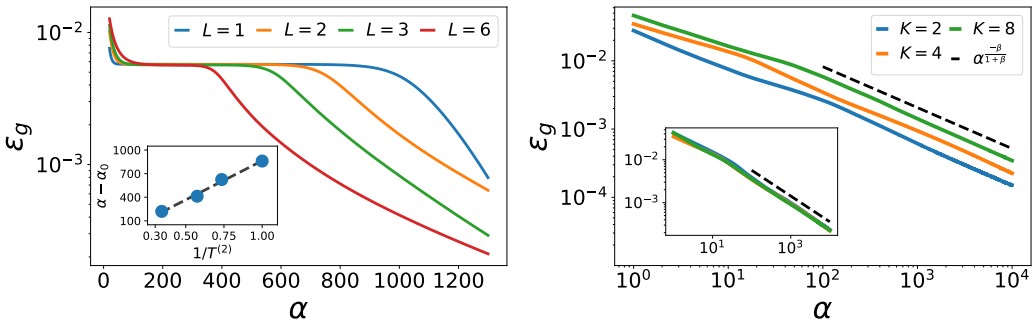

Figure 23: Left: Generalization error as a function of $\alpha$ for the over-parametrized case obtained via the solution of differential equations of the order parameters up to $\mathcal{O}(\eta)$ for $K = 4$, $M = 2$, $\eta = 0.1$, $\sigma_J = 0.01$, $N = 10^4$ and $\beta = 1$. The inset shows the measured scaling of the plateau length with the inverse of $T^{(2)}$ for this single initialization, where we find a linear dependency. Right: Generalization error as a function of $\alpha$ for the over-parametrized case obtained for the training of a student-teacher setup for $\eta = 0.01$, $\sigma_J = 0.01$, $N = L = 1024$ and $\beta = 1.5$. The inset shows the curve collapse for the scaling of the learning rate with $K$. We choose the learning rate such that the ratio $\frac{\eta}{K}$ is constant. For this example, we choose $\eta_{K=2} = 0.005$, $\eta_{K=4} = 0.01$ and $\eta_{K=8} = 0.02$. We average the generalization error over ten random seeds.

## E   ReLU activation function

For the ReLU activation function, (Straat & Biehl, 2019) derived an expression for the generalization error analytically and analyzed the learning dynamics for isotropic input samples. The analytical expression for the generalization error for correlated inputs becomes

$$
\epsilon_{g,\text{ReLU}} = \sum_{i,j}^{M} \frac{1}{2M} \left[ \frac{Q_{ij}^{(1)}}{4} + \frac{1}{2\pi} \left[ \sqrt{Q_{ii}^{(1)} Q_{jj}^{(1)} - \left(Q_{ij}^{(1)}\right)^2} + Q_{ij} \arcsin\left( \frac{Q_{ij}^{(1)}}{\sqrt{Q_{ii}^{(1)} Q_{jj}^{(1)}}} \right) \right] \right]
$$
$$
- \sum_{i,n}^{M} \frac{1}{M} \left[ \frac{R_{in}^{(1)}}{4} + \frac{1}{2\pi} \left[ \sqrt{Q_{ii}^{(1)} T_{nn}^{(1)} - \left(R_{in}^{(1)}\right)^2} + R_{in}^{(1)} \arcsin\left( \frac{R_{in}^{(1)}}{\sqrt{Q_{ii}^{(1)} T_{nn}^{(1)}}} \right) \right] \right]
$$
$$
+ \sum_{n,m}^{M} \frac{1}{2M} \left[ \frac{T_{nm}^{(1)}}{4} + \frac{1}{2\pi} \left[ \sqrt{T_{nn}^{(1)} T_{mm}^{(1)} - \left(T_{nm}^{(1)}\right)^2} + T_{nm} \arcsin\left( \frac{T_{nm}^{(1)}}{\sqrt{T_{nn}^{(1)} T_{mm}^{(1)}}} \right) \right] \right],
$$
$$(134)$$

again for the soft committee machine in the realizable case $K = M$. In the following, we solve the differential equations for the student weights rather than for the order parameters numerically. As for the error function and linear activation function, we approximate $\frac{d\boldsymbol{J}_i}{d\alpha} \approx -\eta \nabla_J \epsilon_{g,\text{ReLU}}$ and find

$$
\frac{d\boldsymbol{J}_i}{d\alpha} \approx \frac{\eta}{M} \boldsymbol{\Sigma} \sum_{m=1}^{M} \left( \frac{\boldsymbol{J}_i \sqrt{Q_{ii}^{(1)} T_{mm}^{(1)} - \left(R_{im}^{(1)}\right)^2}}{2\pi Q_{ii}^{(1)}} + \frac{\boldsymbol{B}_m}{2\pi} \arcsin\left( \frac{R_{im}^{(1)}}{\sqrt{Q_{ii}^{(1)} T_{mm}^{(1)}}} \right) + \frac{\boldsymbol{B}_m}{4} \right)
$$
$$
- \sum_{j=1}^{K} \left( \frac{\boldsymbol{J}_i \sqrt{Q_{ii}^{(1)} Q_{jj}^{(1)} - \left(Q_{ij}^{(1)}\right)^2}}{2\pi Q_{ii}^{(1)}} + \frac{\boldsymbol{J}_j}{2\pi} \arcsin\left( \frac{Q_{ij}^{(1)}}{\sqrt{Q_{ii}^{(1)} Q_{jj}^{(1)}}} \right) + \frac{\boldsymbol{J}_j}{4} \right), \tag{135}
$$

and solve this equation numerically. The top left panel of Figure 24 shows the generalization error for the ReLU activation function for different $L$. As for the error function, the plateau length decreases as the number of distinct eigenvalues $L$ and the plateau length is inversely proportional to $T^{(2)} \propto L$. However, a formal proof has to be provided.

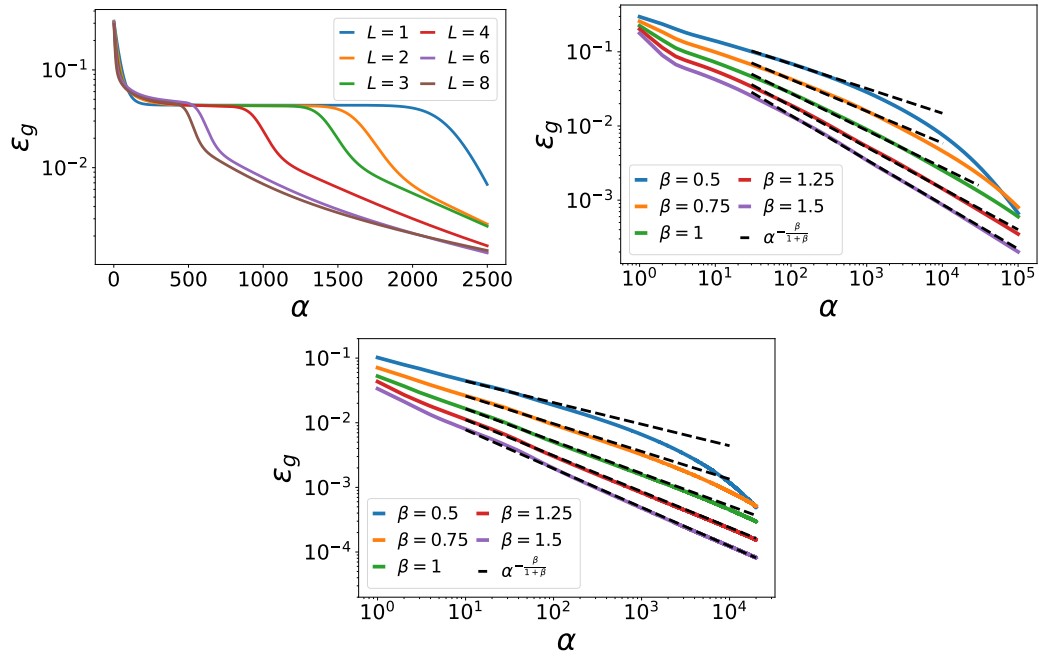

Figure 24: Generalization error as a function of $\alpha$ for the ReLU activation function obtained via the solution of differential equations given in Eq. (135) (top) and experiments (bottom) for $K = M = 2$. Top Left: Plateau behavior for $\eta = 0.01$, $\sigma_J = 10^{-6}$, $N = 10^4$ and $\beta = 1$. Top Right: Asymptotic behavior of $\epsilon_g$ for $K = N = 1024$, $\eta = 0.001$, and $\sigma_J = 1$. We find the same scaling exponent as for the error function $\alpha^{-\frac{\beta}{1+\beta}}$. We average over 25 seeds of student and teacher initializations. Bottom: We train a student based on the outputs of a teacher network and synthetic input data for $\eta = 0.01$, $\sigma_J = 10^{-2}$, $N = L = 2048$. We obtain the same scaling exponent as for the error function $\alpha^{-\frac{\beta}{1+\beta}}$.

Moreover, we observe a slow-down in the convergence of the asymptotic generalization error. The top right panel of Figure 24 shows the solution of the differential equations for $L = N$. We find the same scaling exponent as for the error function $\alpha^{-\frac{\beta}{1+\beta}}$. These numerical findings are supported by simulations. The bottom panel of Figure 24 illustrates the results for training a student network by a teacher and synthetic correlated input data with a covariance matrix possessing a power-law spectra. For numerical simplicity, we solve the differential equations for diagonal covariance matrices. Figure 27 presents simulation experiments where a student is trained on the outputs of a teacher using synthetic input data for various activation functions, with $K = M = 2$. For all setups, we obtain the same scaling exponent for the asymptotic phase as for the error function $\alpha^{-\frac{\beta}{1+\beta}}$. However, the time window for which the scaling law is achieved varies across different activation functions.

## F    GENERALIZED PERCEPTRON

In this section, we aim to investigate the numerical implications of training both layers of the student model on the scaling of the asymptotic generalization error. To this end, we examine a fully trained two-layer student model, where $K = 1$, which we refer to as the generalized perceptron. The teacher model in this scenario is a single-layer perceptron. Our analysis will concentrate on the error function and ReLU activation.
We now redefine the student output to

$$\sigma(\boldsymbol{J}, \boldsymbol{\xi}) = cg(x), \qquad\qquad \zeta(\boldsymbol{B}, \boldsymbol{\xi}) = g(y), \qquad\qquad (136)$$

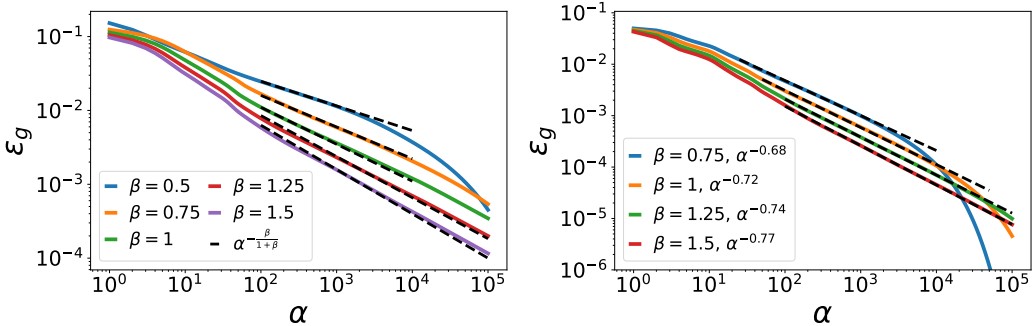

Figure 25: Generalization error as a function of $\alpha$ for the generalized perceptron for $K = N = 1024$, $\eta = 0.001$, and $\sigma_J = 0.1$. We average over 50 seeds of student and teacher initializations. Left: Solution for the error function activation obtained via Eq. (137). We choose the initial output weight from the normal distribution $c_0 \sim \mathcal{N}(0, 0.1)$. We find the same scaling exponent as for the soft committee machine $\alpha^{-\frac{\beta}{1+\beta}}$. Right: Solution for the ReLU activation function obtained via Eq. (138). We initialize the output weight by $c_0 = 0.5$. The scaling exponent differs from that of the soft committee machine.

where $c$ is now an additional trainable parameter. Next, we approximate the dynamics of the weights by $\frac{d\boldsymbol{J}}{d\alpha} \approx -\eta \nabla_J \epsilon_g$ and $\frac{dc}{d\alpha} \approx -\eta \nabla_c \epsilon_g$ and obtain for the error function

$$\frac{\mathrm{d}\boldsymbol{J}}{\mathrm{d}\alpha} \underset{\eta \to 0}{\approx} -\frac{2\eta c}{\pi \left(1 + Q^{(1)}\right)} \boldsymbol{\Sigma} \left[ \frac{R^{(1)} \boldsymbol{J} - \boldsymbol{B}\left(1 + Q^{(1)}\right)}{\sqrt{-\left(R^{(1)}\right)^2 + (1 + T^{(1)})(1 + Q^{(1)})}} + \frac{\boldsymbol{J}}{\sqrt{2Q^{(1)} + 1}} \right]$$

$$\frac{\mathrm{d}c}{\mathrm{d}\alpha} \underset{\eta \to 0}{\approx} -\frac{2\eta N}{\pi} \left[ c \arcsin\left( \frac{Q^{(1)}}{1 + Q^{(1)}} \right) - \arcsin\left( \frac{R^{(1)}}{\sqrt{(1 + Q^{(1)})(1 + T^{(1)})}} \right) \right] \tag{137}$$

and ReLU activation function

$$\frac{\mathrm{d}\boldsymbol{J}}{\mathrm{d}\alpha} \underset{\eta \to 0}{\approx} -\eta c \boldsymbol{\Sigma} \left[ \frac{\boldsymbol{J}c}{2} - \frac{\boldsymbol{B}}{4} - \frac{\boldsymbol{B}}{2\pi} \arcsin\left( \frac{R^{(1)}}{\sqrt{Q^{(1)}}} \right) \right.$$

$$\left. - \frac{1}{2\pi} \left( \frac{T^{(1)} \boldsymbol{J} - R^{(1)} \boldsymbol{B}}{\sqrt{Q^{(1)} T^{(1)} - \left(R^{(1)}\right)^2}} + \frac{R^{(1)}}{\sqrt{Q^{(1)} T^{(1)} - \left(R^{(1)}\right)^2}} \frac{Q^{(1)} \boldsymbol{B} - R^{(1)} \boldsymbol{J}}{Q^{(1)}} \right) \right],$$

$$\frac{\mathrm{d}c}{\mathrm{d}\alpha} \underset{\eta \to 0}{\approx} -\eta N \left[ c \frac{Q^{(1)}}{2} - \left( \frac{R^{(1)}}{4} + \frac{1}{2\pi} \left[ \sqrt{Q^{(1)} T^{(1)} - \left(R^{(1)}\right)^2} + R^{(1)} \arcsin\left( \frac{R^{(1)}}{\sqrt{Q^{(1)} T^{(1)}}} \right) \right] \right) \right]. \tag{138}$$

Note that the hidden-to-output weight $c$ operates on a different time scale compared to the student vector, primarily due to the additional factor $N$. Figure 25 illustrates the solutions to the ordinary differential equations (ODEs) under consideration. Our analysis reveals that, for the error function activation, there is no significant deviation from the predicted scaling law. Conversely, for the ReLU activation function, we observe an improvement in the scaling law when training the second layer weights.

One potential reason for the improved scaling could be the shifting of fixed points for ReLU activation, which is missing in the error function case. From Equation (137), we identify the following fixed point for the error function:

$$\boldsymbol{J}^* = \boldsymbol{B} \quad \text{and} \quad c^* = 1, \tag{139}$$

which corresponds to the same weight configuration as that of a perceptron student. Furthermore, in more complex architectures where $K = M > 1$, the fixed point equation continues to yield the soft committee machine with $c_i^* = 1$ and $J_i^* = B_i$ for $K = M$ and $i = 1, .., K$ when the teacher output

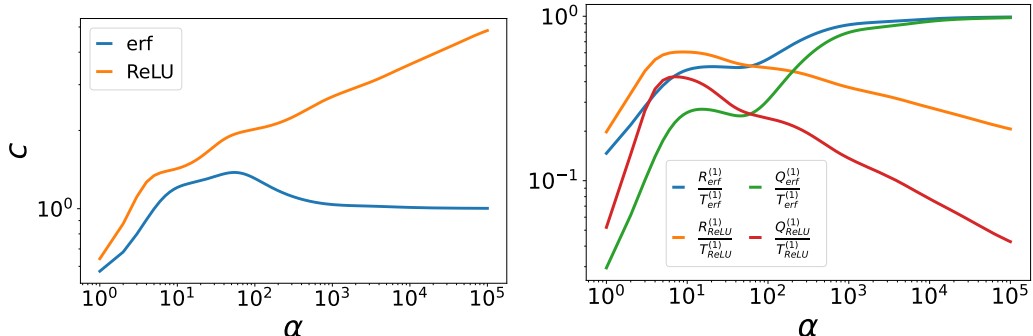

Figure 26: Output weight and first-order order parameters as a function of $\alpha$ for both error function and ReLu activation based on the solution of Eq. (137) and Eq. (138), respectively. We show the solution for one random initialization for $\eta = 1$, $N = L = 1024$ and $\beta = 1$. Left: Output weight $c$ with initial value $c_0 = 0.5$. Right: Rescaled first-order order parameters based on initial student vector coming from the normal distribution with standard deviation $\sigma_J = 0.1$.

weights are set to one. However, for the ReLU activation, the fixed point that corresponds to the soft committee machine is no longer applicable and becomes shifted as shown in Figure 26. The precise value of this shifted fixed point necessitates numerical evaluation of the implicit equations

$$\boldsymbol{J}^* = \frac{\boldsymbol{B}}{c^*} \quad \text{and} \quad c^* = \frac{\left(R^{(1)}\right)^*}{\left(Q^{(1)}\right)^*}, \tag{140}$$

with $c^* \neq 1$ in general. As a result, the student network can establish a new weight configuration for the ReLU activation function more rapidly than it retrieves the "old" weight configuration for the error function that corresponds to the soft committee machine.

## G    REMARKS ON NUMERICAL SOLUTIONS

Evaluating a large number of distinct eigenvalues becomes computationally challenging as $L$ increases. The expectation value of the teacher-teacher order parameters is given by $\left\langle T_{nm}^{(l)} \right\rangle = \delta_{nm} \frac{1}{N} \operatorname{Tr}(\Sigma^l)$. In this context, the highest order trace term in the differential equations is $L - 1$, and for large $L$, we can approximate $\operatorname{Tr}(\Sigma^l)$ as $\lambda_+^{L-1} \sum_{i=1}^{L} \frac{1}{i^{(1+\beta)(L-1)}} \approx (\lambda_+)^{L-1}$. Since $\lambda_+$ scales with $L$ for large values of $L$, the expectation values increase in a 'super-exponential' manner with $L$. This growth also applies to the standard deviation of the off-diagonal entries of $T_{nm}^{(L-1)}$, further complicating numerical evaluations as $L$ grows.

As a result, numerical investigations are restricted to small values of $L$. This limitation applies to solving the differential equations, evaluating fixed points, and analyzing the generalization error in the asymptotic phase. For instance, to evaluate Eq. (17) and generate Figures 10 through 13, we utilized Julia, a high-level scripting language, with arbitrary precision arithmetic.

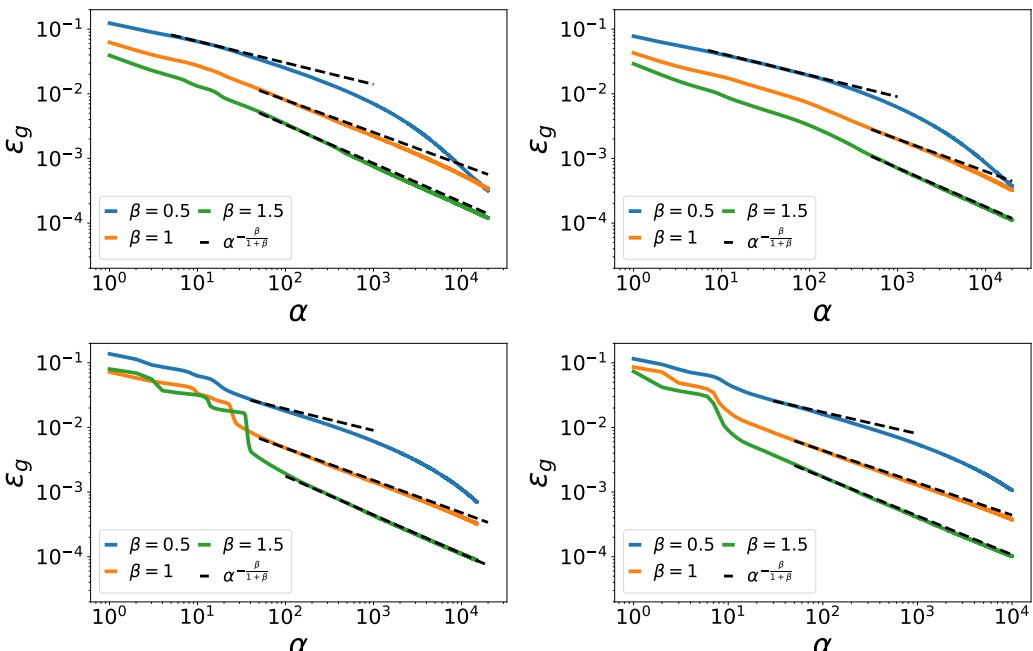

Figure 27: Generalization error as a function of $\alpha$ for different acitvation functions based on simulations for $K = M = 2$. We train a student based on the outputs of a teacher network and synthetic input data for $\eta = 0.01$, $\sigma_J = 10^{-2}$, $N = L$. For all setups, we obtain the same scaling exponent as for the error function $\alpha^{-\frac{\beta}{1+\beta}}$. We average over ten different student and teacher initializations. Upper left: elu activation with $N = L = 1024$. Upper right: tanh activation $N = L = 1024$. Bottom left: gelu activation with $N = L = 2048$. Bottom right: swish activation with $N = L = 2048$.

