# OpenReview forum: "Analyzing Neural Scaling Laws in Two-Layer Networks with Power-Law Data Spectra"
_ICLR.cc/2025/Conference — ICLR 2025 Spotlight_

### Official Review · Reviewer_5FqQ · 2024-10-22

**Soundness:** 3
**Presentation:** 3
**Contribution:** 2
**Rating:** 6
**Confidence:** 3

**Summary:**

The authors analyze the dynamics of learning a two-layer neural network with single-pass SGD, in the limit of large data dimension, for data with a power-law covariance spectrum. For linear regression, they derive an expression for the rate of decay of the error with time, which matches previous results in the literature in related settings. In non-linear cases, they analytically determine the length of the plateau in the learning dynamics, and in particular how it depends on the number of distinct eigenvalues in the spectrum.

**Strengths:**

The paper is very well written, results are clearly exposed and connected to related works. Abundant and clear numerical experiments are provided to support the main results.

The question explored is interesting. To the best of my reading, the main technical contributions are
- Establishing the rate of decay of linear regression for one-pass SGD, which happens to match previous full-batch results, e.g. Bordelon and Pehlevan (2022), Bahri et al (2024). The model and training are much simpler than in those papers, but I believe this particular case has not yet been covered in the literature, although I am not completely familiar with it.
- Generalizing the escape time analysis of Biehl (1996) to structured data, again to the best of my understanding of the literature. In particular, the result that the length of the plateau decreases with the number of eigenvalues is an interesting one.

I have a number of concerns, which I detail in the following sections. Overall, I think the paper is sound, although I have not checked the derivations in detail, and am overall in favor of acceptance if my concerns are addressed by the authors.

**Weaknesses:**

I have a number of observations and questions. I regroup my main concerns in this section, and leave the more minor points for the next.

- l.191 I have strong doubts about this statement. I believe preactivations are always Gaussian, as linear combination of the Gaussian inputs. The Gaussian Equivalence principle is needed when discussing the post-activations. Please correct me if I'm wrong.

- l. 417. Is a $l$ missing somewhere, why does the variance not depend on it ? Furthermore, it seems to me the variance scales as $1/N$, which does imply self-averaging. I do not understand the author's claim that higher-order overlaps do not self-average, which is linked to the result that $M$ distinct plateaus are present for the dynamics of e.g. the $R$ overlap. Further clarification would be very helpful.

**Questions:**

- l.190 Is a squareroot missing, i.e. should it be $\xi^\mu (\sigma)^{l/2}J_i$ ? It does not seem consistent with l. 194 otherwise.

- l.223 Shouldn't each term in parenthesis in the characteristic polynomial be to the power $N/L$ ? I might be missing something, further clarification would be helpful.

Minor comments and recommendation.

- Plotting the predicted slope for the power law (l.314) and the associated window of validity in Fig. 2 (right) would be a very compelling illustration of the theory.

- I believe the scaling of $\tau_{esc}\sim M^2/L$  to be one of the most interesting results of the work. I would be curious to see a plot where the escape time is experimentally measured, alongside this predicted rate, to illustrate and support this finding, but this is just a recommendation.

- In the continuation of the last point, could the authors provide more intuition why the length of the plateau decreases (increases) with $L$ ($M$)?

---

> ### Author Response · Authors · 2024-11-23
>
> We thank the reviewer for their thorough evaluation of our manuscript and their constructive comments. We appreciate the positive feedback on the clarity of our writing and the soundness of our results. We address the reviewer's concerns below.
>
> i) Regarding the statement on line 191, we agree with the reviewer that the preactivations are indeed Gaussian because they are linear combinations of Gaussian inputs. The Gaussian equivalence principle is only necessary when discussing the post-activations, not the preactivations. We have corrected this in the manuscript by removing the reference to the Gaussian equivalence principle in that context.
>
> ii) Concerning line 417, we acknowledge the missing factor of $ l $ in the exponent of the eigenvalues in the expression for the variance of $T_{nm}^{(l)} $. We have corrected this mistake in the revised manuscript. To compute the scaling of the variance with $ N $, we note that $ \lambda_l = \lambda_+ / l^{1+ \beta} $, and the convergent sum $\sum_{l=1}^N 1/l^{2 l (1 + \beta)} $ is of order one, not of order $N $. Therefore, the variance scales as $ \lambda_+^{2 l} / N^2 $, and the expectation value scales as $ \lambda_+^{l} / N $. This implies that the ratio of the standard deviation to the expectation value remains of order one, indicating that higher-order overlaps do not self-average. This lack of self-averaging leads to the presence of $ M $ distinct plateaus in the dynamics of quantities like $ R_{in}^{(l)} $, as observed in our results.
>
> Questions:
>
> i) Regarding the discrepancy in line 190, we thank the reviewer for pointing this out. We have corrected the expression for the covariances now in l.189 such that they are consistent with the definition of preactivations (which were correct).
>
> ii) For line 223, we appreciate the reviewer's attention to detail. The reviewer is correct that the expression refers to the minimal polynomial, not the characteristic polynomial. We have updated the terminology in the text to ensure clarity.
>
> Minor comment:
>
> i) We agree that plotting the predicted slope for the power law and indicating the associated window of validity would enhance the illustration of our theory. We have updated Figure 2 (right panel) to include the predicted slope and have extended the analysis in Figure 9 in the appendix to test additional configurations. This also allowed us to improve the upper bound, making it tighter and more consistent with the observed behavior.
>
> ii) We concur that demonstrating the scaling of the escape time is valuable. In the left panel of Figure 5, we now show the linear dependence of the plateau length on $M^2 $, and in the right panel, we illustrate the $ 1/L $ dependence. This provides experimental support for our theoretical findings regarding the scaling of the escape time $ \tau_{\text{esc}} $.
>
> iii) Regarding the intuition behind why the length of the plateau decreases with $ L $ and increases with $ M $, we offer the following explanation. The network output scales with $ 1/\sqrt{M} $ due to the averaging effect of the $ M $ hidden units. Consequently, the generalization error and the gradient scale with $ 1/M $. Learning $M $ features is inherently more challenging than learning a single feature, which contributes an additional factor of $ M $ to the plateau length, leading to the $ M^2$ dependence. As for the $ 1/L $ dependence, normalizing the average covariance to one means that the largest eigenvalue of the input covariance matrix scales with $ L $. Directions associated with larger eigenvalues are learned more rapidly, causing the departure from the plateau to occur sooner and shortening the plateau length as $ L $ increases.
>
> We hope that these clarifications and additions address the reviewer's concerns. We have updated the manuscript accordingly and are grateful for the opportunity to improve our work based on the insightful feedback.

---

> > ### Comment · Reviewer_5FqQ · 2024-11-25
> > **Acknowlegement of rebuttal**
> >
> > I thank the authors for the many clarifications and the improvement they made to the manuscript.
> >
> > I would still need further clarification on one technical point, although it is possible it is a misunderstanding on my part. Regarding the variance and mean of e.g. $T^l_{nm}$,  I seem to understand from (1) and the discussion around it that
> >  $T^l_{nm}=\delta_{nm}\frac{1}{N}\sum_{k=1}^L \frac{N}{L}(\frac{\lambda_+}{k^{1+\beta}})^l=O(1)$ , and not $O(1/N)$ (if I understand correctly that $L=O_N(1)$ as seems to be the case in Fig. 4) ?

---

> > > ### Author Response · Authors · 2024-11-26
> > > **Response to Technical Question**
> > >
> > > We thank the reviewer for acknowledging the improvements in our manuscript. We appreciate the opportunity to address the technical question regarding the variance and mean of $T^{(l)}_{nm}$.
> > >
> > >
> > > The reviewer is correct when pointing out that for a small ratio $L/N$ (or more precisely for $L/N \to 0$ in the thermodynamic limit $N \to \infty$) the $T^{(l)}_{nn} $ are of order $ O(1) $, while their variance is of order $ O(1/N) $, as relevant for the plots in Fig. 4. In our answer to the original question of the reviewer, we had the scenario of a finite ratio $L/N$ in mind, for which the arguments we made are applicable. For finite $L/N$ in the thermodynamic limit, the teacher-teacher order parameters are not self-averaging for $l > 0$. We have added a clarification to the manuscript above Eq. (13), and have added the new appendix C.2 with a calculation of expectation value and variance in the two scenarios.
> > >
> > > We hope that these clarifications and additions resolve the reviewer's concerns and contribute to a clearer understanding of our work. We are happy to provide further clarifications or make additional adjustments during the ongoing discussion period.

---

> > > > ### Comment · Reviewer_5FqQ · 2024-11-26
> > > > **Acknowledgement of response**
> > > >
> > > > I thank the authors very much for their reply and clarification. The added appendix answers my concerns. I wish to maintain my evaluation, and a moderate confidence score due to me not having carefully checked all derivations, but would like to clarify for the AC that my score is a strong 6/ 7, and that I am in favor of acceptance.

---

> ### Author Response · Authors · 2024-11-26
> **Appreciation for Valuable Feedback**
>
> Dear Reviewer 5FqQ,
>
> Thank you very much for your kind feedback and for taking the time to review our responses. We are pleased that the additional appendix addressed your concerns and appreciate your thoughtful evaluation to help us improve the quality of our work.
>
> Best regards,
>
> The Authors

---

### Official Review · Reviewer_WSqh · 2024-11-03

**Soundness:** 4
**Presentation:** 4
**Contribution:** 2
**Rating:** 8
**Confidence:** 3

**Summary:**

In this paper, the authors investigate neural scaling laws in a two-layer student-teacher network, where the data spectrum is generated to have L distinct eigenvalues following a power-law distribution. The authors use one-pass stochastic gradient descent with MSE loss. First, the authors derive analytical expressions for the generalization error in the case of a linear activation function and establish a condition for power-law scaling. Then, for non-linear activation functions, where plateaus may emerge, they derive expressions to predict both the height and width of these plateaus and investigate the asymptotic solution. For large $L$ and the number of hidden units $M$, the width exhibits an elegant scaling law of $\sim M^{2}/L$

**Strengths:**

The paper is well-written overall. The arguments are well-founded, the text is concise and clear, and the authors skillfully focus on essential points in the main text, leaving detailed calculations for the appendix. All of the derived analytical results are verified by numerical simulations. The contribution, particularly the investigation of the properties of the plateau, appears to be quite solid as well.

**Weaknesses:**

The main problem with the manuscript is that the setting is quite narrow - exact degeneracy of the covariance eigenvalue, 2nd layer neurons are all untrainable and *identical*, only one nonlinear activation tested, etc. It is not clear how robust these results are to to even slight variation of the setting. IMHO, This is the main point that would determine the impact of this work.

Clearly, it is very difficult to extend the analytical results beyond the setting described by the authors, but it would strengthen the manuscript considerably if the authors investigated numerically the robustness of their results. Since the networks are relatively small, this should be easy to do. Concretely, to what extent do the conclusions hold when:
  - $K\neq M$?
 - The weights of the 2nd layer are not identical?
- The covariance is only almost degenerate?
- Other (and more common) nonlinear activations are used?

(even a subset of these would be useful, but as I wrote above, the experiments are not challenging)

**Questions:**

- In Sec. 4.3 the authors model some learning behavior by training only $N_{l}$ of the student vector. Could the authors clarify the intended learning behavior they are simulating? Specifically, what motivates the use of $N_l$ in this context? Could you clarify the content of the statement "each entry of the student
vector directly corresponds to an eigenvalue"? This should be clear to the reader without referring to the appendices.

- Fig. 5: To enable a quantitative comparison with the derived expressions, could the authors include the width of the plateau (Eqs. (12) and (15)) in the figure?

- Can the authors give more intuition about the underlying cause for the existence of the plateau, and for the scaling law of the escape time? It is a bit hard to infer it from the analytical derivation, and a heuristic/intuitive/hand-wavy explanation for this it would be helpful.

- There is a bit of missing discussion regarding the generalization of the results. Are they specific to this particular setup, or do we expect some of these results to apply in other scenarios? For instance, how would the findings change i?

- Line 209: To avoid confusion, consider renaming $\alpha=p/N$, since it is already used as the time in the rest of the paper.

- Around line 222: Have the authors mistaken the minimal polynomial with the characteristic polynomial?

---

> ### Author Response · Authors · 2024-11-23
>
> We thank the reviewer for their thorough evaluation of our manuscript and for their constructive comments. The reviewer is mainly concerned about the generality of the findings in our work and asks about possible generalizations.
>
> i) Regarding the case when  $K \neq M  $: The scenario where $ K > M  $ is interesting because, even for isotropic input patterns, the generalization error obeys a scaling law $ \epsilon_g \propto \alpha^{-2}  $ in the asymptotic regime of large  $ \alpha  $(see Phys. Rev. E L052302 (2022)). We have added an analysis of this case to the appendix, based on a solution to differential equations for the dynamics of the student vector. We find that the asymptotic power law is independent of  $K  $; that is, the power law determined by the data covariance matrix is not modified due to the intrinsic power-law learning dynamics of the model. This is presumably because the power-law exponent  $\beta / (\beta + 1)  $ is always smaller than the intrinsic power-law exponent. The results are provided in Appendix D.
>
> On the other hand, the case $ K < M $ would be a generalization of the situation where the student cannot fully learn the teacher weights due to a lack of parameters. For every $ K < M $, the generalization error would reach an asymptotic plateau value, and we expect a scaling $\epsilon_g \propto K^{-\gamma} $. Since this additional scaling law does not affect the generality of our results, we have not performed additional analysis for this rebuttal.
>
> ii) Concerning the scenario where the student weights are not identical but equal to the teacher weights, we expect the scaling law to be unchanged. When the student weights are not identical and differ from the teacher weights, we anticipate that perfect learning is no longer possible. Hence, the generalization error reaches an asymptotic plateau beyond which there is no power-law scaling anymore.
>
> Perhaps the most interesting generalization is when the student weights in the second layer are trained. In this situation, one finds that the exponent for learning does not change for the error function activation; it remains unchanged at $\beta / (\beta + 1) $. However, for ReLU activation, the learning exponent does improve, becoming larger than $ \beta / (\beta + 1) $. We have added a solution to modified differential equations describing this situation to the appendix. When training the second layer, the resulting fixed points for the error function activation correspond to the weight configuration of the soft committee machine. In contrast, the dynamics for the ReLU activation lead to a new fixed point that deviates from the soft committee machine configuration. Future research could focus on understanding the mechanisms behind improving the learning exponent for ReLU activation and why this improvement is absent for the error function activation. We provide the results in Appendix F.
>
> iii) Regarding the case where the covariance matrix is only partially degenerate: If we start from a situation with $ L = N/2 $ and a two-fold degeneracy, which is then lifted, we expect a smooth crossover to the case $ L = N $, in which no degeneracy occurs. We anticipate similar crossovers when higher-order degeneracies are lifted.
>
> iv) As for other activation functions, we have added solutions to differential equations describing the case $  K = M = 2 $  with a ReLU activation function. The results are provided in Appendix E.
>
> Additionally, we performed simulation experiments for $ K = M = 2 $ using various activation functions (see Figure 27), where a student is trained on the outputs of a teacher using synthetic input data. We find that the asymptotic power law remains unchanged compared to the error function activation.
>
> Regarding the questions raised:
>
> 1)  The dependence of the generalization error on $ N_l $ is intended to mimic the scaling with the size of the network, analogous to the study by Maloney et al. (2022) [1]. In the main manuscript, we examine a simplified setup with a diagonal covariance matrix, so each trainable component of the student vector corresponds to learning a direction in input space with a specific variance. In this sense, each entry of the student vector directly corresponds to an eigenvalue of the covariance matrix. We have revised this section to enhance clarity (cf. line 312). The behavior we observe in the simplified model does not change when we relax the condition of a diagonal covariance matrix; please see the new section in Appendix B.3.1.
>
> 2) We have included the widths of the plateaus in the figure to demonstrate the scaling with $ M $ and $ L$. Specifically, we have added a new figure (Figure 5, left panel) to illustrate this quantitatively.
>
> $\mathbf{Continuation\ of\ discussion\ in\ the\ next\ comment}$

---

> ### Author Response · Authors · 2024-11-23
> **Continuation of discussion**
>
> 3. We have the following intuition for the scaling of the plateau with $ M^2 $: Due to the scaling of the network output with $ 1/\sqrt{M}$, the generalization error and, hence, the gradient acquire a factor of $1/M $. Additionally, learning $ M $ features is more challenging than learning one feature, leading to another factor of $ M $ in the plateau length. Regarding the scaling of the plateau length with $ 1/L $: By normalizing the average covariance to one, the largest eigenvalue of the input covariance matrix scales with $ L $, making the corresponding direction easier to learn. This accelerates the departure from the plateau, thereby shortening the plateau length.
>
> 4. As discussed in our response to the main weaknesses, the results generalize to ReLU activations and to the case $ K > M $. A detailed discussion has been added to the appendix of the manuscript.
>
> 5. Regarding the suggestion to rename $ \alpha $ to avoid confusion, we note that the definition $ \alpha = p/N $ coincides with its use as training time. Since the network is updated after the presentation of each input example, the number of examples seen can be interpreted as a discrete-time measure. In the thermodynamic limit $ N \to \infty $ and $ p \to \infty $ with their ratio fixed, $ \alpha $ becomes a continuous variable that measures the progress of training.
>
> 6) We thank the reviewer for pointing out the incorrect naming of the polynomial and have corrected it accordingly.
>
> We hope that these clarifications and additions address the reviewer's concerns. We have updated the manuscript accordingly and are grateful for the opportunity to improve our work based on the insightful feedback.
>
> [1] Maloney, A., Roberts, D. A., & Sully, J. (2022). A solvable model of neural scaling laws. arXiv preprint arXiv:2210.16859.

---

> > ### Comment · Reviewer_WSqh · 2024-11-25
> >
> > Thank you for your responses. They do strengthen the paper, and I think (as I thought before) that it should be accepted. I updated my score from 6 to 8.

---

> > > ### Author Response · Authors · 2024-11-26
> > > **Appreciation for Valuable Feedback**
> > >
> > > Dear Reviewer WSqh,
> > >
> > > Thank you very much for your feedback! We are pleased to address your concern. We truly appreciate your positive and insightful comments to help us improve the quality of our work.
> > >
> > > Best regards,
> > >
> > > The authors

---

### Official Review · Reviewer_fBBP · 2024-11-04

**Soundness:** 4
**Presentation:** 3
**Contribution:** 3
**Rating:** 8
**Confidence:** 4

**Summary:**

This paper studies the learning dynamics of committee machines trained on random data with power law covariance structure. They utilize a hierarchy of order parameters which they can analytically close for linear activation functions. They obtain scaling laws with training observations $\alpha$ (also time for online learning) and model size $N$ when trained on spectra with $L$ distinct eigenvalues. The authors show a number of interesting effects including the disappearance of learning plateaus when $L$ increases and a transition from exponential convergence (for isotropic covariates) and power law convergence for large $L$. The authors demonstrate their derived theory is accurate for linear networks. They also examine the specialization transition from their hierarchy of ODEs and argue that the escape time scales inversely with $L$.

**Strengths:**

This paper studies an important problem, namely the theoretical origin of neural scaling laws, and studies this in nonlinear two layer committee machines. For linear activations, they can obtain very precise learning curves in terms of the spectrum. They obtain power law exponents for both training time $\alpha$ and number of student features $N$. For nonlinear networks, they predict escape times for the specialization transition in terms of $M,L$.

**Weaknesses:**

While the paper introduces a very promising approach of utilizing a hierarchy of order parameters to deal with power law structured covariates, most of the closed form theoretical predictions require restricting to the linear activation case. However, these results are also of interest. There are some remaining questions and issues, which if answered/addressed, could cause me to increase my score.

**Questions:**

1. Are most power laws derived under the assumption of random teacher vectors $T$? Can analysis be performed for a fixed realization of the teacher $T$?
2.  In Figure 3, how is the CIFAR-5M plot made? Are the true target labels from the dataset used or is an artificial teacher network used? If the target is generated from a synthetic teacher in this Figure, do the authors think they can use their theory to predict the learning curves for the true labels?
3. Could some of the plots like Figure 6 be plotted on log-log scale to see that the power law exponent depends on $\beta$?
4. In the linear network case, does the committee machine *learn features*? The neural tangent kernel for the committee machine with linear activations would be constant over training.
5. How sensitive are the solutions in the committee machine to the initialization of the student weights?

---

> ### Author Response · Authors · 2024-11-23
>
> We thank the reviewer for their thorough evaluation of our manuscript and for their constructive comments. We are pleased to address the questions and concerns raised.
>
> 1. Most of our results are not self-averaging with respect to a random teacher vector $\boldsymbol{B}$. The generalization error depends on the order parameter $T^{(1)} = \boldsymbol{B}^\mathrm{T} \boldsymbol{\Sigma} \boldsymbol{B}$ (inline equation above Eq.~(5)), which is not self-averaging. This lack of self-averaging arises from the power-law distribution of eigenvalues in $\boldsymbol{\Sigma}$ implying that the order parameter is dominated by a relatively small number of components of the teacher vector. Therefore, we have explicitly averaged over the teacher vectors in the results presented in our manuscript.
>
> 2. Regarding the CIFAR-5M plot in Figure 3, we used labels generated by a teacher network. The question about scaling laws with the true labels is intriguing. However, the classification performance of committee machines on CIFAR-5M is not sufficient to enter a scaling regime. We believe that training a more powerful architecture, such as a ResNet, on CIFAR-5M with the true labels would be necessary to observe scaling. We expect that the scaling in this case could exhibit a larger exponent than predicted for our setup, due to the training of multiple layers (see the new plot in Figure 25 of Appendix F in our revised manuscript). In general, scaling exponents depend on the network architectures. For CIFAR-10, Rosenfeld et al. [1] used wide residual networks (WRN) and found a scaling exponent $\beta \approx 0.5$with the model size, whereas Sharma and Kaplan [2] found $\beta \approx 0.23$ for a simple CNN.
>
>       To provide additional information on how the plot in Figure 3 was obtained: We used the parametrization from Eq. (80) to produce the results shown in the right panel. We tested our prediction for the generalization error from Eq. (79) on a student network trained on CIFAR-5M images using approximately $10^6$ input examples (see Nakkiran et al., 2021 for details on the dataset). We used only the first channel of the images, resulting in a total input dimension of $N = 1024$ after flattening. To approximate the true covariance matrix $\boldsymbol{\Sigma}$, we numerically estimated the feature-feature covariance matrix based on input examples from the training dataset. During training, we updated only the first $N_l$ entries of $\tilde{\boldsymbol{J}}$, resetting the remaining entries to their initial values after each iteration. Based on the spectra of the feature-feature covariance matrix depicted in Figure 14, we estimated $\beta \approx 0.3$ and used this spectrum to evaluate Eq. (80).
>
> 3. In response to the request about plotting Figure 6 on a log-log scale, we have replotted the left panel accordingly. We would like to point out that the inset of this plot was already on a log-log scale in the original version of our manuscript, allowing us to observe the dependence of the scaling exponent on $\beta$. Additionally, in response to another reviewer's request, we have added Figures 22 and 24 to the appendix, displaying the solution of our differential equations on a log-log scale for both error function and ReLU activation functions.
>
> 4. Regarding whether the committee machine learns features in the linear network case, for a linear activation function, it cannot learn features. As argued at the beginning of Section 4.1, the teacher output is fully determined by the average teacher vector, i.e., the student does not have access to the outputs of individual teacher units and cannot learn the features encoded by them.
>
> 5. Concerning the sensitivity of the solutions in the committee machine to the initialization of the student weights, as described by Eq. (12), the variance with which the student is randomly initialized determines the length of the symmetric plateau. According to Eq. (15), the length of this plateau depends on the teacher's initialization, not on the initialization of the student. Therefore, we expect the dependence on the student's initialization to be weaker than the dependence on the teacher's initialization and to have little effect on the asymptotic power law. In our numerical experiments, we have simultaneously changed the initialization of both the teacher and the student when performing averages.
>
>    Our differential equations show that the plateau length depends on the initial conditions, specifically on the $Q^{(l)}$, which are not self-averaging. However, the fluctuations in the plateau length are less pronounced compared to those arising from different initializations of the teacher.
>
> We hope these clarifications and additions adequately address the reviewer's concerns. We have revised the manuscript accordingly and appreciate the opportunity to refine our work based on the valuable feedback.

---

> ### Author Response · Authors · 2024-11-23
> **References for the previous response**
>
> [1] Rosenfeld, J. S., Rosenfeld, A., Belinkov, Y., & Shavit, N. (2019). A constructive prediction of the generalization error across scales. *arXiv preprint arXiv:1909.12673*.
>
> [2] Sharma, U., & Kaplan, J. (2022). Scaling laws from the data manifold dimension. *Journal of Machine Learning Research*, 23(9), 1-34.

---

> ### Author Response · Authors · 2024-11-26
>
> Dear Reviewer fBBP,
>
> We sincerely appreciate your valuable time and insightful comments on our work. As the deadline for the Author-Reviewer discussion draws near, we kindly request your feedback on whether our responses have effectively addressed your concerns.
>
> Thank you once again for your contributions, and we hope you have a wonderful day!
>
> Best regards,
>
> The Authors

---

> > ### Comment · Reviewer_fBBP · 2024-11-26
> >
> > I thank the authors for their detailed rebuttal and for answering all of my questions. I will update my score accordingly.

---

> > > ### Author Response · Authors · 2024-11-26
> > > **Appreciation for Valuable Feedback**
> > >
> > > Dear Reviewer fBBP,
> > >
> > > Thank you for your valuable feedback and for taking the time to review our responses. We appreciate your thoughtful comments and your efforts to help us improve our work.
> > >
> > > Best regards,
> > >
> > > The Authors

---

### Author Response · Authors · 2024-11-23
**Global Response**

We thank the reviewers for their detailed evaluation of our manuscript and their constructive feedback. Based on the recurring questions and suggestions, we have made revisions to the manuscript, which are marked in blue in the revised draft. Below, we summarize the key changes and additions.

1. To strengthen the visualization of the power-law behavior $\epsilon_g \propto \alpha^{-\beta/(\beta+1)}$, we now use a log-log plot in the left panel of Figure 6. We have also added Figures 22 and 24 to the appendix, displaying the solutions of our differential equations on a log-log scale for both error function and ReLU activation functions.

2. We have added a systematic investigation of the case where $ K > M $ to Appendix D, showing that the asymptotic power-law scaling $\epsilon_g \propto \alpha^{-\beta/(\beta+1)}$ and the plateau length scaling with $1/L$ remain valid and are independent of $K $.

3. For the ReLU activation function, we included solutions to differential equations describing the plateau and asymptotic cases for $ K = M = 2 $ in Appendix E and Figure 24. We performed simulations for $ K = M = 2 $ using various nonlinear activation functions and added the results to Figure 27. These findings support the conclusion that the asymptotic power-law scaling remains unchanged compared to the error function activation.

4. We extended our analysis to include cases where the second-layer weights are trainable, detailed in Appendix F. For ReLU activation, we show that the scaling exponent improves compared to the error function activation, which remains unchanged (see Figure 25). Differential equation solutions demonstrate that the error function activation causes the student weights to converge to the soft committee machine configuration, as shown in Figure 26. In contrast, ReLU activation converges to a distinct fixed point, leading to improved scaling exponents.

5. We tightened the upper bound of the time window for the power-law regime in the case of linear activations in Figures 2 and 9, and visualized the scaling by adding a line representing the power law. The linear scaling of the plateau length with $M^2 $ and $ 1/L $ is confirmed by simulation experiments in Figure 5.

6. For the training of only $ N_l $ student components for linear activations, we added a section in Appendix B.3.1. We compare the training dynamics of the student vector for a diagonal covariance matrix with those in the general case. Results are illustrated in Figures 15 and 16.

7. We have improved the readability and clarity of the main text and appendix throughout the manuscript. All typos and inaccuracies mentioned by the referees have been corrected.

8. To enhance readability, we moved some technical remarks to the appendix. Specifically:

    i) “Although $ \boldsymbol{U} $ is a rank-1 matrix, standard perturbation methods are not applicable to find the eigenvalues of the shifted matrix $ \boldsymbol{A}_1 + \eta \boldsymbol{U} $ because $ \boldsymbol{U} $ may have a large eigenvalue, making it unsuitable as a small perturbation.”

    ii) “The asymptotic convergence is governed by the smaller group of eigenvalues $ \left(2 - \sqrt{3}\right) \lambda_k $.”

9. Finally, we added a discussion on the limitations in the conclusion section.

We hope these updates adequately address the reviewers' concerns and enhance the clarity, generality, and relevance of our work. We are happy to make further adjustments or provide additional clarifications during the remaining discussion period.

---

### Meta-Review · Area_Chair_tuCU · 2024-12-20

**Metareview:**

This work investigates the presence of neural scaling laws in a two-layer student-teacher setup under the assumption that the data has a prescribed number of distinct eigenvalues from a power-law distribution, and in the limit of large data and dimension. Reviewers found the paper to be well-written and significant, yielding precise findings under an appropriately formulated setup. While acknowledging that the assumptions imposed greatly restricted interpretability and generality, the precision of the findings made up for these deficiencies. Reviewers appreciated the novelty of the results, and were in universal agreement that the paper should be accepted for publication.

**Additional Comments On Reviewer Discussion:**

Reviewer fBBP mentioned that the restriction to the linear activation case was unfortunate, but not disqualifying. They asked several assorted questions to the authors, all of which were addressed. Reviewer WSqh also drew attention to the strong assumptions limiting the impact of the work, but also acknowledged the difficulty with extending the analytical findings beyond the cases considered. They also asked about whether it is possible to investigate more general settings using numerical methods. In a lengthy response, the authors addressed each setting; Reviewer WSqh felt the inclusion of this material consistuted a significant strengthening of the work, recommending acceptance. Reviewer 5FqQ raised a few additional technical points that were addressed both in discussion, and in newly included supplementary material.

---

### Decision · Program_Chairs · 2025-01-22

Accept (Spotlight)